# GRADIENT-NORMALIZED SMOOTHNESS FOR OPTIMIZATION WITH APPROXIMATE HESSIANS

**Andrei Semenov** [*]
EPFL

**Martin Jaggi** [†]
EPFL

**Nikita Doikov** [‡]
Cornell University

## ABSTRACT

In this work, we develop new optimization algorithms that use approximate second-order information combined with the gradient regularization technique to achieve fast global convergence rates for both convex and non-convex objectives. The key innovation of our analysis is a novel notion called Gradient-Normalized Smoothness, which characterizes the maximum radius of a ball around the current point that yields a good relative approximation of the gradient field. Our theory establishes a natural intrinsic connection between Hessian approximation and the linearization of the gradient. Importantly, Gradient-Normalized Smoothness does not depend on the specific problem class of the objective functions, while effectively translating local information about the gradient field and Hessian approximation into the global behavior of the method. This new concept equips approximate second-order algorithms with universal global convergence guarantees, recovering state-of-the-art rates for functions with Hölder-continuous Hessians and third derivatives, Quasi-Self-Concordant functions, as well as smooth classes in first-order optimization. These rates are achieved automatically and extend to broader classes, such as generalized self-concordant functions. We demonstrate direct applications of our results for global linear rates in logistic regression and softmax problems with approximate Hessians, as well as in non-convex optimization using Fisher and Gauss-Newton approximations.

## 1 INTRODUCTION

**Motivation.** Numerical optimization methods that use *preconditioning* or *second-order* information—such as Newton-type methods—are extensively applied in machine learning, artificial intelligence, and scientific computing. While gradient-based methods—such as Gradient Descent—form a solid foundation for many large-scale applications due to their low per-iteration cost and well-established convergence theory, second-order methods are known to significantly accelerate convergence by taking into account the curvature information of the objective function. However, although the modern theory of second-order optimization establishes strong complexity guarantees for the Newton method with appropriate regularization techniques (Nesterov, 2018; Nesterov & Polyak, 2006; Cartis et al., 2011a; Doikov et al., 2024a), the theory for *inexact Hessians* is usually much more limited, suggesting that errors coming from the Hessian inexactness might drastically slow down convergence, causing the method to converge as slowly as Gradient Descent (Agafonov et al., 2024; Chayti et al., 2023). In this work, we aim to develop a new convergence theory for second-order methods with approximate Hessians, that matches state-of-the-art rates for the exact Newton method and bridges the geometry of the objective function with conditions on the Hessian approximation. The form of our method is very simple. For unconstrained minimization of the function $f$, using the standard Euclidean norm, we perform:

$$\boldsymbol{x}_{k+1} \;=\; \boldsymbol{x}_k - \left(\mathbf{H}_k + \tfrac{\|\nabla f(\boldsymbol{x}_k)\|}{\gamma_k}\mathbf{I}\right)^{-1}\nabla f(\boldsymbol{x}_k), \qquad k \geq 0, \tag{1}$$

where $\mathbf{H}_k \succeq \mathbf{0}$ is a Hessian approximation matrix, and $\gamma_k > 0$ is a (second-order) step-size. This parametrization ensures that each step is bounded, $\|\boldsymbol{x}_{k+1} - \boldsymbol{x}_k\| \leq \gamma_k$, and for $\mathbf{H}_k = \mathbf{0}$ we obtain iterations of the normalized gradient descent (Nesterov, 2024). Moreover, in the case of the exact

---

[*]Machine Learning and Optimization Laboratory. Email: andrii.semenov@epfl.ch

[†]Machine Learning and Optimization Laboratory. Email: martin.jaggi@epfl.ch

[‡]Work primarily carried out while the author was at EPFL. Email: nikita.doikov@cornell.edu

Hessians, $\mathbf{H}_k = \nabla^2 f(\boldsymbol{x}_k)$, the gradient regularization (1) was shown to achieve both very fast *quadratic local convergence*, as for the classical Newton method (Polyak, 2007), and strong global rates for a wide range of convex problem classes (Polyak, 2009; Doikov et al., 2024a; Doikov, 2023). In this paper, we relax $\mathbf{H}_k \approx \nabla^2 f(\boldsymbol{x}_k)$ to be a Hessian approximation in (1). We consider the following condition for our method:

$$\|\nabla^2 f(\boldsymbol{x}_k) - \mathbf{H}_k\| \leq \mathbf{C_1} + \mathbf{C_2}\|\nabla f(\boldsymbol{x}_k)\|^{1-\beta}, \qquad 0 \leq \beta \leq 1, \tag{2}$$

for certain $\mathbf{C_1}, \mathbf{C_2} \geq 0$, and $\beta$ is a fixed *approximation degree*. This condition appears to be essentially satisfied by many natural approximations of the Hessian, such as Fisher or Gauss-Newton approximations. For example, for the finite-sum structure of the objective $f(\boldsymbol{x}) = \sum_{i=1}^n f_i(\boldsymbol{x})$, that is popular in applications from machine learning and statistics, one can take the Fisher approximation,

$$\mathbf{H}_k := \sum_{i=1}^n \nabla f_i(\boldsymbol{x}_k)\nabla f_i(\boldsymbol{x}_k)^\top. \tag{3}$$

For simplicity, we consider here all gradients computed at the same point $\boldsymbol{x}_k$, while in practice the gradients can be taken from the past (Frantar et al., 2021) (see also (Martens, 2020) and (Kunstner et al., 2019) for an in-depth analysis of the Natural Gradient Descent and its variants). It appears that this approximation (3), e.g., for the logistic regression problem or softmax with linear models (Examples 6, 8) satisfies (2) with $\beta = 0$, and $\mathbf{C_1} = f^\star$ (the global optimum), which can be small or even zero for the well-separable data.

As a direct consequence of our new theory, we show that method (1), using the approximate Hessian (3), exhibits the *global linear rate*, as soon as $f^\star$ is sufficiently small. This stands in stark contrast to classical gradient methods, which typically achieve only sublinear convergence rates, unless additional assumptions—such as strong or uniform convexity—are imposed. Notably, our method remains formally first-order, relying solely on access to the first-order oracle.

Other examples include nonconvex problems with nonlinear operators, which satisfy (2) even with $\mathbf{C_1} = 0$ and $\beta = 0$, where $\mathbf{H}_k$ is a specific combinations of Gauss-Newton and Fisher matrices (see Examples 7, 8). We show that in these cases, when the degree $\beta$ of Hessian approximation is smaller than the degree of smoothness $\alpha$ (see the formal definition in Section 4), the errors coming from Taylor's approximation dominate the Hessian inexactness. In these situation ($\mathbf{C_1} \approx 0$, $\mathbf{C_2} > 0$, and $\alpha \geq \beta$), our method with inexact Hessians has the *same global rate* as the full Newton method ($\mathbf{C_1} = \mathbf{C_2} = 0$), see Figure 1.

**Contributions.** In this work, we develop a new framework for describing the global behavior of second-order methods using a universal (problem-class free) local characterization of the objective's gradient field and Hessian approximation, called *Gradient-Normalized Smoothness* (Section 2). We propose a unified treatment for the errors coming from both Hessian inexactness and Taylor's approximation, thereby showing an intrinsic connection between them. Our theory provides method (1) with a universal step-size rule for $\gamma_k$, which adapts automatically to the right problem class (which is described by the *degree of smoothness*, $0 \leq \alpha \leq 1$, introduced in Section 4) and the Hessian approximation error (2). See Table 1 for the summary of the complexity results covered by our Gradient-Normalized Smoothness, for particular problem classes.

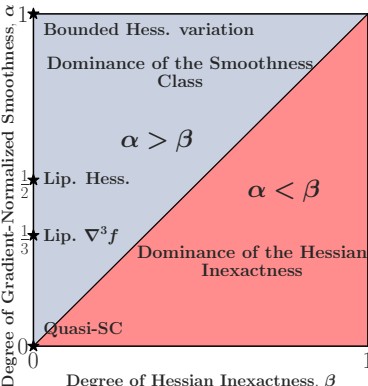

**Figure 1: Global Convergence Diagram for Algorithm 1.** We see that, for $\alpha \geq \beta$, the problem class of $f$ dominates the Hessian inexactness, and our method achieves the same rate as full Newton.

**(I) For the case of Exact Newton**, $\mathbf{H}_k = \nabla^2 f(\boldsymbol{x}_k)$, we ultimately recover the state-of-the-art rates obtained in (Doikov et al., 2024a; Doikov, 2023) for functions with Hölder continuous Hessian ($\frac{1}{2} \leq \alpha \leq 1$), Hölder continuous third derivative ($\frac{1}{3} \leq \alpha \leq \frac{1}{2}$), and Quasi-Self-Concordant functions ($\alpha = 0$). Our theory also extends to generalized Self-Concordant functions (Sun & Tran-Dinh, 2018), which correspond to $0 \leq \alpha \leq \frac{1}{2}$, establishing novel global rates in this range. Beyond that, the Gradient-Normalized Smoothness framework allows us to treat $(L_0, L_1)$-smooth functions (Zhang et al., 2019; Xie et al., 2024) from both first-order and second-order optimization (see examples in Section 2). Our convergence theory works both in convex and nonconvex cases (Theorems 1,2).

**(II) For the Inexact Hessian**, we use condition (2) to control the approximation errors, which is automatically covered by our notion of Gradient-Normalized Smoothness and provides us with the

| Problem class | Exact Case ($\mathbf{C_1} = \mathbf{C_2} = 0$) | | Inexact Hess. (ours) |
|---|---|---|---|
| Bounded Hess. variation | $O\left(\frac{M_0 D^2}{\varepsilon}\right)$ | (Nesterov, 2018) | $O\left(\frac{(M_0 + \mathbf{C_2})D^2}{\varepsilon} + \frac{\mathbf{C_1} D^2}{\varepsilon}\right)$ |
| Lip. Hess. | $O\left(\frac{M_{1/2} D^{3/2}}{\varepsilon^{1/2}}\right)$ | (Nesterov & Polyak, 2006) | $O\left(\frac{(M_{1/2} + \mathbf{C_2})D^{3/2}}{\varepsilon^{1/2}} + \frac{\mathbf{C_1} D^2}{\varepsilon}\right)$ |
| Lip. $\nabla^3 f$ | $O\left(\frac{M_{1/3} D^{4/3}}{\varepsilon^{1/3}}\right)$ | (Doikov et al., 2024a) | $O\left(\frac{(M_{1/3} + \mathbf{C_2})D^{4/3}}{\varepsilon^{1/3}} + \frac{\mathbf{C_1} D^2}{\varepsilon}\right)$ |
| Gen.-SC, $0 < \alpha \leq 1/2$ | $O\left(\frac{M_{1-\alpha} D^{1+\alpha}}{\varepsilon^\alpha}\right)$ | (ours) | $O\left(\frac{(M_{1-\alpha} + \mathbf{C_2})D^{1+\alpha}}{\varepsilon^\alpha} + \frac{\mathbf{C_1} D^2}{\varepsilon}\right)$ |
| Quasi-SC, $\alpha = 0$ | $\widetilde{O}\left(M_1 D\right)$ | (Doikov, 2023) | $\widetilde{O}\left((M_1 + \mathbf{C_2})D + \frac{\mathbf{C_1} D^2}{\varepsilon}\right)$ |

Table 1: Global complexities for our Algorithm 1 on different problem classes with convex objectives and using inexact Hessian. We show the number of iterations $K$ required to find $\varepsilon$-solution to our problem: $f(\boldsymbol{x}_K) - f^\star \leq \varepsilon$. Note that we recover state-of-the-art rates for the exact Newton ($\mathbf{C_1} = \mathbf{C_2} = 0$) in all particular cases, and extend them to the inexact Hessians. The global rates for the Generalized Self-Concordant (Gen.-SC) functions, introduced in (Sun & Tran-Dinh, 2018), are also novel in the exact case.

corresponding convergence rates. An interesting observation from our theory is that, in the regime $\alpha \geq \beta$ and $\mathbf{C_1} \approx 0$, *the smoothness class of the objective dominates the Hessian approximation*, and we recover the same rates as for the exact Hessian (see Fig. 1). As a by-product, we establish new global convergence rates for several practical problems (see Section 5) particularly when using approximate Hessian information, such as Fisher and Gauss-Newton matrices, which are popular in machine learning.

**(III) Numerical experiments** (Section 6 and Appendix B) illustrate our theory and confirm excellent performance of method (2) with our step-size selection and Hessian approximations.

**Related Work.** Using a scalable approximation of the Hessian matrix in Newton's method remains an attractive and popular approach to addressing the ill-conditioning of the function by better capturing the problem's geometry. Various examples include: low-rank approximations of the Hessian or quasi-Newton methods (Dennis & Moré, 1977; Jorge & Stephen, 2006; Rodomanov & Nesterov, 2021; Rodomanov, 2022; Jin & Mokhtari, 2023), spectral preconditioning (Ma et al., 2023; Zhang et al., 2023; Doikov et al., 2024b), first- and zeroth-order approximations (Cartis et al., 2012; Grapiglia et al., 2022; Doikov & Grapiglia, 2025), the Fisher and Gauss-Newton approximations (Nesterov, 2007; Kunstner et al., 2019; Arbel et al., 2023), stochastic subspaces or sketches (Cartis & Scheinberg, 2018; Gower et al., 2019; Fuji et al., 2022; Zhao et al., 2025; Hanzely, 2025), and many others. Modern techniques to globalize Newton's method, include the cubic regularization (Griewank, 1981; Nesterov & Polyak, 2006; Cartis et al., 2011a;b) and gradient regularization (Polyak, 2009; Mishchenko, 2023; Doikov & Nesterov, 2023; Doikov et al., 2024a; Doikov, 2023), that constitute the main basis of our work. Another popular approach consists in trust-region methods (Conn et al., 2000; Jiang et al., 2023; Xie et al., 2024), the notion Hessian stability (Karimireddy et al., 2018), and quasi-Newton methods with global convergence (Kamzolov et al., 2023; Scieur, 2024; Rodomanov, 2024; Jin et al., 2024). In recent years, we have seen more and more interesting deviations from the classical picture of complexity theory (Nemirovski & Yudin, 1983), with new important problem classes emerging from modern applications. These include the notion of *relative smoothness* (Bauschke et al., 2017; Lu et al., 2018), or $(L_0, L_1)$-*smoothness* (see (Zhang et al., 2019; Koloskova et al., 2023; Gorbunov et al., 2024; Vankov et al., 2024) and references therein), especially motivated by empirical smoothness properties of neural networks. While each of these new problem classes typically requires special attention—designing a new method and establishing the corresponding convergence theory—it is becoming increasingly evident that *the most natural optimization schemes are universal*, in the sense that they can automatically adapt to the appropriate degree of smoothness without requiring knowledge of any specific parameters (Nesterov, 2015; 2024).

**Notation.** Let us consider unconstrained minimization problem,

$$\min_{\boldsymbol{x} \in \mathbb{R}^n} f(\boldsymbol{x}), \tag{4}$$

where $f : \mathbb{R}^n \to \mathbb{R}$ is a differentiable function, that can be *non-convex*. Let $f^\star := \inf_{\boldsymbol{x} \in \mathbb{R}^n} f(\boldsymbol{x})$, which we assume to be finite: $f^\star > -\infty$. We denote by $\nabla f(\boldsymbol{x}) \in \mathbb{R}^n$ the gradient vector at point $\boldsymbol{x} \in \mathbb{R}^n$ and by $\nabla^2 f(\boldsymbol{x}) \in \mathbb{R}^{n \times n}$ the Hessian, which is a symmetric matrix. The third derivative, $\nabla^3 f(\boldsymbol{x})$, is a tri-linear symmetric form. We denote by $\nabla^3 f(\boldsymbol{x})[\boldsymbol{h}_1, \boldsymbol{h}_2, \boldsymbol{h}_3] \in \mathbb{R}$ its action onto arbitrary directions $\boldsymbol{h}_1, \boldsymbol{h}_2, \boldsymbol{h}_3 \in \mathbb{R}^n$. Let us fix a symmetric positive-definite matrix $\mathbf{B} \succ 0$, which

we use to define a pair of *global* Euclidean norms in our space:

$$\|\boldsymbol{h}\| \; := \; \langle \mathbf{B}\boldsymbol{h}, \boldsymbol{h} \rangle^{1/2}, \qquad \|\boldsymbol{s}\|_* \; := \; \langle \boldsymbol{s}, \mathbf{B}^{-1}\boldsymbol{s} \rangle^{1/2}, \qquad \boldsymbol{h}, \boldsymbol{s} \in \mathbb{R}^n,$$

which satisfy the Cauchy-Schwarz inequality: $|\langle \boldsymbol{s}, \boldsymbol{h} \rangle| \leq \|\boldsymbol{s}\|_* \|\boldsymbol{h}\|$. We use the dual norm to measure the size of the gradients. In the simplest case, we can set $\mathbf{B} := \mathbf{I}$ (identity matrix), which gives the classical Euclidean norm, while, in some cases, the use of a specific $\mathbf{B}$ can significantly improve the global geometry and convergence of our methods (see Section 5 for examples). Correspondingly, we use the induced spectral norm for symmetric matrices and multi-linear forms, e.g.

$$\|\nabla^2 f(\boldsymbol{x})\| \; := \; \max_{\boldsymbol{h}:\|\boldsymbol{h}\|\leq 1} |\langle \nabla f(\boldsymbol{x})\boldsymbol{h}, \boldsymbol{h} \rangle|, \quad \|\nabla^3 f(\boldsymbol{x})\| \; := \; \max_{\boldsymbol{h}:\|\boldsymbol{h}\|\leq 1} \nabla^3 f(\boldsymbol{x})[\boldsymbol{h}, \boldsymbol{h}, \boldsymbol{h}].$$

Along with the global norm in our space, we can also define the following *local norm* (Nesterov & Nemirovski, 1994), which is induced by the Hessian of the objective, for any $\boldsymbol{x} \in \mathbb{R}^n$: $\|\boldsymbol{h}\|_{\boldsymbol{x}}^2 := \langle \nabla^2 f(\boldsymbol{x})\boldsymbol{h}, \boldsymbol{h} \rangle$, $\boldsymbol{h} \in \mathbb{R}^n$. Note that we use this notion even for points where the Hessian is not positive definite. However, $\|\cdot\|_{\boldsymbol{x}}$ is a well-defined norm for $\boldsymbol{x}$ where $\nabla^2 f(\boldsymbol{x}) \succ \mathbf{0}$, which holds for strictly convex functions.

## 2 GRADIENT-NORMALIZED SMOOTHNESS

Our aim is to characterize and approximate the behavior of the gradient field $\nabla f(\cdot)$, induced by our objective. Along with it, we denote by $\mathbf{H}(\cdot) \in \mathbb{R}^{n \times n}$, the *matrix field* which assigns to every point $\boldsymbol{x} \in \mathbb{R}^n$ a symmetric positive-semidefinite matrix which serves as our Hessian approximation, $\mathbf{H}(\boldsymbol{x}) \approx \nabla^2 f(\boldsymbol{x})$. We will use this matrix directly in our algorithms (see Section 3 and corresponding examples). We would like to use it for the following *linear approximation* of the gradient field in a neigbourhood of the current point:

$$\nabla f(\boldsymbol{x} + \boldsymbol{h}) \; \approx \; \nabla f(\boldsymbol{x}) + \mathbf{H}(\boldsymbol{x})\boldsymbol{h}. \tag{5}$$

The examples include: $\mathbf{H} \equiv \nabla^2 f$, exact Hessian, which provides us with the Newton approximation in (5), or $\mathbf{H} \equiv \mathbf{0}$, zero matrix. The latter case corresponds to first-order methods.

**Definitions.** For a given $\gamma > 0$, we denote the ball $B_\gamma := \{\boldsymbol{h} : \|\boldsymbol{h}\| \leq \gamma\}$. Moreover, employing the local norm, we define the following *local region*, at point $\boldsymbol{x}$ and for an arbitrary direction $\boldsymbol{g} \in \mathbb{R}^n$:

$$\mathcal{O}_{\boldsymbol{x},\boldsymbol{g}} \; := \; \{\boldsymbol{h} : \|\boldsymbol{h}\|_{\boldsymbol{x}}^2 + \langle \boldsymbol{g}, \boldsymbol{h} \rangle \leq 0\}. \tag{6}$$

Note that for $\nabla^2 f(\boldsymbol{x}) \succ \mathbf{0}$ this set is an ellipsoid centered around the Newton direction: $\mathcal{O}_{\boldsymbol{x},\boldsymbol{g}} = \{\boldsymbol{h} : \|\boldsymbol{h} + \frac{1}{2}\nabla^2 f(\boldsymbol{x})^{-1}\boldsymbol{g}\|_{\boldsymbol{x}}^2 \leq \frac{1}{4}\|\boldsymbol{g}\|_{\boldsymbol{x},*}^2 := \frac{1}{4}\langle \boldsymbol{g}, \nabla^2 f(\boldsymbol{x})^{-1}\boldsymbol{g} \rangle\}$, and its geometry depends on the properties of the objective. For non-convex functions, $\mathcal{O}_{\boldsymbol{x},\boldsymbol{g}}$ can be unbounded. Nevertheless, we always intersect it with the Euclidean ball $B_\gamma$, thus working solely with bounded directions. Using our local regions, we introduce new characteristic, called the *Gradient-Normalized Smoothness*:

> **Definition 1.** *For any $\boldsymbol{x} \in \mathbb{R}^n$ and direction $\boldsymbol{g} \in \mathbb{R}^n$, denote*
>
> $$\gamma(\boldsymbol{x}, \boldsymbol{g}) := \max\{\gamma \geq 0 : \|\nabla f(\boldsymbol{x} + \boldsymbol{h}) - \nabla f(\boldsymbol{x}) - \mathbf{H}(\boldsymbol{x})\boldsymbol{h}\|_* \leq \tfrac{\|\boldsymbol{g}\|_* \|\boldsymbol{h}\|}{\gamma}, \forall \boldsymbol{h} \in B_\gamma \cap \mathcal{O}_{\boldsymbol{x},\boldsymbol{g}}\}.$$

Thus, quantity $\gamma(\boldsymbol{x}, \boldsymbol{g})$ describes the maximal radius of the Euclidean ball around point $\boldsymbol{x}$, within which the error of linear approximation of the gradient field (5) is relatively small across all feasible directions $\boldsymbol{h}$. Note that the local region $\mathcal{O}_{\boldsymbol{x},\boldsymbol{g}}$ only restricts the set of possible directions, and hence it can only improve $\gamma(\boldsymbol{x}, \boldsymbol{g})$. It appears that including set $\mathcal{O}_{\boldsymbol{x},\boldsymbol{g}}$ in the definition is crucial to make the modulus of smoothness $\gamma(\cdot)$ large enough, for second-order problem classes that we present below.

In order to better understand the definition, let us introduce the following univariate function, at a given point $\boldsymbol{x} \in \mathbb{R}^n$: $\rho(\gamma) := \min_{\boldsymbol{h} \in B_\gamma \cap \mathcal{O}_{\boldsymbol{x},\boldsymbol{g}}} \{\|\nabla f(\boldsymbol{x} + \boldsymbol{h}) - \nabla f(\boldsymbol{x}) - \mathbf{H}(\boldsymbol{x})\boldsymbol{h}\|_*^{-1} \|\boldsymbol{g}\|_* \|\boldsymbol{h}\|\}$, where $\gamma \geq 0$. Clearly, $\rho(\cdot)$ is monotonically decreasing, starting from some large limit [1] value $\rho(0)$. Its graph is shown in Fig. 2. Then, the value of $\gamma(\boldsymbol{x}, \boldsymbol{g})$ is the intersection of $\rho(\cdot)$ with the main diagonal. These observations also demonstrate *monotonicity in $\gamma$*: if the inequality from the definition holds for some $\gamma \geq 0$, then it also holds *for all $0 \leq \gamma' \leq \gamma$*, and, by definition, $\gamma(\boldsymbol{x}, \boldsymbol{g})$ is the *maximal possible* radius. Among all

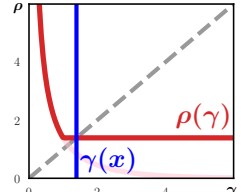

**Figure 2:** The plot of $\rho(\cdot)$ for $f(x) = e^x$. In this case, $\gamma(x) \equiv (e-2)^{-1} \approx 1.39$ for all $x \in \mathbb{R}$.

---

[1] The limit always exists when $f$ is sufficiently smooth at $\boldsymbol{x}$.

possible directions at $\boldsymbol{x}$, the most important is $\boldsymbol{g} = \nabla f(\boldsymbol{x})$. For that, we naturally define:

$$\gamma(\boldsymbol{x}) \quad := \quad \gamma(\boldsymbol{x}, \nabla f(\boldsymbol{x})).$$

As we will see in Section 3, $\gamma(\boldsymbol{x})$ provides us with the right *step-size* in our algorithm, that *automatically adjusts* to the best problem class and the degree of the Hessian approximation $\nabla^2 f(\boldsymbol{x}) \approx \mathbf{H}(\boldsymbol{x})$ at the current point. It is possible to generalize our results to Composite Optimization Problems (Appendix C), which includes constrained optimization and non-smooth regularizers. In this case, we need to use for $\boldsymbol{g}$ a *perturbed gradient direction*, that depends on the composite component.

**Basic Properties.** First, let us consider a stationary point $\boldsymbol{x}^\star$ which is a *strict local minimum*, so it holds: $\nabla f(\boldsymbol{x}^\star) = \mathbf{0}$ and $\nabla^2 f(\boldsymbol{x}^\star) \succ \mathbf{0}$. Then, by our definition we have $\gamma(\boldsymbol{x}^\star) = +\infty$, which means *no regularization* in our method. This implies that being in a neighborhood of the solution, the algorithm will switch to pure Newton steps, which confirms the intuition that the classical Newton's method has the best local behavior. Note that for quadratic functions and setting $\mathbf{H} := \nabla^2 f$, the linearization (5) is exact, and we also have $\gamma \equiv +\infty$. At the same time, when $\gamma(\boldsymbol{x})$ is small, it indicates a need for regularization.

Now, we can state how the Gradient-Normalized Smoothness $\gamma(\cdot)$ changes under simple operations [2]

1. *Scale-invariance.* Let $\gamma_f(\boldsymbol{x})$ be the Gradient-Normalized Smoothness for function $f$ and let $g := c \cdot f$ for some $c > 0$. Accordingly, we set $\mathbf{H}_g := c\mathbf{H}_f$. Then, $\gamma_g(\boldsymbol{x}) \equiv \gamma_f(\boldsymbol{x})$.

2. *Affine substitution.* Let $g(\boldsymbol{x}) := f(\mathbf{A}\boldsymbol{x} + \boldsymbol{b})$ for some invertible $\mathbf{A} \in \mathbb{R}^{n \times n}$ and $\boldsymbol{b} \in \mathbb{R}^n$. Set $\mathbf{H}_g(\boldsymbol{x}) := \mathbf{A}^\top \mathbf{H}_f(\mathbf{A}\boldsymbol{x} + \boldsymbol{b})\mathbf{A}$. Then, $\gamma_g(\boldsymbol{x}) \geq \gamma_f(\boldsymbol{x}) \cdot \|\mathbf{A}\|^{-1}$.

3. *Sum of functions.* Let $f := \sum_{i=1}^d f_i$. Then, $\gamma_f$ is bounded by the Harmonic mean:
$$\gamma_f(\boldsymbol{x}, \boldsymbol{g}) \quad \geq \quad \left( \textstyle\sum_{i=1}^d \gamma_{f_i}(\boldsymbol{x}, \boldsymbol{g})^{-1} \right)^{-1}, \qquad \boldsymbol{x}, \boldsymbol{g} \in \mathbb{R}^n.$$

4. *Hessian inexactness.* Let $\gamma_1(\boldsymbol{x})$ be the Gradient-Normalized Smoothness of $f$ when using matrix field $\mathbf{H}_1$. Let $\mathbf{H}_2$ be such that $\|\mathbf{H}_1(\boldsymbol{x}) - \mathbf{H}_2(\boldsymbol{x})\| \leq \|\nabla f(\boldsymbol{x})\| \cdot \gamma_{12}(\boldsymbol{x})^{-1}$, for a certain function $\gamma_{12}$. Then, the Gradient-Normalized Smoothness of $f$ when using $\mathbf{H}_2$ is bounded by the Harmonic mean: $\gamma_2(\boldsymbol{x}) \geq [\gamma_1(\boldsymbol{x})^{-1} + \gamma_{12}(\boldsymbol{x})^{-1}]^{-1}$.

**Examples.** Let us study the behavior of $\gamma(\cdot)$ when using the exact Hessian matrix, $\mathbf{H} \equiv \nabla^2 f$, under classical global second-order assumptions. Then, employing the known properties, we can translate it to an arbitrary Hessian approximation.

**Example 1** (Hölder Hessian). *Assume that $f$ has Hölder continuous Hessian of degree $\nu \in [0, 1]$: $\|\nabla^2 f(\boldsymbol{x}) - \nabla^2 f(\boldsymbol{y})\| \leq L_{2,\nu}\|\boldsymbol{x} - \boldsymbol{y}\|^\nu$, for all $\boldsymbol{x}, \boldsymbol{y} \in \mathbb{R}^n$. Then,*

$$\gamma(\boldsymbol{x}, \boldsymbol{g}) \quad \geq \quad \left( \tfrac{1+\nu}{L_{2,\nu}}\|\boldsymbol{g}\|_* \right)^{\frac{1}{1+\nu}}, \qquad \boldsymbol{x}, \boldsymbol{g} \in \mathbb{R}^n. \tag{7}$$

The most interesting are extreme cases: $\nu = 0$ (functions with bounded variation of the Hessian) and $\nu = 1$ (functions with Lipschitz Hessian) that gives, correspondingly:

$$\gamma(\boldsymbol{x}) \equiv \gamma(\boldsymbol{x}, \nabla f(\boldsymbol{x})) \geq \frac{\|\nabla f(\boldsymbol{x})\|_*}{L_{2,0}} \quad \text{and} \quad \gamma(\boldsymbol{x}) \equiv \gamma(\boldsymbol{x}, \nabla f(\boldsymbol{x})) \geq \sqrt{\frac{2\|\nabla f(\boldsymbol{x})\|_*}{L_{2,1}}}.$$

The following problem class was initially attributed to the third-order tensor methods (Birgin et al., 2017; Cartis et al., 2019; Nesterov, 2021a; Agafonov et al., 2024). Later on, as it was shown in (Nesterov, 2021b; Grapiglia & Nesterov, 2021; Doikov et al., 2024a), it appears to be appropriate for second-order optimization.

**Example 2** (Hölder Third Derivative). *Assume that $f$ is convex and its third derivative is Hölder of degree $\nu \in [0, 1]$: $\|\nabla^3 f(\boldsymbol{x}) - \nabla^3 f(\boldsymbol{y})\| \leq L_{3,\nu}\|\boldsymbol{x} - \boldsymbol{y}\|^\nu$, for all $\boldsymbol{x}, \boldsymbol{y} \in \mathbb{R}^n$. Then,*

$$\gamma(\boldsymbol{x}, \boldsymbol{g}) \quad \geq \quad \left( \tfrac{1+\nu}{2^{1+\nu}L_{3,\nu}}\|\boldsymbol{g}\|_* \right)^{\frac{1}{2+\nu}}, \qquad \boldsymbol{x}, \boldsymbol{g} \in \mathbb{R}^n.$$

---

[2]Missing proofs are provided in the appendix.

> **Example 3** (Quasi-Self-Concordance)**.** *Assume that $f$ is Quasi-Self-Concordant with parameter $M \geq 0$: $\langle \nabla^3 f(\boldsymbol{x})\boldsymbol{h}, \boldsymbol{h}, \boldsymbol{u} \rangle \leq M \|\boldsymbol{h}\|_{\boldsymbol{x}}^2 \|\boldsymbol{u}\|$, for all $\boldsymbol{x}, \boldsymbol{h}, \boldsymbol{u} \in \mathbb{R}^n$. Then,*
>
> $$\gamma(\boldsymbol{x}) \quad \geq \quad \tfrac{1}{M}.$$

The following examples of $(L_0, L_1)$-smooth functions are popular in the context of studying smoothness properties of neural networks, gradient clipping, and trust-region methods (Zhang et al., 2019; Koloskova et al., 2023; Xie et al., 2024).

> **Example 4** (($L_0, L_1$)-smooth functions (Zhang et al., 2019))**.** *Assume that $\|\nabla^2 f(\boldsymbol{x})\| \leq L_0 + L_1 \|\nabla f(\boldsymbol{x})\|_*$, for all $\boldsymbol{x} \in \mathbb{R}^n$. Then,*
>
> $$\gamma(\boldsymbol{x}, \boldsymbol{g}) \quad \geq \quad \tfrac{\|\boldsymbol{g}\|_*}{L_0 + L_1 \|\nabla f(\boldsymbol{x})\|_*} \cdot \big( 1 + \exp\big( \tfrac{\|\boldsymbol{g}\|_*}{\|\nabla f(\boldsymbol{x})\|_*} \big) \big)^{-1}, \qquad \boldsymbol{x}, \boldsymbol{g} \in \mathbb{R}^n.$$

> **Example 5** (Second-order $(M_0, M_1)$-smooth functions (Xie et al., 2024))**.** *Assume that $\|\nabla^2 f(\boldsymbol{x}) - \nabla^2 f(\boldsymbol{y})\| \leq (M_0 + M_1 \|\nabla f(\boldsymbol{x})\|_*) \|\boldsymbol{x} - \boldsymbol{y}\|$, for all $\boldsymbol{x}, \boldsymbol{y} \in \mathbb{R}^n$. Then,*
>
> $$\gamma(\boldsymbol{x}, \boldsymbol{g}) \quad \geq \quad \big( \tfrac{2\|\boldsymbol{g}\|_*}{L_0 + L_1 \|\nabla f(\boldsymbol{x})\|_*} \big)^{1/2}, \qquad \boldsymbol{x}, \boldsymbol{g} \in \mathbb{R}^n.$$

In practice, the objective function can belong to several of problem classes simultaneously, and optimal parameters can vary with $\boldsymbol{x}$. Therefore, it is important that the definition of $\gamma(\cdot)$ is *local*, just adjusting universally to the best of these cases. This allows the method to achieve the fastest rate.

## 3 ALGORITHM

The method is very simple.

---

**Algorithm 1** Gradient-Regularized Newton with Approximate Hessians

---

**Initialization:** $\boldsymbol{x}_0 \in \mathbb{R}^n$.

1: **for** $k \geq 0$ **do**
2:     Choose $\mathbf{H}(\boldsymbol{x}_k) \succeq \mathbf{0}$ and $\gamma_k > 0$.        ▷ In practice, use adaptive search for $\gamma_k$ (Alg. 3).
3:     Perform update: $\boldsymbol{x}_{k+1} \leftarrow \boldsymbol{x}_k - \big( \mathbf{H}(\boldsymbol{x}_k) + \tfrac{\|\nabla f(\boldsymbol{x}_k)\|_*}{\gamma_k} \mathbf{B} \big)^{-1} \nabla f(\boldsymbol{x}_k)$.
4: **end for**

---

In this algorithm, $\mathbf{H}(\boldsymbol{x}_k) = \mathbf{H}(\boldsymbol{x}_k)^\top \succeq \mathbf{0}$ could be the Hessian or its approximation, and $\gamma_k > 0$ is a second-order step-size. Our theory suggests to set $\boxed{\gamma_k = \gamma(\boldsymbol{x}_k)}$ which takes into account both the right problem class and the level of Hessian approximation. We can also use an adaptive search to choose the parameter $\gamma_k$ automatically, that we describe in Appendix D.

For simplicity of presentation, we assume that at each iteration $k \geq 0$ we solve the linear system exactly, which can be done easily in case the matrix $\mathbf{H}(\boldsymbol{x}_k)$ has a simple structure, e.g. a low-rank decomposition. We present several practical examples in Section 5. In general, using a linear system solver such as the conjugate gradient method, it will require only to compute matrix-vector products of the form $\mathbf{H}(\boldsymbol{x}_k)\boldsymbol{h}$, for an arbitrary $\boldsymbol{h} \in \mathbb{R}^n$. Such linear solver will typically have a linear rate of convergence due to strong convexity of the objective, and therefore it will require only a few matrix-vector products each iteration.

Using the first power of gradient norm as a normalizing constant is very natural due to several reasons:

- This ensures: $\|\boldsymbol{x}_{k+1} - \boldsymbol{x}_k\| \leq \gamma_k$, so the steps are normalized to be bounded in the Euclidean ball of a fixed radius $\gamma_k$, as in trust-region methods (Conn et al., 2000).
- When $\mathbf{H} \equiv \nabla^2 f$, the first power of the gradient norm ensures *local quadratic convergence*, as for classical Newton's method, and we are interested to choose $\gamma_k$ as large as possible (locally, being close to a solution, we admit $\gamma_k := +\infty$, no regularization).
- When $\mathbf{H} \equiv \mathbf{0}$, we obtain the *normalized gradient method* with a fixed preconditioning $\mathbf{B}$:

$$\boldsymbol{x}_{k+1} \quad = \quad \boldsymbol{x}_k - \tfrac{\gamma_k}{\|\nabla f(\boldsymbol{x}_k)\|_*} \mathbf{B}^{-1} \nabla f(\boldsymbol{x}_k)$$

In this case, our theory recovers the standard rates of the first-order smooth optimization.

**Global Progress.** With Definition 1, we prove the progress for each iteration of Algorithm 1:

> **Lemma 1.** *Let $0 \leq \gamma_k \leq \gamma(\boldsymbol{x}_k)$. Then*
>
> $$f(\boldsymbol{x}_k) - f(\boldsymbol{x}_{k+1}) \quad \geq \quad \frac{\gamma_k}{8} \cdot \frac{\|\nabla f(\boldsymbol{x}_{k+1})\|_*^2}{\|\nabla f(\boldsymbol{x}_k)\|_*}. \tag{8}$$

Inequality (8) does not depend on the structure of $\gamma(\boldsymbol{x}_k)$, showing that Algorithm 1 converges for an arbitrary well-defined $\gamma_k$. It is also important that this method converges for *any problem class* and *for any Hessian approximation*, as we did not specify them yet. Notably, for a specific problem class and for a specific $\mathbf{H}$, we can lower bound $\gamma(\cdot)$ globally as in the previous section, which yields state-of-the-art global convergence rates. Let us present a direct consequence of (8), which is a convergence for our algorithm in a general non-convex case.

> **Theorem 1** (Non-Convex Functions). *Let $K \geq 1$ be a fixed number of iterations and let (8) hold for every step. Assume that $\min\limits_{1 \leq i \leq K} \|\nabla f(\boldsymbol{x}_i)\|_* \geq \varepsilon$ and let $\gamma_\star := \min\limits_{1 \leq i \leq K} \gamma_i > 0$. Then,*
>
> $$K \quad \leq \quad \frac{8F_0}{\gamma_\star \varepsilon} + \log \frac{\|\nabla f(\boldsymbol{x}_0)\|_*}{\varepsilon}, \qquad where \quad F_0 := f(\boldsymbol{x}_0) - f^\star. \tag{9}$$

Note that up to now we did not say anything about smoothness assumptions on our objective, thus the result (9) is very general. Let us assume that $\mathbf{H}(\boldsymbol{x}_k) \equiv \nabla^2 f(\boldsymbol{x}_k) \succeq \mathbf{0}$, and that the Hessian is Hölder continuous of degree $\nu \in [0, 1]$, which according to (7) ensures that $\gamma_\star \geq [(1 + \nu)\varepsilon L_{2,\nu}^{-1}]^{1/(1+\nu)}$. Plugging this bound immediately provides us with the complexity of $O(1/\varepsilon^{(2+\nu)/(1+\nu)})$ iterations to find a point such that $\|\nabla f(\bar{\boldsymbol{x}})\|* \leq \varepsilon$. For $\nu = 1$, it gives $O(1/\varepsilon^{3/2})$, which corresponds to the rate of the cubically regularized Newton method (Nesterov & Polyak, 2006), and for every $0 < \nu \leq 1$, this complexity is strictly better than $O(1/\varepsilon^2)$ of the gradient descent (Nesterov, 2018). In the next sections we show the advanced convergence rates for our methods, under structural assumption on $\gamma(\cdot)$, that will recover state-of-the-art rates in all particular cases and allow for inexact Hessians.

## 4 GLOBAL CONVERGENCE THEORY

**Structural Assumption on $\gamma(\boldsymbol{x})$.** Let us assume that the Gradient-Normalized Smoothness $\gamma(\cdot)$ from Definition 1 admits the following structural lower bound, which is the harmonic mean of monomials of the gradient norm.

> For all $i$, there exist fixed degrees $0 \leq \alpha_i \leq 1$ and nonnegative coefficients $\{M_{1-\alpha}\}_{0 \leq \alpha \leq 1}$, such that:
>
> $$\gamma(\boldsymbol{x}) \quad \geq \quad \pi(\|\nabla f(\boldsymbol{x})\|_*) \quad := \quad \left( \sum_{i=1}^{d} \frac{M_{1-\alpha_i}}{\|\nabla f(\boldsymbol{x})\|_*^{\alpha_i}} \right)^{-1} \quad \geq \quad \frac{1}{d} \min_{1 \leq i \leq d} \frac{\|\nabla f(\boldsymbol{x})\|_*^{\alpha_i}}{M_{1-\alpha_i}}. \tag{10}$$

Here, the coefficients $\{M_{1-\alpha}\}_{0 \leq \alpha \leq 1}$ serve as the main complexity parameters. Note that all our Examples from Section 2 satisfy this assumption. In Examples 1, 2, 3, $\pi(\|\nabla f(\boldsymbol{x})\|_*) = \|\nabla f(\boldsymbol{x})\|_*^\alpha \cdot M_{1-\alpha}^{-1}$, for $\alpha \in [0, 1]$, is a simple monomial, and the structure in (10) is preserved under all basic operations with functions, such as summation. In what follows, we show that the lowest of the degrees of $\pi(\cdot)$ characterizes the class of smoothness, while additional exponents contribute to inexact Hessian (see basic properties in Section 2 and examples in Section 5). Defining the coefficients of $\pi(\|\nabla f(\boldsymbol{x})\|_*)$ from the set of $\{M_{1-\alpha}^{-1} : 0 \leq \alpha \leq 1\}$, where $M_{1-\alpha}$ corresponds to the smoothness constant of some problem class, we automatically set the state-of-the-art convergence rates for many partial (see Table 1).

> **Corollary 1** (Non-Convex Functions). *Let us choose $\gamma_k = \gamma(\boldsymbol{x}_k)$ in Algorithm 1, or by performing an adaptive search. Under assumption (10), we can bound $\gamma_\star \geq \pi(\varepsilon)$. Therefore, to ensure $\min\limits_{1 \leq i \leq K} \|\nabla f(\boldsymbol{x}_i)\|_* \leq \varepsilon$ it is enough to perform a number of iterations of*
>
> $$K \quad = \quad \left\lceil 8dF_0 \cdot \max_{1 \leq i \leq d} \frac{M_{1-\alpha_i}}{\varepsilon^{1+\alpha_i}} + \log \frac{\|\nabla f(\boldsymbol{x}_0)\|_*}{\varepsilon} \right\rceil.$$

**Convex Minimization.** Let us define $\alpha := \min_{1 \leq i \leq d} \alpha_i$ and introduce the following complexity

$$\mathcal{C}(\varepsilon) \quad := \quad \frac{d}{\alpha} \max_{1 \le i \le d} \left( \frac{M_{1-\alpha_i} D^{\alpha_i+1}}{\varepsilon^{\alpha_i-\alpha}} \right) \left( \frac{1}{\varepsilon^{\alpha}} - \frac{1}{F_0^{\alpha}} \right) \quad \text{for } \alpha > 0, \tag{11}$$

and for $\alpha \to 0$, we have the limit $\mathcal{C}(\varepsilon) := d \max_{1 \le i \le d} \left( \frac{M_{1-\alpha_i} D^{\alpha_i+1}}{\varepsilon^{\alpha_i-\alpha}} \right) \log\left( \frac{F_0}{\varepsilon} \right)$. For the particular cases $\mathbf{H} \equiv \nabla^2 f$ and $\mathbf{H} \equiv \mathbf{0}$, thus performing the full Newton method or performing the gradient descent, we denote the corresponding complexity by $\mathcal{C}_{\text{NEWTON}}(\varepsilon)$ and by $\mathcal{C}_{\text{GD}}(\varepsilon)$. Note that our theory covers these two important cases as well. We show that complexity $\mathcal{C}(\varepsilon)$ is the number of iteration required by Algorithm 1 to find the global solution, reflecting dynamics of Algorithm 1 and its ability to adapt to the right problem class. We denote by $D := \{\sup \|\boldsymbol{x} - \boldsymbol{x}^\star\| : f(\boldsymbol{x}) \le f(\boldsymbol{x}_0)\}$ the diameter of the initial sublevel set, which we assume to be bounded. We establish the main result.

> **Theorem 2** (Convex Functions). *Let us choose $\gamma_k = \gamma(\boldsymbol{x}_k)$ in Algorithm 1, or by using an adaptive search. Let $f$ be convex. Then, for any $\varepsilon > 0$, to ensure $f(\boldsymbol{x}_K) - f^\star \le \varepsilon$, it is enough to perform a number of iterations of*
>
> $$K \quad = \quad \left\lceil \mathcal{C}(\varepsilon) + 2 \log \frac{\|\nabla f(\boldsymbol{x}_0)\|_* D}{\varepsilon} \right\rceil.$$

We can extend this result for more general classes of *gradient-dominated* functions, that include strongly convex objectives and functions satisfying PL-condition, as well as improved rates for the gradient norm minimization, which we include in Appendix F.

**Recovering Rates for Particular Problem Classes with $\gamma(\boldsymbol{x})$.** To highlight the power of our result, let us consider a simple monomial $\gamma(\boldsymbol{x}) \ge \pi(\|\nabla f(\boldsymbol{x})\|_*) = \|\nabla f(\boldsymbol{x})\|_*^{\alpha} M_{1-\alpha}^{-1}$, for some $0 \le \alpha \le 1$ and $M_{1-\alpha} > 0$. For simplicity, we always assume $K \ge 2 \log \frac{\|\nabla f(\boldsymbol{x}_0)\|_* D}{\varepsilon}$. Then, in view of Theorem 2 and (11), we have the complexity of $O(1/\varepsilon^{\alpha})$, for $\alpha > 0$, that corresponds to the convergence rate inherent to problem classes from Examples 1, 2, 3. In case $\alpha = 0$, the complexity $K = \tilde{O}(M_1 D)$ yields the rate of the Newton method with the Gradient Regularization on Quasi-Self-Concordant functions (Doikov, 2023). As we see, Theorem 2 allows us to obtain a variety of convergence rates by plugging an appropriate global lower bound for $\gamma(\boldsymbol{x}_k)$. In Appendix G we show how the lower bound $\pi(\|\nabla f(\boldsymbol{x})\|)$ varies with the problem class. Corollary 7, shows how state-of-the-art rates for different problem classes are unified by our choice of $\gamma(\boldsymbol{x}_k)$ in Algorithm 1.

## 5 EFFECTIVE HESSIAN APPROXIMATIONS

**Our theory automatically covers a setup with inexact Hessian.** From Corollary 7 we see what happens to the rate when $\gamma(\boldsymbol{x})$ is lower bounded by a simple monomial. However, the case where $\pi(\|\nabla f(\boldsymbol{x})\|_*)$ is not a monomial is also interpretable with our theory. Corresponding convergence rate aligns with that of a second-order method with approximate Hessian, where the approximation error is bounded by some polynomial of $\|\nabla f(\boldsymbol{x})\|_*$. Theorem 2 already covers this case with the complexity of (11) for $\gamma(\boldsymbol{x})$ being bounded as in (10). However, some important practical cases of Hessian approximations can be described with a much simpler condition

$$\|\nabla^2 f(\boldsymbol{x}) - \mathbf{H}(\boldsymbol{x})\|_* \quad \le \quad \mathbf{C_1} + \mathbf{C_2} \|\nabla f(\boldsymbol{x})\|_*^{1-\beta}, \quad 0 \le \beta \le 1. \tag{12}$$

We provide examples of such $\mathbf{H}(\boldsymbol{x})$ that are particularly useful for machine learning applications. See extended examples in Appendix H.

> **Example 6** (Separable Optimization). *Let $f(\boldsymbol{x}) = \sum_{i=1}^{n} f_i(\boldsymbol{x})$, where $f_i(\boldsymbol{x}) := \ell(\langle \boldsymbol{a}_i, \boldsymbol{x} \rangle - b_i)$, for a convex nonnegative loss function. Consider logistic regression, $\ell(t) := \log(1 + \exp(t))$. Set $\mathbf{B} := \sum_{i=1}^{n} \boldsymbol{a}_i \boldsymbol{a}_i^\top$. Then, for the following Hessian approximation*
>
> $$\mathbf{H}(\boldsymbol{x}) \quad := \quad \sum_{i=1}^{n} \nabla f_i(\boldsymbol{x}) \nabla f_i(\boldsymbol{x})^\top = \sum_{i=1}^{n} \left( \ell'(\langle \boldsymbol{a}_i, \boldsymbol{x} \rangle - b_i) \right)^2 \boldsymbol{a}_i \boldsymbol{a}_i^\top \quad \succeq \quad \mathbf{0},$$
>
> *we have $\|\nabla^2 f(\boldsymbol{x}) - \mathbf{H}(\boldsymbol{x})\| \le f(\boldsymbol{x}) \le D\|\nabla f(\boldsymbol{x})\| + f^\star$, for $\boldsymbol{x} \in \mathcal{F}_0$.*

**Example 7** (Nonlinear Equations). *Let $\boldsymbol{u} : \mathbb{R}^n \to \mathbb{R}^d$ be a nonlinear operator, and set $f(\boldsymbol{x}) := \frac{1}{p}\|\boldsymbol{u}(\boldsymbol{x})\|^p \equiv \frac{1}{p}\langle \mathbf{G}\boldsymbol{u}(\boldsymbol{x}), \boldsymbol{u}(\boldsymbol{x})\rangle^{\frac{p}{2}}$, for some $\mathbf{G} = \mathbf{G}^\top \succ \mathbf{0}$ and $p \geq 2$. For this objective, we use:*

$$\mathbf{H}(\boldsymbol{x}) \quad := \quad \|\boldsymbol{u}(\boldsymbol{x})\|^{p-2}\nabla\boldsymbol{u}(\boldsymbol{x})^\top\mathbf{G}\nabla\boldsymbol{u}(\boldsymbol{x}) \ + \ \frac{p-2}{\|\boldsymbol{u}(\boldsymbol{x})\|^p}\nabla f(\boldsymbol{x})\nabla f(\boldsymbol{x})^\top \ \succeq \ \mathbf{0}. \tag{13}$$

*Assuming $\nabla\boldsymbol{u}(\boldsymbol{x})\mathbf{B}^{-1}\nabla\boldsymbol{u}(\boldsymbol{x})^\top \succeq \mu\mathbf{G}^{-1}$ and $\|\nabla^2\boldsymbol{u}(\boldsymbol{x})\| \leq \xi_1$, for some $\mu, \xi_1 > 0$, we have:*

$$\|\nabla^2 f(\boldsymbol{x}) - \mathbf{H}(\boldsymbol{x})\| \quad \leq \quad \xi_1\|\boldsymbol{u}(\boldsymbol{x})\|^{p-1} \quad \leq \quad \tfrac{\xi_1}{\sqrt{\mu}}\|\nabla f(\boldsymbol{x})\|_*. \tag{14}$$

**Example 8** (Soft Maximum). *In applications with multiclass classification, graph problems, and matrix games, we use $f(\boldsymbol{x}) := s(\boldsymbol{u}(\boldsymbol{x}))$, where $\boldsymbol{u} : \mathbb{R}^n \to \mathbb{R}^d$ is an operator (e.g. a linear or nonlinear model), and $s(\boldsymbol{y}) := \log\sum_{i=1}^d e^{y_i}$ is the LogSumExp loss. Note that $s(\cdot)$ is Quasi-Self-Concordant (Ex. 3), and $[\nabla s(\boldsymbol{y})]_i = \frac{e^{y_i}}{\sum_{j=1}^d e^{y_j}}$ is softmax. For this objective, we can use the following approximation of the Hessian in our algorithm:*

$$\mathbf{H}(\boldsymbol{x}) \quad := \quad \nabla\boldsymbol{u}(\boldsymbol{x})^\top\nabla^2 s(\boldsymbol{u}(\boldsymbol{x}))\nabla\boldsymbol{u}(\boldsymbol{x}) \quad \succeq \quad \mathbf{0}.$$

*Assuming that $\nabla\boldsymbol{u}(\boldsymbol{x})\mathbf{B}^{-1}\nabla\boldsymbol{u}(\boldsymbol{x})^\top \succeq \mu\mathbf{I}$ and $\|\nabla^2\boldsymbol{u}(\boldsymbol{x})\| \leq \xi_1$, for some $\mu, \xi_1 > 0$, we have:*

$$\|\nabla^2 f(\boldsymbol{x}) - \mathbf{H}(\boldsymbol{x})\| \quad \leq \quad \xi_1\|\nabla s(\boldsymbol{u}(\boldsymbol{x}))\| \quad \leq \quad \tfrac{\xi_1}{\sqrt{\mu}}\|\nabla f(\boldsymbol{x})\|_*.$$

**Connection between Gradient-Normalized Smoothness and the Hessian bound 12.** According to the "*Hessian inexactness*" property of $\gamma(\cdot)$, assuming 12, the Gradient Normalized Smoothness is bounded as: $\gamma(\boldsymbol{x}) \geq (\gamma_1(\boldsymbol{x})^{-1} + \frac{\mathbf{C_1}}{\|\nabla f(\boldsymbol{x})\|} + \frac{\mathbf{C_2}}{\|\nabla f(\boldsymbol{x})\|^\beta})^{-1}$, where $\gamma_1(\boldsymbol{x})$ is the Gradient Normalized Smoothness for the exact Hessian. In other words, if we know the lower bound $\pi(\cdot)$ for the exact Hessian (e.g. any of the problem classes above), then $\pi(\cdot)$ for the method with inexact Hessian can be computed in a form that satisfies the structural assumption 10. And we immediately obtain the complexity result for the method with inexact Hessian (Corollaries 2 and 3). We see that the total complexity of the method becomes the sum of the complexity for the exact case plus two additional terms that depend on $\mathbf{C_1}$, $\mathbf{C_2}$, and the degree of approximation $\beta$. Fig. 1 shows the interaction of the minimal degree of the monomial in $\pi(\cdot)$ and $\beta$ from Equation (12). And Corollary 2 finalizes Table 1.

**Corollary 2** (Inexact Hessian: Convex Functions). *Assume that condition (12) holds. Then, for any $\varepsilon > 0$, to ensure $f(\boldsymbol{x}_K) - f^\star \leq \varepsilon$, it is enough to perform a number of iterations of*

$$K \quad = \quad \widetilde{O}\Big(\mathcal{C}_{\text{NEWTON}}(\varepsilon) + \tfrac{\mathbf{C_1}D^2}{\varepsilon} + \tfrac{\mathbf{C_2}D^{1+\beta}}{\varepsilon^\beta}\Big), \quad \text{where } \widetilde{O}(\cdot) \text{ hides logarithmic factors.}$$

**Corollary 3** (Inexact Hessian: Non-Convex Functions). *Assume that condition (12) holds. Therefore, to ensure $\min_{1\leq i\leq K}\|\nabla f(\boldsymbol{x}_i)\|_* \leq \varepsilon$ it is enough to perform a number of iterations of*

$$K \quad = \quad \Big\lceil 8F_0 \cdot \Big(d\max_{1\leq i\leq d}\tfrac{M_{1-\alpha_i}}{\varepsilon^{1+\alpha_i}} + \tfrac{\mathbf{C_1}}{\varepsilon^2} + \tfrac{\mathbf{C_2}}{\varepsilon^{1+\beta}}\Big) + \log\tfrac{\|\nabla f(\boldsymbol{x}_0)\|_*}{\varepsilon}\Big\rceil.$$

## 6 EXPERIMENTS

Let us present illustrative numerical experiments that validate our theoretical findings. Extra experiments are in Appendix B, and the code is available at: https://github.com/epfml/grad-norm-smooth.

**Exact Hessian.** In Figure 3 **(a)**, we show convergence of Algorithm 1 with exact Hessian on the Softmax problem (LogSumExp) with linear models. We compare our adaptive search rule $\gamma_k = \gamma(\boldsymbol{x}_k)$ (see Lemma 1) with the strategies from Doikov et al. (2024a). We see that our theory predicts the best value of $\gamma_k$, which also serves an upper bound on empirical values of other adaptive search procedures **(b)**. In **(c)**, we compare our method with the gradient descent on the problem from Example 7. Our adaptive search denoted by "Func. Search". As a Hessian approximation for LogSumExp, we use Equation (19).

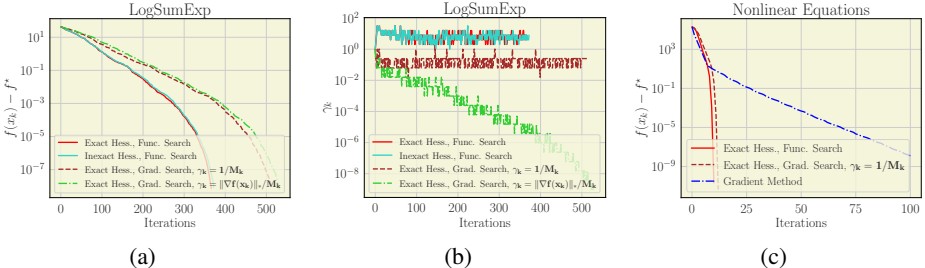

**Figure 3: Convergence of our methods.** We see that the second-order methods show outstanding performance, confirming our choice of the step-size $\gamma_k$.

**Inexact Hessian**. Our theory is compatible with different Hessian approximations we present in Table 2. In Appendix H we prove the bounds of $\|\nabla f(\boldsymbol{x}) - \mathbf{H}(\boldsymbol{x})\|_*$ for these approximations. Notably, condition 12 does not necessarily hold for all approximations considered; however, Theorem 2 allows us to derive the right convergence rate. We extensively evaluate our methods on both convex and non-convex benchmarks, including extensions of the Rosenbrock function Rosenbrock (1960) and difficult problems that involve Chebyshev polynomials Gürbüzbalaban & Overton (2012); Cartis et al. (2013). In Appendix B we also examine adaptive search, convergence, wall-clock time, and the numerical stability of our method with inexact Hessian on all problems from Table 2.

**Table 2: Hessian approximations aligned with our theory, evaluated on convex and non-convex problems.** Importantly, when $\boldsymbol{u}(\boldsymbol{x})$ is a linear operator, the "Inexact Hessian" turns out to be the full Hessian. Thus, we use the "Fisher Term of $\mathbf{H}$." in that case.

| Problem | Naming | Approximation |
|---|---|---|
| LogSumExp 15 | Weighted Gauss-Newton 19 | $\frac{1}{\mu} \mathbf{A}^\top \operatorname{Diag}\left(\operatorname{softmax}\left(\mathbf{A}, \boldsymbol{x}\right)\right) \mathbf{A}$ |
| Equations with linear operator | Fisher Term of $\mathbf{H}$ 21 | $\frac{p-2}{\|\boldsymbol{u}(\boldsymbol{x})\|^p} \nabla f(\boldsymbol{x}) \nabla f(\boldsymbol{x})^\top$ |
| Nonlinear Equations & Rosenbrock 23 | Inexact Hessian (Example 7) | $\|\boldsymbol{u}(\boldsymbol{x})\|^{p-2} \nabla \boldsymbol{u}(\boldsymbol{x})^\top \mathbf{G} \nabla \boldsymbol{u}(\boldsymbol{x}) + \frac{p-2}{\|\boldsymbol{u}(\boldsymbol{x})\|^p} \nabla f(\boldsymbol{x}) \nabla f(\boldsymbol{x})^\top$ |
| Nonlinear Equations & Chebyshev polynomials 24 | Inexact Hessian (Example 7) | $\|\boldsymbol{u}(\boldsymbol{x})\|^{p-2} \nabla \boldsymbol{u}(\boldsymbol{x})^\top \mathbf{G} \nabla \boldsymbol{u}(\boldsymbol{x}) + \frac{p-2}{\|\boldsymbol{u}(\boldsymbol{x})\|^p} \nabla f(\boldsymbol{x}) \nabla f(\boldsymbol{x})^\top$ |

# 7 DISCUSSION

We introduced the notion of Gradient-Normalized Smoothness, $\gamma(\boldsymbol{x})$, which treats the level of smoothness and the Hessian approximation error in a unified manner, leading to fast global rates for both convex and non-convex problems, and recovering various smoothness assumptions such as Hölder continuous Hessian or Quasi-Self-Concordant objectives. It is interesting to note that, in the case where the Hessian approximation satisfies bound (12) with $\beta = 0$ and $\mathbf{C_1} \approx 0$, the convergence rate of the method with an inexact Hessian is the same as that one of the full Newton method.

For example, for the logistic regression with the Fisher approximation of the Hessian (Example 6), we are able to establish the *global linear rate* of convergence in the case where the data is well-separable ($f^\star \approx 0$), complementing previously known results for gradient descent methods (Axiotis & Sviridenko, 2023) and for Newton-type methods (Karimireddy et al., 2018; Carmon et al., 2020; Doikov, 2023). Our theory extends beyond this setting to soft max and of non-linear models.

Another interesting example is the power loss function, $f(\boldsymbol{x}) = \frac{1}{p}\|\boldsymbol{x}\|^p$. As we show in Section G, this function is both *generalized self-concordant* and *uniformly convex*. These properties ensure an automatic global linear rate of our method for *all $p \geq 2$*.

While in this work we discuss only basic versions of the method, it is known in Convex Optimization that algorithms can be *accelerated*, achieving optimal convergence rates (Nesterov, 2018). Developing accelerated versions of our methods that automatically adapt to the problem's smoothness, as in (Carmon et al., 2022), while simultaneously adjusting to the potential inexactness in the Hessian, is an interesting direction that we leave for future research. It is also interesting to compare our results with recently proposed *anisotropic smoothness* (Laude & Patrinos, 2025), $\ell$-smoothness (Li et al., 2023; Tyurin, 2024), and recent advances on global convergence rates for the damped Newton method (Hanzely et al., 2024). We leave these comparisons for further investigation.

ACKNOWLEDGMENTS

This work was supported by the Swiss State Secretariat for Education, Research and Innovation (SERI) under contract number 22.00133.

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

CONTENTS

## A    EXTENDED RELATED WORK

**Connections and differences with other methods with gradient regularization.**

We propose Algorithm 1 with the only one hyperparameter—second-order step-size $\gamma_k$. Importantly, this step-size has an immediate natural interpretation of the radius of the ball within which we can rely on our approximate model, similarly in spirit to trust-region methods (Conn et al., 2000). However, instead of directly adding the ball constraints into minimization of the model, we apply the gradient regularization technique, considered in (Polyak, 2009; Ueda & Yamashita, 2009; Mishchenko, 2023; Doikov & Nesterov, 2024). This technique simplifies every step of the method, requiring to solve only one linear system per iteration. As compared to these previous works, we do not require to use an exact Hessian. We show that the inexactness condition on the matrix aligns well with the smoothness condition of the problem class, and our method works in a universal manner among the widest possible range of smoothness conditions.

In (Doikov et al., 2024a), it was shown that the Newton method with gradient regularization and adaptive search automatically adjusts to the problem classes with Hölder second or third derivative, and in (Doikov, 2023), the global linear rate of convergence on Quasi-Self-Concordant functions was proven for a similar method. In contrast to these works, we prove universal global rates for a method with inexact Hessian, both unifying analysis from (Doikov et al., 2024a; Doikov, 2023) and extending it beyond to the *generalized Self-Concordant functions* (Sun & Tran-Dinh, 2018). To the best of our knowledge, our global rates for generalized self-concordant functions are new.

An interesting insight from our analysis is that for different smoothness classes of the objective, we can allow different degrees of Hessian inexactness. Moreover, for popular choices of the Hessian approximations, such as the Fisher or Gauss-Newton approximations, the resulting methods perform *as well as for the method with the exact Hessian*. Using such approximations ultimately allows us to extend the gradient regularization technique for non-convex problems.

For the choice of the second-order step-size $\gamma_k$, we consider three possible strategies:

- The theoretical choice $\gamma_k = \gamma(\boldsymbol{x}_k, F'(\boldsymbol{x}_k))$, which is the best *local* value at $\boldsymbol{x}_k$ following our Definition 1.

- The constant choice, $\gamma_k \equiv \gamma_\star$ for a certain value of $\gamma_\star > 0$ (see Section D.1). Our value of $\gamma_\star$ is defined for a wide range of classes. However, despite seemingly too conservative, this rule recovers all state-of-the-art rates for the Newton method with gradient regularization, including those from (Doikov et al., 2024a; Doikov, 2023). To the best of our knowledge, this constant choice is new.

- Adaptive search to ensure the progress from Lemma 1. We discuss this strategy deeply in Appendix D.2. Importantly, our Algorithm 1 with adaptive search is equivalent to the Super-Universal Newton Method from (Doikov et al., 2024a). However, we use a different stopping condition in the adaptive search: we follow the condition based on the function value, while Doikov et al. (2024a) ensures a different inequality (17). Thus, our stopping condition is also suitable for *non-convex problems*.

From the theoretical perspective, equipped with the notion of Gradient-Normalized Smoothness, we cover the complexity results of the Super-Universal method, and extend the analysis to the Quasi-Self-Concordant functions (Doikov, 2023) and furthermore, to the generalized Self-Concordant functions (Sun & Tran-Dinh, 2018). Thus, even for the exact Hessians, we cover all results from (Doikov et al., 2024a; Doikov, 2023), and enhance them with the undiscovered rates for the generalized Self-Concordant class. Our framework also covers recently popular for the first-order methods ($L_0$, $L_1$)-functions (Zhang et al., 2019), and beyond. The breadth of rates for second-order methods covered by our framework (see Table 1) is unmatched in the existing literature.

**First state-of-the-art rates for inexact Hessians.**

The main power of our framework is the ability to *automatically* obtain the best rates for inexact Hessians. Moreover, thanks to Gradient-Normalized Smoothness and the structural assumption (10), we derive these results for convex (Theorem 2) and non-convex (Corollary 1) objectives *for free*. We additionally use condition (12) that connects the error coming from the Hessian inexactness with the gradient norm. We find such a condition very appealing for some practical Hessian approximations on fundamental problems such as logistic regression and softmax. Condition (12) allows us to write

complexity results with inexact Hessian in a compact and feasible manner. Combining both $\gamma_k$ and Eq. (12), we study the interplay between the smoothness class and the Hessian approximation—Figure 1—also a new result.

**Discussions of other types of smoothness.**

Since we are introducing a new notion for characterizing smoothness of the objective, we should mention previously established relative (Lu et al., 2018) and anisotropic (Laude & Patrinos, 2025) smoothness. While being theoretically appealing, these concepts are designed specifically for the first-order methods. Indeed, given a function that is anisotropic smooth or relative smooth (w.r.t. some reference function), and adding a quadratic function to this, will generally change the smoothness parameter of the objective. However, our Definition 1 is insensitive to such perturbations because of the second-order formulation. We also would like to highlight a very recently proposed, for the first-order, methods Glocal (Fox et al., 2025) and directional (Mishkin et al., 2025) smoothness. While we define Gradient-Normalized smoothness independently of the problem class in the way to use the local information to judge on the global behaviour of the method, the authors of Glocal smoothness are inspired by a similar plot. With their framework, it occurs that near the solution the curvature is much milder and line search or adaptive step-sizes can take advantage of that by increasing the steps, yielding faster progress in that region. Glocal smoothness allows a fair comparison between complexity results of many gradient-based methods, and the authors obtain better iteration-complexity bounds compared to using global smoothness only. The authors also claim that Gradient Method with line search can beat the accelerated gradient. These studies are worth further investigation.

# B    EXPERIMENTS

In this section, we explore two main aspects:

• how our adaptive search approach aligns with theoretical findings and with the previously established adaptive search variant (Doikov et al., 2024a) (see Appendix B.1 for experimental details); and

• the convergence behavior of Algorithm 1 for inexact and true Hessians[3] (see Appendices B.2, B.3, B.4).

We open-source our code at: https://github.com/epfml/grad-norm-smooth.

**Setup.** We consider two well-adopted problems: LogSumExp and Nonlinear Equations. We use a very simple setup for both. We define the LogSumExp problem as:

$$f(\boldsymbol{x}) \quad = \quad \mu \log \sum_{i=1}^{d} \exp\left( \frac{\langle \boldsymbol{a}_i, \boldsymbol{x} \rangle - b_i}{\mu} \right), \tag{15}$$

here $\boldsymbol{a}_i$ denotes the $i$-th row of the design matrix $\mathbf{A}$, and $\mu$ is a smoothing parameter varied across experiments (see Figure 6). The Nonlinear Equations problem is:

$$f(\boldsymbol{x}) \quad = \quad \tfrac{1}{p} \|\boldsymbol{u}(\boldsymbol{x})\|^p,$$

where the choice of the operator $\boldsymbol{u}(\boldsymbol{x})$ ranges from the linear model (Appendix B.3), to non-convex problems — Rosenbrock function, Chebyshev polynomials (Appendix B.4).

For the basic examples in the main part and in Appendix B.3 that can be run on real datasets, we utilize the linear operator $\boldsymbol{u}(\boldsymbol{x}) = \mathbf{A}\boldsymbol{x} - \boldsymbol{b}$. The design matrix $\mathbf{A}$ and the vector of labels $\boldsymbol{b}$ may represent a real or synthetic dataset. If we randomly generate both $\mathbf{A}$ and $\boldsymbol{b}$, we do this by sampling their entries independently from a uniform distribution $\mathrm{U}\left[-1, 1\right]$. In all experiments, our method is run with an adaptive search procedure (see Appendix D), which ensures the inequality (8) and thus selects the best value of $\gamma_k$. Furthermore, the values of $\gamma_k$ obtained through this adaptive scheme also serve as upper bounds for the empirical values obtained by the alternative adaptive search strategy studied in (Doikov et al., 2024a). In particular, we see that Algorithm 1 with inexact Hessians performs similarly to the method with true Hessian on the LogSumExp problem, which also highlights the power of our theoretical result: Indeed, the functions we consider in our experiments correspond to Examples 7 and 8, and in both cases, this aligns with the $\mathbf{C_1} = 0$ scenario in Table 1.

---

[3]We emphasize that there is no need for an extensive comparison between Algorithm 1 and variants of the Newton Method with alternative adaptive search procedures, as the practical improvement primarily lies in obtaining a better constant within the adaptive scheme, while the overall convergence pattern remains unchanged.

We use the following naming throughout the experiments:

1. **Exact Hess., Func. Search** or **Exact Newton:** stands for the partial case of Algorithm 1 using our adaptive search procedure (16), and $\mathbf{H}(\boldsymbol{x}) \equiv \nabla^2 f(\boldsymbol{x})$.

2. **Inexact Hess., Func. Search** or **Weighted Gauss-Newton:** refers to the variant of Algorithm 1 with Hessian approximation of the form (13) or (19), combined with our adaptive search (16).

3. **Exact Hess., Grad. Search, $\gamma_{\mathbf{k}} = \frac{1}{M_{\mathbf{k}}}$:** denotes the Gradient-Regularized Newton Method with adaptive search as in (Doikov et al., 2024a), using $\gamma_k := \frac{1}{M_k}$ and $M_k$ is chosen to satisfy the condition (17).

4. **Exact Hess., Grad. Search, $\gamma_{\mathbf{k}} = \frac{\|\nabla \mathbf{f(x_k)}\|_*}{M_{\mathbf{k}}}$:** denotes the Gradient-Regularized Newton Method with adaptive search as in (Doikov et al., 2024a), using $\gamma_k := \frac{\|\nabla f(\boldsymbol{x}_k)\|_*}{M_k}$ and $M_k$ is chosen to satisfy the condition (17).

5. **Gradient Method:** is a partial case of Algorithm 1 where $\mathbf{H}(\boldsymbol{x}) \equiv \mathbf{0}$ and $\mathbf{B} \equiv \mathbf{I}$, using our adaptive search strategy (16).

6. **Gauss-Newton:** the Gradient Method with a preconditioning matrix $\mathbf{B} \equiv \mathbf{A}^\top \mathbf{A}$, also combined with our adaptive search (16).

To demonstrate that our theoretical findings are reflected in practice, we validate the effects observed in (Fig. 3 (a, b, c)) on additional standard classification problems from `libsvm` (Chang & Lin, 2011).

We elaborate further on the connection between our theory and experiments in the following sections.

## B.1 COMPARISON OF ADAPTIVE SEARCH APPROACHES

First, we compare two different adaptive search procedures. Our approach, that is described in Appendix D, and uses a condition based on the function value:

$$f(\boldsymbol{x}_k) - f(\boldsymbol{x}_{k+1}) \geq \frac{\gamma_k}{8} \frac{\|\nabla f(\boldsymbol{x}_{k+1})\|_*^2}{\|\nabla f(\boldsymbol{x}_k)\|_*}. \tag{16}$$

And the adaptive search strategy from (Doikov et al., 2024a), where the sequence $M_k$ is selected in order to ensure the following condition

$$\langle \nabla f(\boldsymbol{x}_{k+1}), \boldsymbol{x}_k - \boldsymbol{x}_{k+1} \rangle \geq \frac{\|\nabla f(\boldsymbol{x}_{k+1})\|_*^2}{4 M_k \|\nabla f(\boldsymbol{x}_k)\|_*^l}, \quad \text{where} \ \ l \in \left[\frac{2}{3}, 1\right]. \tag{17}$$

Importantly, our theory covers this adaptive search scheme as a special case. Specifically, by selecting $\gamma_k$ as a simple monomial $\pi\left(\|\nabla f(\boldsymbol{x})\|_*\right) = \frac{1}{M_k}\|\nabla f(\boldsymbol{x})\|_*^{1-\alpha}$, we recover the behavior of the method from (Doikov et al., 2024a). This connection is formally established in Section 4.

From our experiments, we observe a nice property of $\gamma_k$: it tends to remain nearly constant throughout the iterations of our method (1). Moreover, our value of $\gamma_k$ is typically larger than the one obtained by the alternative adaptive search procedure proposed in (Doikov et al., 2024a). For representing (Fig. 3 (a)), we use a randomly generated matrix $\mathbf{A} \in \mathbb{R}^{1000 \times 500}$ and set a large factor $\mu = 1$, which simplifies the problem and yields a more numerically stable and smooth approximation of the maximum function. By varying both $\mu$ and the data size, we present in Figures 4 and 5 a comparison of convergence behavior and the Gradient-Normalized Smoothness values measured throughout training. These results further support our theoretical findings by illustrating how $\gamma_k$, computed via our adaptive search procedure, either increases over time or remains a sufficiently large constant. In both cases, it provides an upper bound for the corresponding $\gamma_k$ values observed under other variants of the Newton Method and the Gradient Method (i.e., when $\mathbf{H}(\boldsymbol{x}) \equiv \mathbf{0}$).

## B.2 INEXACT HESSIAN: LOGSUMEXP

We see that our theory correctly predicts the behavior of the Gradient-Normalized Smoothness across various practical problems. Now, let us focus on the performance of our Algorithm 1 with inexact Hessian. Our goal is to show that our method with inexact Hessian achieves the rate of the full Newton Method and, thus, significantly outperforms the Gradient Method. It is known, that preconditioning the gradient descent direction with an informative matrix substantially improves the

convergence of first-order methods. For instance, one may consider using a method with the inverse curvature matrix $\mathbf{B}^{-1}$ or a family of polynomials (Doikov & Rodomanov, 2023) as preconditioner in $\boldsymbol{x}_{k+1} = \boldsymbol{x}_k - \gamma_k \mathbf{B}^{-1} \frac{\nabla f(\boldsymbol{x}_k)}{\|\nabla f(\boldsymbol{x}_k)\|_*}$, instead of the standard Gradient Method that does not take into account the physics of the problem and uses $\mathbf{B} \equiv \mathbf{I}$. Building on this insight, we outline the method we call the *Gauss-Newton* as our algorithm with

$$\mathbf{H}(\boldsymbol{x}) \;\; \coloneqq \;\; \mathbf{A}^\top \mathbf{A}. \tag{18}$$

Note that the Newton Method with matrix (18) corresponds to the classical Gauss-Newton method for linear models, where the Jacobian is simply $\mathbf{A}$. In Figures 6 and 7, we show that the Gauss-Newton algorithm significantly outperforms the plain Gradient Method with $\mathbf{B} := \mathbf{I}$.

However, our theory suggests that Algorithm 1 with Hessian approximation that satisfies condition (2) with $\mathbf{C_1} \approx 0$ should achieve the same convergence rate as the full Newton and outperform both the Gradient and Gauss-Newton methods on the LogSumExp problem. For that, we consider the following approximation that we call the *Weighted Gauss-Newton*:

$$\mathbf{H}(\mathbf{x}) \;\; \coloneqq \;\; \tfrac{1}{\mu} \mathbf{A}^\top \mathrm{Diag}\left(\mathrm{smax}\left(\mathbf{A}, \boldsymbol{x}\right)\right) \mathbf{A}, \quad \left[\mathrm{smax}\left(\mathbf{A}, \boldsymbol{x}\right)\right]_k \;\; \coloneqq \;\; \frac{\exp\left[\frac{1}{\mu}\left(\langle \boldsymbol{a}_k, \boldsymbol{x}\rangle - b_k\right)\right]}{\sum\limits_{j=1}^{d} \exp\left[\frac{1}{\mu}\left(\langle \boldsymbol{a}_j, \boldsymbol{x}\rangle - b_j\right)\right]}. \tag{19}$$

In other words, Equation (19) corresponds to Equation (18) with entries of $\mathbf{A}$, weighted by $\mathrm{smax}\left(\cdot\right) \in \mathbb{R}^d$. Since the Hessian of the LogSumExp objective (15) is given by

$$\begin{aligned}
\nabla^2 f(\boldsymbol{x}) \;\; &= \;\; \tfrac{1}{\mu} \mathbf{A}^\top \left(\mathrm{Diag}\left(\mathrm{smax}\left(\mathbf{A}, \boldsymbol{x}\right)\right) - \mathrm{smax}\left(\mathbf{A}, \boldsymbol{x}\right) \mathrm{smax}\left(\mathbf{A}, \boldsymbol{x}\right)^\top\right) \mathbf{A} \\
&= \;\; \mathbf{H}(\boldsymbol{x}) - \tfrac{1}{\mu} \mathbf{A}^\top \left(\mathrm{smax}\left(\mathbf{A}, \boldsymbol{x}\right) \mathrm{smax}\left(\mathbf{A}, \boldsymbol{x}\right)^\top\right) \mathbf{A} \;\; = \;\; \mathbf{H}(\boldsymbol{x}) - \tfrac{1}{\mu} \nabla f(\boldsymbol{x}) \nabla f(\boldsymbol{x})^\top,
\end{aligned}$$

we see that with such approximation $\mathbf{H}(\boldsymbol{x})$ we can perfectly match the condition (2) (refer to Appendix H for more details). This is one of the main examples of approximations that are covered by our analysis.

We show that the Newton Method with Hessian approximation (19) and with $\gamma$ selected by an adaptive search (Algorithm 3), performs comparably to the Newton Method with the exact Hessian and the same adaptive search procedure, see Figure 6. Moreover, as our theory suggests, the LogSumExp objective corresponds to the case $\mathbf{C_1} = 0$ with $\mathbf{C_2} > 0$ being some constant (see Example 8). Thus, according to the results in Table 1, which are also visualized in Figure 1, it places us in the regime where the smoothness of the objective dominates the Hessian inexactness, i.e., we should observe the rate of the full Newton Method for our Algorithm 1 with approximation (19). We actually see this behavior in further examples as well.

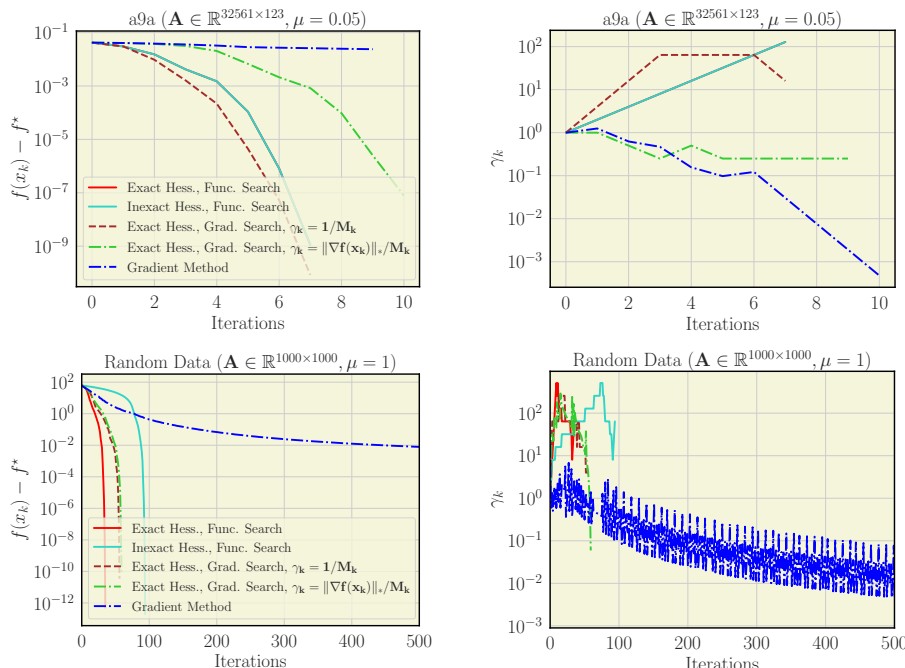

**Figure 4: LogSumExp objective. Convergence (left) and the Gradient-Normalized Smoothness (right).** An interesting effect we observe in the figure is that, e.g., on the a9a dataset, ever since Algorithm 1 exhibits a sharper convergence on the log-scaled plot, its corresponding $\gamma_k$ values start increasing more rapidly than those produced by other adaptive search procedures. Additionally, for a9a, we observe an almost perfect match between the exact Hessian and its approximation, likely due to the simplicity of the problem (the Newton Method need only 8 iterations to converge). Notably, we also observe a predictable, rapid decrease in $\gamma_k$ for the Gradient Method on a relatively hard problem ($\sim 30$ iterations of the Newton Method to converge).

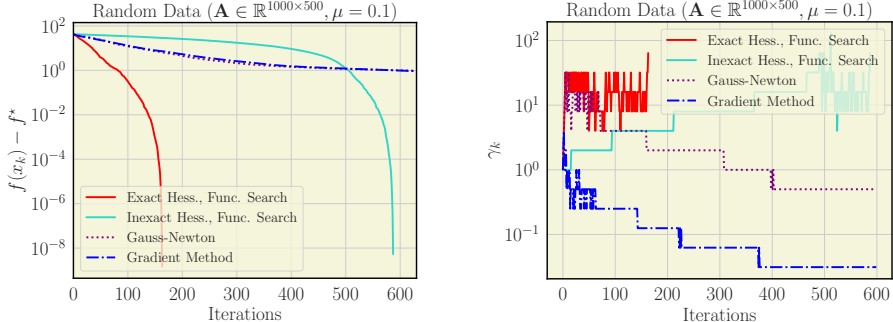

**Figure 5: LogSumExp objective. Convergence (left) and the Gradient-Normalized Smoothness (right).** In the left figure, we may observe almost identical performance of Gradient Method and Gauss-Newton on a sufficiently hard task ($\sim 150$ iterations of Newton Method to converge). However, we see that the empirical values of Gradient-Normalized Smoothness for Gauss-Newton decrease significantly more slowly than those for Gradient Method. For our future experiments, this indicates a particular power of Gradient Method with Gauss-Newton preconditioner that adapts better to the physics of the problem.

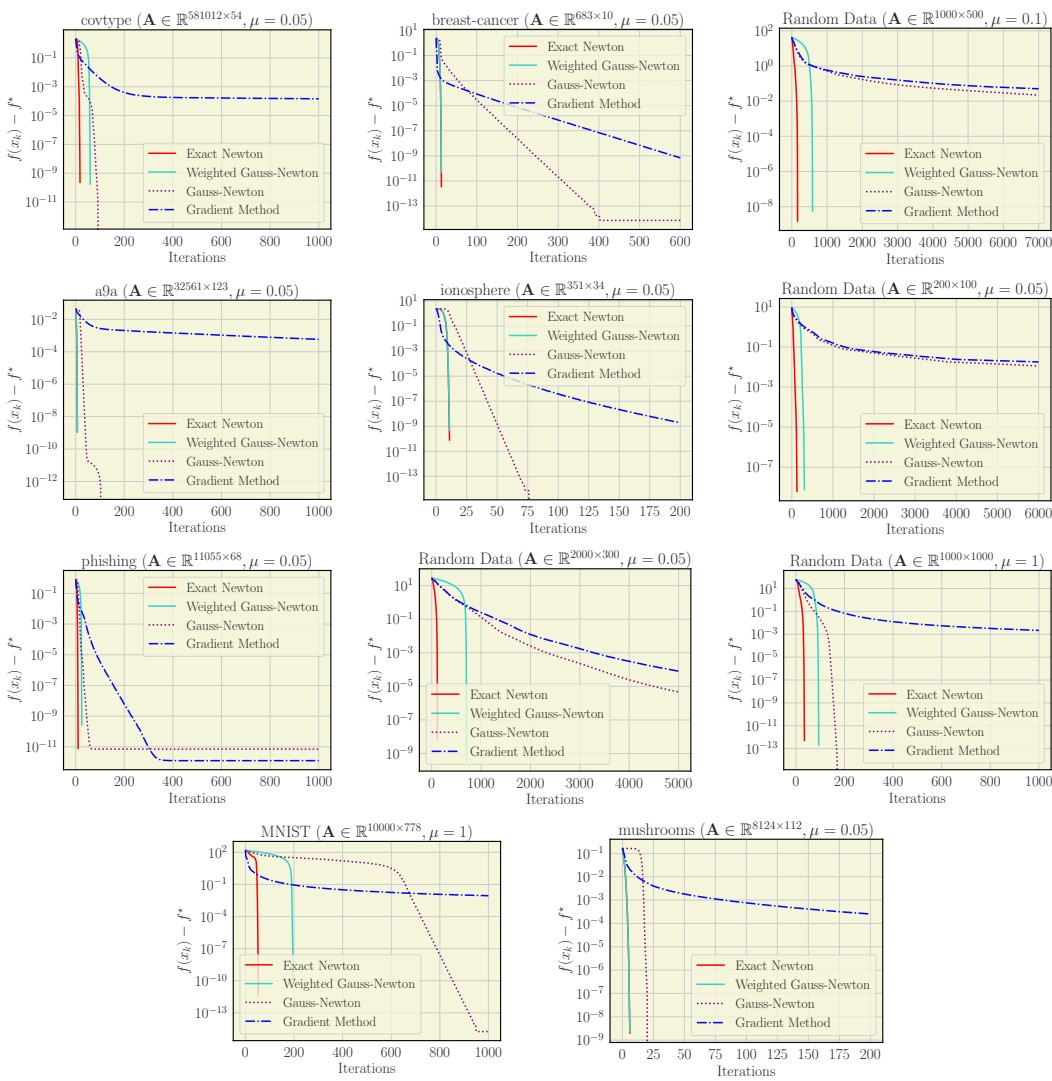

**Figure 6: LogSumExp objective. Our method with approximation noticeably outperforms both Gradient Method and its Gauss-Newton preconditioned variant.** According to our theoretical results, this problem falls into a regime where the convergence rate matches that of the full Newton method. Indeed, we observe consistent convergence behavior between the Exact and Inexact variants of our method across a wide range of examples.

### B.3 Nonlinear Equations: Studying the Degree of Smoothness

Let us consider a setup with the Nonlinear Equations objective defined in Example 7:

$$f(\boldsymbol{x}) \ := \ \tfrac{1}{p}\|\boldsymbol{u}(\boldsymbol{x})\|^p \ \equiv \ \tfrac{1}{p}\langle\mathbf{G}\boldsymbol{u}(\boldsymbol{x}), \boldsymbol{u}(\boldsymbol{x})\rangle^{\frac{p}{2}}, \quad \mathbf{G} = \mathbf{G}^\top \succ 0, \quad p \geq 2.$$

**The Exact Case.** First, we study the case of the linear operator $\boldsymbol{u}(\cdot)$. We elaborate the nonlinear case in Appendix B.4. Here, the Hessian approximation $\mathbf{H}(\boldsymbol{x})$ suggested by our theory is equal to the true Hessian if the operator $u(\boldsymbol{x})$ is linear. Indeed, from Example 7 and taking into account that $\boldsymbol{u}(\boldsymbol{x}) = \mathbf{A}\boldsymbol{x} - \boldsymbol{b}$ and $\nabla^2 \boldsymbol{u}(\boldsymbol{x}) = \mathbf{0}$, we have

$$\nabla^2 f(\boldsymbol{x}) \ = \ \|\boldsymbol{u}(\boldsymbol{x})\|^{p-2}\mathbf{A}^\top\mathbf{G}\mathbf{A} + \tfrac{p-2}{\|\boldsymbol{u}(\boldsymbol{x})\|^p}\nabla f(\boldsymbol{x})\nabla f(\boldsymbol{x})^\top \ \equiv \ \mathbf{H}(\boldsymbol{x}). \tag{20}$$

Thus, we demonstrate the comparison of the exact Newton Method with the Gradient Method and the Gauss-Newton. Importantly, if $p = 2$, $\mathbf{H}(\boldsymbol{x})$ appears to be the scaled Gauss-Newton matrix, while for any $p > 2$ we also have an additional rank-one term. As increasing $p$ complicates our problem, in this experiment we vary the power $p$, starting with a simple quadratic problem, $p = 2$, and extending up to $p = 5$. Our results show that, for all values $p$ considered, the Gradient Method preconditioned with the curvature matrix $\mathbf{B} \equiv \mathbf{A}^\top\mathbf{A}$ performs comparably to, or even outperforms, the variant of our algorithm that uses the full Hessian $\nabla^2 f(\boldsymbol{x}) \equiv \mathbf{H}(\boldsymbol{x})$. This observation highlights a significant consequence of our theoretical analysis, particularly in relation to Example 7.

**Towards Inexact Hessians for Linear Operators.** Nevertheless the Hessian approximation $\mathbf{H}(\boldsymbol{x})$ suggested in Example 7 is equivalent to the exact Hessian when the operator $\boldsymbol{u}(\cdot)$ is linear, one still can treat this matrix as a sum of two: a rank-one term $\tfrac{p-2}{\|\boldsymbol{u}(\boldsymbol{x})\|^p}\nabla f(\boldsymbol{x})\nabla f(\boldsymbol{x})^\top$, and the summand with Jacobians $\|\boldsymbol{u}(\boldsymbol{x})\|^{p-2}\mathbf{A}^\top\mathbf{B}\mathbf{A}$. The latter term can be viewed as the Gauss-Newton preconditioner scaled by $\|\boldsymbol{u}(\boldsymbol{x})\|^{p-2}$, while the last term corresponds to the Fisher approximation up to a multiplicative factor $\tfrac{p-2}{\|\boldsymbol{u}(\boldsymbol{x})\|^p}$. Hence, we can consider those chunks of the Hessian $\mathbf{H}(\boldsymbol{x})$ from Example 7 as a potential approximations. While usage of the scaled Gauss-Newton term of $\mathbf{H}(\boldsymbol{x})$ resembles the Gauss-Newton method from our previous experiments, the Fisher term of $\mathbf{H}(\boldsymbol{x})$ arouses interest. Therefore, in this part of our experimental validations, we consider the Exact Newton Method, Gradient Method and the following algorithm

$$\boldsymbol{x}_{k+1} \ = \ \boldsymbol{x}_k - \left(\tfrac{p-2}{\|\boldsymbol{u}(\boldsymbol{x})\|^p}\nabla f(\boldsymbol{x}_k)\nabla f(\boldsymbol{x}_k)^\top + \tfrac{\|\nabla f(\boldsymbol{x}_k)\|_*}{\gamma_k}\mathbf{A}^\top\mathbf{A}\right)^{-1}\nabla f(\boldsymbol{x}_k). \tag{21}$$

I.e., the derived update corresponds to the Gauss-Newton method with rank-one correction. In experiments, we call this method — **Fisher Term of H**.

We run the comparison of methods on the same the Nonlinear Equations problem on MNIST and a small, randomly generated dataset. Note, that for the random dataset, we chose exactly the same runs of the Exact Newton and the Gradient Method as in Figure 7. Results for this experiment are in Figures 8 and 9. Importantly, our version of Algorithm 1 with Fisher approximation not only achieves the comparable convergence as the Exact Newton, but also a way more faster in terms of the wall-clock time. In Figure 8, we show that our method with the Fisher-type approximation consistently outperforms the Gradient Method and, in some cases, even the Exact Newton Method, while having almost as cheap per-iteration cost as the Gradient Method, especially if the problem is a small dimensional.

We pose that, in practice, the most time-consuming part of computations for Algorithm 1 with approximation from (21) are in the inverting of the $\mathbf{A}^\top\mathbf{A}$, however the overall computational time can be significantly accelerated via the usage of the Woodbury-Sherman-Morrison formula (Max, 1950) to exactly compute the invert in Equation (21). We do this in practice and achieve almost the same speed on small-scale problems, while having an insignificant slowdown on large-scale ones.

Besides the results in Figure 8, we also investigate the behavior of our method with Fisher-like approximation on the Nonlinear Equations problem varying the power $p$. Throughout those experiments with modifications of $p$, we observe a consistent improvement of our method with approximation over the Gradient Method and the comparable performance of it compared to the Exact Newton Method. We summarize our experimental observations in Figure 9. This result suggests that one of the approximations suggested by our theory (see Appendix H) not only resembles the converge of the full Newton, but also is very cheap in per-iteration costs (compared to the Gradient Method), which makes it a powerful tool for such a problems and verifies our theoretical findings.

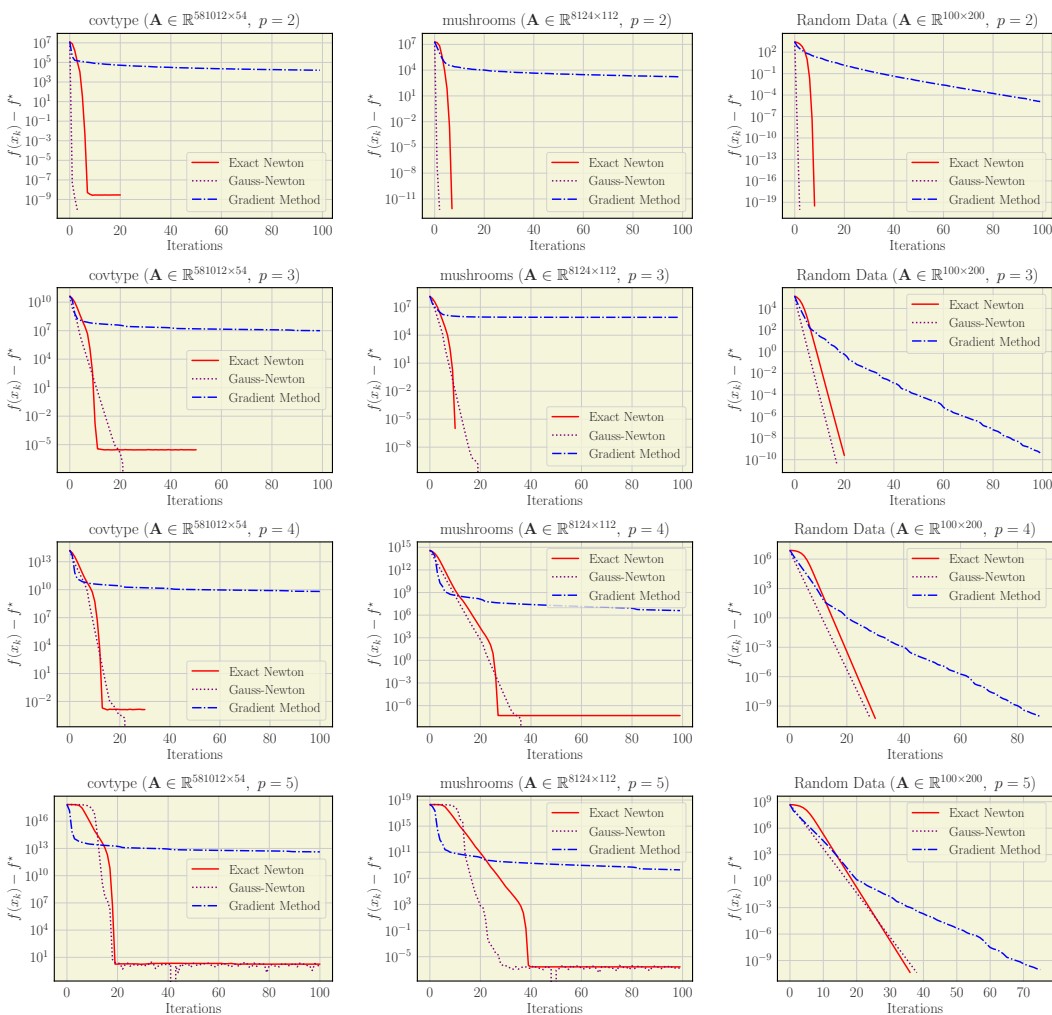

**Figure 7: Objective with Linear Operator.** In this setting, our method is identical when using either the exact or inexact Hessian given by (20). Notably, the Gauss-Newton preconditioning enables the Gradient Method to achieve performance comparable to the Newton Method. In contrast, the Gradient Method with our adaptive search exhibits significantly slower convergence for large values of $p$.

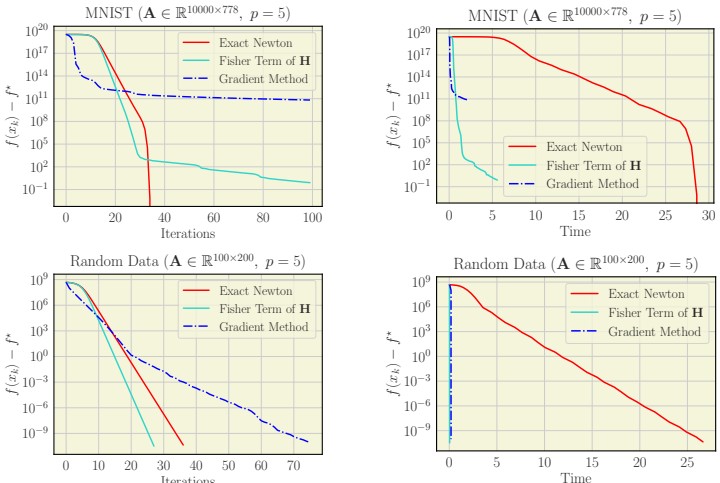

**Figure 8: Objective with Linear Operator. Our method with the inexact Hessian of a Fisher-type form (21) performs comparably to the Exact Newton Method.** In this experiment, we compare method (21) with the Exact Newton Method with the Hessian of (20) and the Gradient Method. We utilize objective from Example 7 with large values of $p = 2$, which complicates the problem. As our theory suggests, one can consider instead of the full Hessian (20) only its rank-one Fisher-type term $\frac{p-2}{\|\mathbf{u}(\mathbf{x})\|^p} \nabla f(\mathbf{x}) \nabla f(\mathbf{x})^\top$. If additionally we set $\mathbf{B} := \mathbf{A}^\top \mathbf{A}$ in Algorithm 1 with the approximation above, then we get a matrix that is can be inverted fast with by the Woodbury-Sherman-Morrison formula. Moreover, it appears that such a method performs relatively similar to the Exact Newton. Indeed, in all cases with large $p$, both exact and inexact algorithm significantly outperform the Gradient Method, as well as for the moderate powers $p$ in Figure 9. At the same time, Algorithm 1 with the Fisher-type approximation works much faster than the Exact Newton in the wall-clock time comparisons. Interestingly, that we also can observe how the difference in the wall-clock time performance for the Inexact Newton and the Gradient Method increases with dimensionality of the problem (MNIST versus small synthetic dataset). We see that this difference diminishes when the problem is small-dimensional, since the inversion in (21) happens much faster.

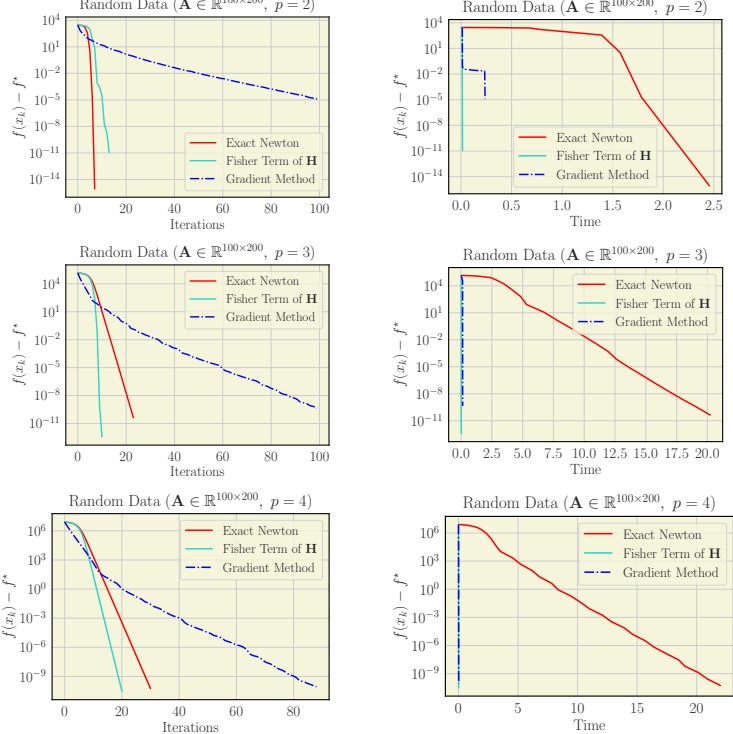

**Figure 9: Objective with Linear Operator. Our method with the inexact Hessian of a Fisher-type form (21) performs comparably to the Exact Newton Method.** In this experiment, we show that problem in Example 7 with linear operator $\mathbf{u}(\mathbf{x}) = \mathbf{A}\mathbf{x} - \mathbf{b}$ can also be considered via the inexact Hessians perspective. Indeed, the update rule for the Exact Newton Method with (20), in this case, resembles a combination of the Gauss-Newton term and the rank-one Fisher update. However, if one would use the matrix $\mathbf{B} := \mathbf{A}^\top \mathbf{A}$ in Algorithm 1, then it is possible to get rid of the Gauss-Newton term and use only rank-one correction as in (21). Thus, the most computationally expensive part in the algorithm is inversion of $\mathbf{A}^\top \mathbf{A}$. But, with the Woodbury-Sherman-Morrison formula this inversion becomes a particularly cheap operation and can be done relatively fast. As we see, our algorithm with the inexact Hessian not only remains the convergence of the Exact Newton in this case, but also works almost as fast as the Gradient Method in the wall-clock time comparison.

### B.4 NON-CONVEX OBJECTIVES

Let us also delve into numerical experiments on non-convex functions. In this part, we consider a simple yet widespread problem of the optimization of the Rosenbrock function (Rosenbrock, 1960). And the the Nonlinear Equations problem with the operator being from a family of Chebyshev polynomials. The latter example is particularly novel for the experiments and, to the best of our knowledge, has been investigated in practice in the non-smooth case in (Kabgani & Ahookhosh, 2025; Gürbüzbalaban & Overton, 2012) (a so-called Chebyshev oscillator problem), and in the smooth case in (Cartis et al., 2013). We extend all prior experimental results on these objectives to the Nonlinear Equations problem with different powers $p$ and usage of the inexact Hessian.

**Two-Dimensional Rosenbrock Function.** We utilize a non-convex smooth objective of the following form

$$f(\boldsymbol{x}) \;=\; (1 - x_1)^2 + 100 \left(x_2 - x_1^2\right)^2, \quad \text{where } \boldsymbol{x} := (x_1, \; x_2)^\top \in \mathbb{R}^2. \tag{22}$$

Note that (22) can be seen as a smooth variant of the Nesterov-Chebyshev-Rosenbrock function (Gürbüzbalaban & Overton, 2012). In scientific computing, this function is used as a benchmarking problem for optimization algorithms. It has a unique global minimizer $(1, 1)$, where $f^\star = 0$. This global minimum is inside a parabolic-shaped valley (Figure 12) that is easy to find, but, for the Gradient Method, it takes thousands of iterations to approach the vicinity of the solution (Figure 10).

**Nonlinear Equations with the Rosenbrock Function.** To follow our theoretical justifications, we not only investigate the convergence of Algorithm 1 on the plain Rosenbrock function, but also introduce a new objective that relates to our previous finding and to Example 7. We formalize it as

$$f(\boldsymbol{x}) \;=\; \tfrac{1}{p}\|\boldsymbol{u}(\boldsymbol{x})\|^p, \quad \text{where } \boldsymbol{u}(\boldsymbol{x}) := \left(1 - x_1, \; 10(x_2 - x_1^2)\right)^\top. \tag{23}$$

We call such an operator $\boldsymbol{u}(\boldsymbol{x})$ — vector of the Rosenbrock residuals and refer to the problem of minimizing (23) as **Nonlinear Equations & Rosenbrock**. For the case $p = 2$, the objective (23) resembles (22) up to a constant factor $\tfrac{1}{2}$, thus both problems have the same optimum and are similar (see Figure 10). However, reformulation (23) allows us to introduce the notion of Hessian inexactness that we cover in our theoretical analysis. Thus, using our approximation from Example 7, we can approach problem (23) with Algorithm 1 using an inexact Hessian. As theory suggests, our method should achieve the same convergence rate as the full Newton, which we observe in Figure 10 (b) and in Figure 11 when varying the power $p$, making the problem harder for the Gradient Method.

Furthermore, we see that the Gradient Method and our algorithm with exact and inexact Hessian follow the same optimization direction through this narrow parabolic-shaped valley of the Rosenbrock function — Figure 12. However, our method accelerates much when finds a sweet spot in this valley.

As an advantage of using the Hessian approximation in this setup, we pose the fact that the Newton Method with an exact Hessian fails to converge given some inappropriate starting point which can actually be close to the optimum – Figure 13. Which happens due to the inability to invert the regularized Hessian of the objective at the beginning of the run. At the same time, Algorithm 1 with an inexact Hessian succeeds for any starting points we have tried.

**Inexact Hessian and the Chebyshev Polynomials.** We illustrate our theoretical finding on a new, particularly interesting problem — Nonlinear Equations with Chebyshev polynomials. We formulate our objective as follows

$$f(\boldsymbol{x}) \;=\; \tfrac{1}{p}\|\boldsymbol{u}(\boldsymbol{x})\|^p; \quad \text{where } \boldsymbol{u}(\boldsymbol{x}) = (u_1(\boldsymbol{x}), \ldots, u_d(\boldsymbol{x}))^\top, \quad \text{such that}$$
$$u_1(\boldsymbol{x}) = \tfrac{1}{2}(1 - x_1), \quad u_i(\boldsymbol{x}) = x_i - \mathrm{p}_2(x_{i-1}), \quad \mathrm{p}_2(\tau) = 2\tau^2 - 1. \tag{24}$$

Where $\mathrm{p}_2$ is the Chebyshev polynomial of degree two. Clearly, for the case $p = 2$, our objective (24) resembles the smooth Nesterov-Chebyshev-Rosenbrock function studied in (Jarre, 2011; Cartis et al., 2013) up to a constant factor $\tfrac{1}{2}$. Indeed,

$$\|\boldsymbol{u}(\boldsymbol{x})\|^2 \;=\; \tfrac{1}{4}(1 - x_1)^2 + \sum_{i=1}^{d-1}\left(x_{i+1} - 2x_i^2 + 1\right)^2.$$

As for the plain Rosenbrock function (22) and our adaptation of it to the Nonlinear Equations problem (23), the only stationary point $(1, \ldots, 1)$ of the Nesterov-Chebyshev-Rosenbrock objective is the

global minimizer. Although this function is very difficult for numerical methods in both its smooth (Jarre, 2011) and non-smooth (Gürbüzbalaban & Overton, 2012) variants.

In our experiments, we extend the Nesterov-Chebyshev-Rosenbrock function to (24). Thus, we are able to use the approximation of Example 7 in our method. We demonstrate the convergence of the full Newton, Algorithm 1 with the inexact Hessian and the Gradient Method. And show how it depends on the effects that come from the increase in the dimensionality of the vector function $\boldsymbol{u}(\cdot)$ and the increase of power $p$ in our objective (24). We notice that our method with approximation performs remarkably well in all settings we have tried, as depicted in Figures 14 and 15. In particular, both exact and inexact variants of Algorithm 1 perform significantly better than the Gradient Method when $p$ is small enough (Figure 14), but when increasing $p$ (Figure 15), we observe that the Gradient Method also starts to perform better.

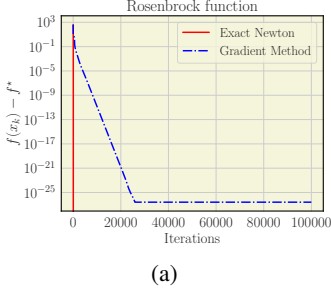
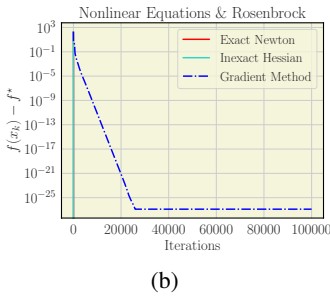

(a)  (b)

**Figure 10: Optimization of the Rosenbrock function (a), and the Nonlinear Equation problem for $\mathbf{p} = 2$ with Rosenbrock residuals (b), looks quite similar.** For the second problem we can use the Hessian approximation suggested by our theory (see Example 7). With such an approximation our method is in the regime where it has the same convergence as the Newton Method with the full Hessian. However, by using an inexact Hessian, we obtain a more numerically stable algorithm with respect to the choice of the starting iterate, as depicted in Figure 13.

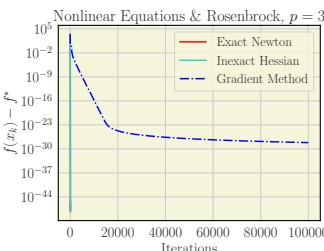
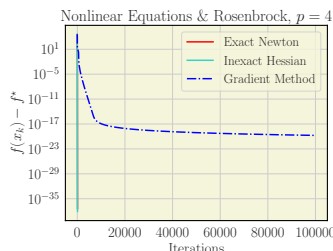
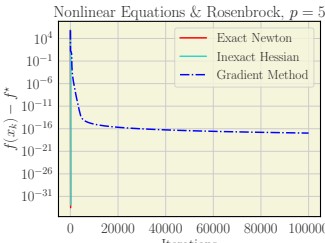

**Figure 11: Varying the power $\mathbf{p}$ in the Nonlinear Equations problem (23)** we complicate the convergence for the Gradient Method. However, our method with an approximation and the Newton Method works quite similarly even for different values of $p$. Here, "Inexact Hessian" stands for the approximation suggested by our theory in Example 7.

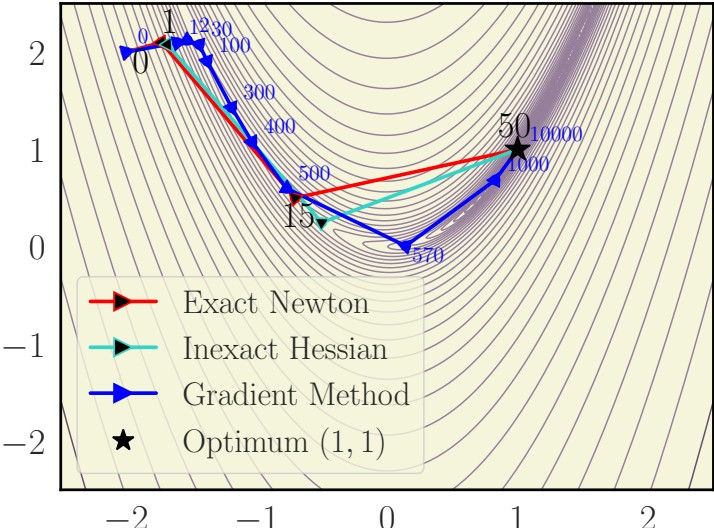

**Figure 12: Contour plot of the Nonlinear Equations problem with the Rosenbrock residuals in the two-dimensional case.** In a similar way that the authors of (Kabgani & Ahookhosh, 2025; Gürbüzbalaban & Overton, 2012) do for the problems they consider, we plot the level contours for the objective $f(\boldsymbol{x}) = \frac{1}{2}\|\boldsymbol{u}(\boldsymbol{x})\|^2$ with operator $\boldsymbol{u}(\boldsymbol{x}) := \begin{pmatrix} 1 - x_1, & 10(x_2 - x_1^2) \end{pmatrix}^\top$ being a vector-function of two Rosenbrock residuals as in Equation (23). The contour we obtain for this objective also similar to that of the *non-smooth* variant of the Nesterov-Chebyshev-Rosenbrock function from (Gürbüzbalaban & Overton, 2012). However, our objective (23) remains a *non-convex smooth* function, thus, serves as a good example complementing our theory. Points connected by line segments show the iterates generated by Algorithm 1 with exact Hessian, Algorithm 1 with approximation described in Example 7, and the Gradient Method. The markers for Exact and Inexact methods are of the same color because the optimization trajectory of both algorithms is quite similar and their consecutive iterates lie relatively close to each other. For all methods we utilize our adaptive search procedure (see Equation (16) and Appendix D). The comparison run of exactly those methods in terms of the functional residual is depicted in Figure 10 (b) and the starting point is $(-2, 2)$. In our contour plot, we see that all three methods, when initialized outside the parabolic valley of the objective, firstly tend to find this valley as soon as possible, and then follow down to the global minimizer $(1, 1)$. However, both Exact and Inexact methods move significantly faster once they found the valley. Indeed, we see that the first iterate returned by the Gradient Method is closer to the valley, but then, 15-th iterate of Algorithm 1 variations is ahead of 500-th iterate of the Gradient Method.

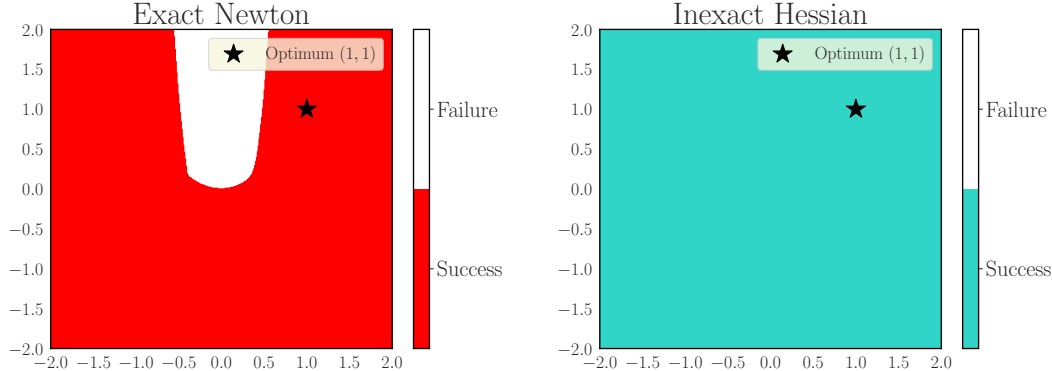

**Figure 13: Region of Starting Points Where Method Fails to Converge.** In this experiment, we consider Algorithm 1 with an exact Hessian and the same method with approximation described in Example 7. We employ our adaptive search procedure (see Equation (16) and Appendix D), and run both methods on the Nonlinear Equations problems with the Rosenbrock residuals in the two-dimensional case. We chose the starting point $\boldsymbol{x}_0$ from a grid in the range $(-2, 2)$ for both its coordinates. Interestingly, the method with the full Hessian fails to converge given certain starting points that are relatively close to the global minimizer. This happens due to the ill-conditioning issues with the exact Hessian matrix during the inversion. Fixing those issues with another choice of regularization or update rule means changing the algorithm, therefore we did not perform those changes. However, our algorithm with inexact Hessian works remarkably well without for any staring iterate given from the range we considered. Therefore, our experimental validations suggests that Algorithm 1 with inexact Hessian not only performs similarly to the full Newton if the approximation is in accordance with our theory, but also serves as more numerically stable method.

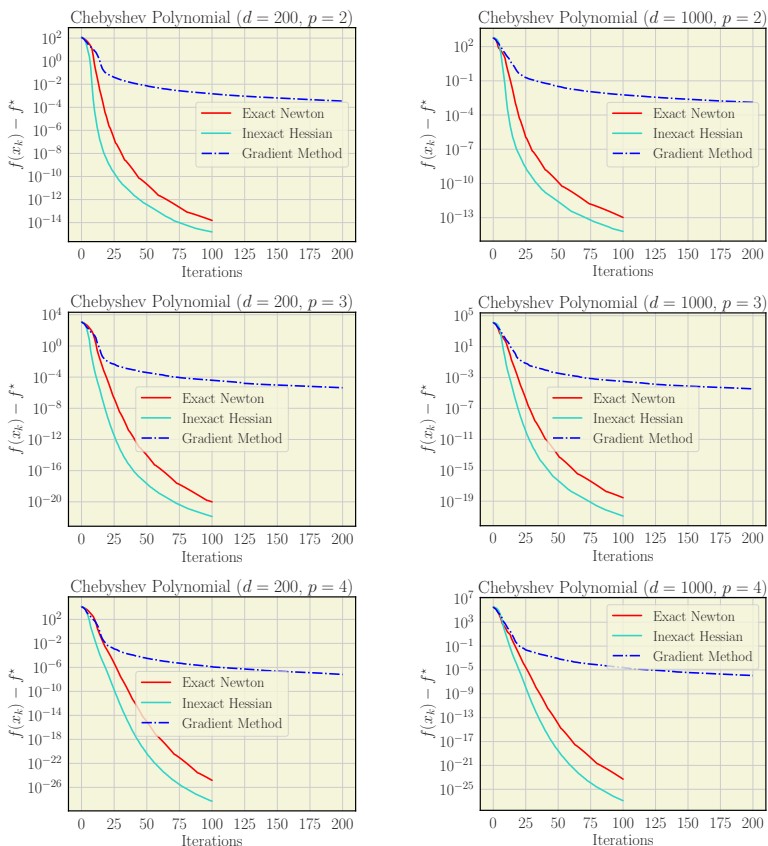

**Figure 14: Algorithm 1 with the inexact Hessian noticeably outperforms both the full Newton and the Gradient Method on the Nonlinear Equations problem with the Chebyshev polynomial objective.** For this experiment, we utilize the objective (24), where $d$ stands for the dimension of the vector-function $\boldsymbol{u}(\cdot)$. Importantly, $\|\boldsymbol{u}(\boldsymbol{x})\|^2$ correspond to the smooth variant of the Nesterov-Chebyshev-Rosenbrock function which is known as a hard problem for numerical methods. Throughout this experiment, we vary both $p$ and $d$ to complicate the optimization of our objective. Noticeably, in all these cases Algorithm 1 with approximation outperforms the full Newton.

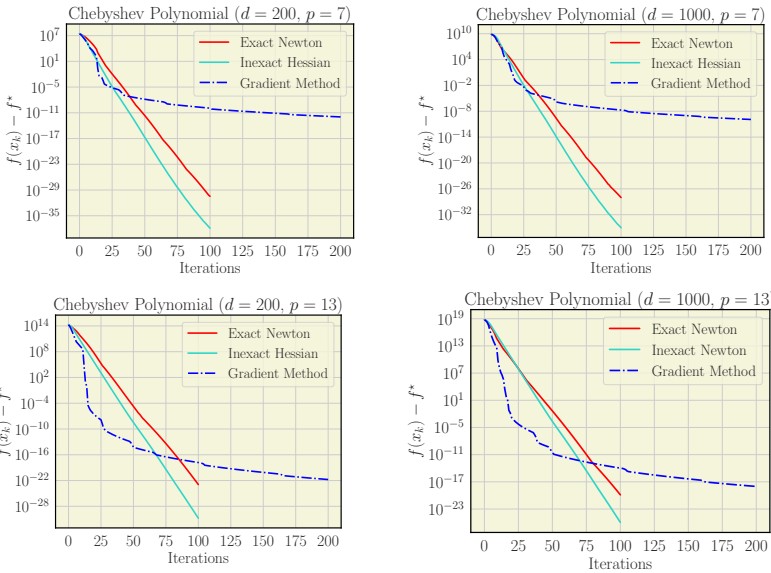

**Figure 15: When increasing the power p, we see that the gap between Algorithm 1 and the Gradient Method narrows.** We run three methods on objective (24) and, in the same way as in Figure 14, we increase both $p$ and the dimension of $\boldsymbol{u}(\cdot)$. Clearly, we observe a dynamics that the gain of Algorithm 1 with and without approximation becomes less significant when the $p$ is large.

## B.5 COMPARISON WITH ADAPTIVE AND UNIVERSAL METHODS

In this section, we elaborate more on comparisons of our framework with other adaptive and universal methods. We experimentally study the Cubic Newton method (Nesterov & Polyak, 2006) and two more algorithms, namely—Affine-Invariant Cubic Newton AICN (Hanzely et al., 2022) and Gradient-Regulated Line Search (GRLS) (Hanzely et al., 2024). Both AICN and GRLS are instances of the Damped Newton method, but with different adaptive rules for the step-size selection. While for the Exact Newton method with gradient regularization and its version with the inexact Hessian, we use the adaptive search for parameter $\gamma(\cdot)$, which controls the regularization term in `line 3` of Algorithm 1. We also compare their performance with the Fast Gradient Method (Nesterov, 2018) to observe the advantage from using (an approximate) second-order information. As problems for this comparison visualized in Figure 16, we choose: LogSumExp **(a)**, Nonlinear Equations **(b)** on the a9a dataset, and Chebyshev polynomials **(c)**. Their descriptions match those from prior sections. Subsequently, we employ the Weighted Gauss-Newton approximation (Eq. 19) for LogSumExp, Fisher Term of $\mathbf{H}$ (Eq. 21) for Nonlinear Equations, and the approximation from Example 7 for Chebyshev polynomials.

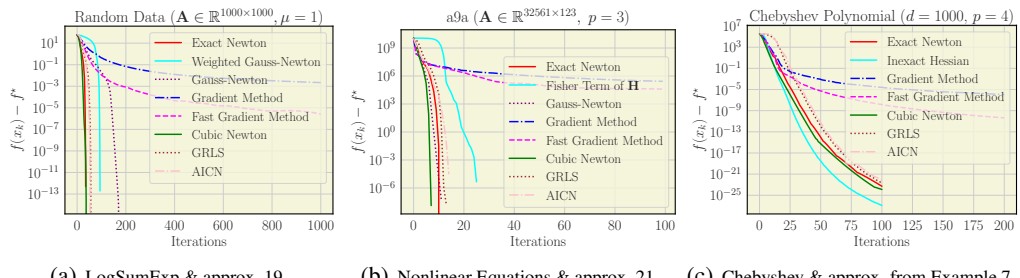

(a) LogSumExp & approx. 19.   (b) Nonlinear Equations & approx. 21.   (c) Chebyshev & approx. from Example 7.

**Figure 16: Comparison with Cubic Newton and versions of the Damped Newton method with adaptive step-size**. In all figures, we demonstrate a clear improvement of second-order methods and Algorithm 1 with inexact Hessians and adaptive search over first-order methods on both convex (**(a,b)**) and non-convex **(c)** objectives, for important problems, corresponding to Examples 8 and 7. Runs in **(a)**, follow the recipe from Appendix B.2, where we used the Weighted Gauss-Newton approximation (Eq. 19), theoretical bounds for this kind of approximation are proven in Example 11. The only difference with a setup from the prior section, is that we study more methods here: Cubic Newton, accelerated gradient method, and two versions of the Damped Newton with adaptive step-sizes—GRLS and AICN. In **(b)**, we replicate our experiments from Appendix B.3, utilizing the Fisher Term of $\mathbf{H}$ approximation (Eq. 21). The main difference here, except for the methods, is a new a9a dataset that is not presented in Figures 7, 8, 9. See proofs for the bounds on the approximation in Example 12. Finally, in **(c)**, we show results for the non-convex smooth Chebyshev polynomials problem, studied in Appendix B.4. When the operator is nonlinear, $\mathbf{H}(\boldsymbol{x})$ from Example 7 does not resemble the full Hessian and we can explicitly use this equation. For this case, extended bounds are also proven in Example 12.

## C    COMPOSITE OPTIMIZATION PROBLEMS

Let us consider a more general formulation of the Composite Optimization Problem (Nesterov, 2018):

$$\min_{\boldsymbol{x} \in Q} \Big\{ F(\boldsymbol{x}) \quad := \quad f(\boldsymbol{x}) + \psi(\boldsymbol{x}) \Big\}, \tag{25}$$

where $f : Q \to \mathbb{R}$ is a differentiable function, which can be non-convex, and $\psi : Q \to \mathbb{R}$ is a *simple* closed convex function with $Q := \operatorname{dom} \psi$. This setup covers optimization problems with simple constraints, in which case $\psi$ is $\{0, +\infty\}$-indicator of a given closed convex set $Q \subset \mathbb{R}^n$.

We denote $F^\star := \inf_{\boldsymbol{x} \in Q} F(\boldsymbol{x}) > -\infty$ which we assume to be bounded.

### C.1    COMPOSITE NEWTON STEP WITH HESSIAN APPROXIMATION

In case of the presence of the composite component $\psi$, we have to modify our method accordingly. Now, begin at point $\boldsymbol{x} \in Q$ and for a certain vector $F'(\boldsymbol{x}) := \nabla f(\boldsymbol{x}) + \psi'(\boldsymbol{x})$, where $\psi'(\boldsymbol{x}) \in \partial \psi(\boldsymbol{x})$, we compute the next iterate $\boldsymbol{x}^+$ as the solution to the following subproblem:

$$\boldsymbol{x}^+ = \arg\min_{\boldsymbol{y} \in Q} \Big\{ \langle \nabla f(\boldsymbol{x}), \boldsymbol{y} - \boldsymbol{x} \rangle + \tfrac{1}{2} \langle \mathbf{H}(\boldsymbol{x})(\boldsymbol{y} - \boldsymbol{x}), \boldsymbol{y} - \boldsymbol{x} \rangle + \tfrac{\|F'(\boldsymbol{x})\|_*}{2\gamma} \|\boldsymbol{y} - \boldsymbol{x}\|^2 + \psi(\boldsymbol{y}) \Big\}, \tag{26}$$

where $\mathbf{H}(\boldsymbol{x}) = \mathbf{H}(\boldsymbol{x})^\top \succeq \mathbf{0}$ is a positive semidefinite approximation of the Hessian of the smooth part, and $\gamma > 0$ is our step-size parameter. Note that the subproblem in (26) is strongly convex, and in case $\psi \equiv 0$ it corresponds to one iteration of Algorithm 1.

In general, the solution to (26) satisfies the following optimality condition (Nesterov, 2018):

$$\Big\langle F'(\boldsymbol{x}) + \Big( \mathbf{H}(\boldsymbol{x}) + \tfrac{\|F'(\boldsymbol{x})\|_*}{\gamma} \mathbf{B} \Big)(\boldsymbol{x}^+ - \boldsymbol{x}), \boldsymbol{x}^+ - \boldsymbol{x} \Big\rangle + \psi(\boldsymbol{y}) \quad \geq \quad \psi(\boldsymbol{x}^+), \qquad \forall \boldsymbol{y} \in Q, \tag{27}$$

or, in other words, the vector

$$\psi'(\boldsymbol{x}^+) \quad := \quad -\nabla f(\boldsymbol{x}) - \mathbf{H}(\boldsymbol{x})(\boldsymbol{x}^+ - \boldsymbol{x}) - \tfrac{\|F'(\boldsymbol{x})\|_*}{\gamma} \mathbf{B}(\boldsymbol{x}^+ - \boldsymbol{x}) \tag{28}$$

belongs to the subdifferential of $\psi$ at new point: $\psi'(\boldsymbol{x}^+) \in \partial \psi(\boldsymbol{x}^+)$.

Let us derive useful inequalities for one step of the composite method. Note that for any stationary point $\boldsymbol{x}^\star$ to problem (25), setting $\boldsymbol{x} := \boldsymbol{x}^\star$ we have $\boldsymbol{x}^+ = \boldsymbol{x}^\star$, as it satisfies the optimality condition (27). Therefore, without loss of generality we can always assume that $\boldsymbol{x} \neq \boldsymbol{x}^\star$.

> **Lemma 2.** *Let $\psi'(\boldsymbol{x}) \in \partial \psi(\boldsymbol{x})$ be an arbitrary subgradient and denote $F'(\boldsymbol{x}) := \nabla f(\boldsymbol{x}) + \psi'(\boldsymbol{x}) \neq \mathbf{0}$. Then, for any $\gamma > 0$, it holds*
>
> $$\langle F'(\boldsymbol{x}), \boldsymbol{x} - \boldsymbol{x}^+ \rangle \quad > \quad 0, \tag{29}$$
>
> $$\|\boldsymbol{x}^+ - \boldsymbol{x}\| \quad \leq \quad \gamma, \tag{30}$$
>
> *and*
>
> $$\|\boldsymbol{x}^+ - \boldsymbol{x}\|_{\boldsymbol{x}}^2 \quad := \quad \langle \nabla^2 f(\boldsymbol{x})(\boldsymbol{x}^+ - \boldsymbol{x}), \boldsymbol{x}^+ - \boldsymbol{x} \rangle \quad \leq \quad \langle F'(\boldsymbol{x}), \boldsymbol{x} - \boldsymbol{x}^+ \rangle$$
> $$+ \|\boldsymbol{x}^+ - \boldsymbol{x}\| \cdot \Big( \|(\nabla^2 f(\boldsymbol{x}) - \mathbf{H}(\boldsymbol{x}))(\boldsymbol{x}^+ - \boldsymbol{x})\|_* - \tfrac{\|F'(\boldsymbol{x})\|_* \|\boldsymbol{x}^+ - \boldsymbol{x}\|}{\gamma} \Big). \tag{31}$$

*Proof.* Indeed, multiplying (28) by $\boldsymbol{x}^+ - \boldsymbol{x}$ and using convexity of $\psi$, we have

$$\langle \mathbf{H}(\boldsymbol{x})(\boldsymbol{x}^+ - \boldsymbol{x}), \boldsymbol{x}^+ - \boldsymbol{x} \rangle + \tfrac{\|F'(\boldsymbol{x})\|_*}{\gamma} \|\boldsymbol{x}^+ - \boldsymbol{x}\|^2 \quad = \quad \langle \nabla f(\boldsymbol{x}) + \psi'(\boldsymbol{x}^+), \boldsymbol{x} - \boldsymbol{x}^+ \rangle$$

$$\leq \quad \langle F'(\boldsymbol{x}), \boldsymbol{x} - \boldsymbol{x}^+ \rangle.$$

Therefore, taking into account that $\mathbf{H}(\boldsymbol{x}) \succeq \mathbf{0}$, we conclude that

$$0 \quad < \quad \tfrac{\|F'(\boldsymbol{x})\|_*}{\gamma} \|\boldsymbol{x}^+ - \boldsymbol{x}\|^2 \leq \langle F'(\boldsymbol{x}), \boldsymbol{x} - \boldsymbol{x}^+ \rangle,$$

which proves (29). Applying Cauchy-Schwartz inequality also gives (30). Now, to establish (31), we notice that

$$\langle \nabla^2 f(\boldsymbol{x})(\boldsymbol{x}^+ - \boldsymbol{x}), \boldsymbol{x}^+ - \boldsymbol{x}\rangle$$

$$= \langle \mathbf{H}(\boldsymbol{x})(\boldsymbol{x}^+ - \boldsymbol{x}), \boldsymbol{x}^+ - \boldsymbol{x}\rangle + \langle (\nabla^2 f(\boldsymbol{x}) - \mathbf{H}(\boldsymbol{x}))(\boldsymbol{x}^+ - \boldsymbol{x}), \boldsymbol{x}^+ - \boldsymbol{x}\rangle$$

$$\leq \langle F'(\boldsymbol{x}), \boldsymbol{x} - \boldsymbol{x}^+\rangle - \frac{\|F'(\boldsymbol{x})\|_*}{\gamma}\|\boldsymbol{x}^+ - \boldsymbol{x}\|^2 + \|(\nabla^2 f(\boldsymbol{x}) - \mathbf{H}(\boldsymbol{x}))(\boldsymbol{x}^+ - \boldsymbol{x})\|_*\|\boldsymbol{x}^+ - \boldsymbol{x}\|,$$

which completes the proof. $\qquad\square$

We see that according to our definition (26), we ensure that every step remains bounded (30) by our parameter $\gamma > 0$. Let us recall our Definition 1 of the Gradient-Normalized Smoothness from the main part, for any $\boldsymbol{x} \in Q$ and $\boldsymbol{g} \in \mathbb{R}^n$:

$$\gamma(\boldsymbol{x}, \boldsymbol{g}) := \max\Big\{\gamma \geq 0 \;:\; \|\nabla f(\boldsymbol{x} + \boldsymbol{h}) - \nabla f(\boldsymbol{x}) - \mathbf{H}(\boldsymbol{x})\boldsymbol{h}\|_* \leq \frac{\|\boldsymbol{g}\|_*\|\boldsymbol{h}\|}{\gamma}, \; \forall \boldsymbol{h} \in B_\gamma \cap \mathcal{O}_{\boldsymbol{x},\boldsymbol{g}}\Big\},$$

where $B_\gamma := \{\boldsymbol{h} : \|\boldsymbol{h}\| \leq \gamma\}$ is the Euclidean ball, and $\mathcal{O}_{\boldsymbol{x},\boldsymbol{g}} := \{\|\boldsymbol{h}\|_{\boldsymbol{x}}^2 + \langle \boldsymbol{g}, \boldsymbol{h}\rangle \leq 0\}$ is the local region. Note that this definition measures the local level of smoothness for our differentiable part $f$, and it does not take into account the composite component $\psi$. However, as we will see, in the composite case we change the direction $\boldsymbol{g}$ in our algorithm to define the step-size, by taking the *perturbed gradient*: $\boldsymbol{g} := \nabla f(\boldsymbol{x}) + \psi'(\boldsymbol{x})$.

First, let us derive simple consequences of the definition of $\gamma(\boldsymbol{x}, \boldsymbol{g})$.

**Lemma 3.** *Let $0 < \gamma \leq \gamma(\boldsymbol{x}, \boldsymbol{g})$. Then, for any $\boldsymbol{h} \in B_\gamma \cap \mathcal{O}_{\boldsymbol{x},\boldsymbol{g}}$, it holds*

$$|f(\boldsymbol{x} + \boldsymbol{h}) - f(\boldsymbol{x}) - \langle \nabla f(\boldsymbol{x}), \boldsymbol{h}\rangle - \tfrac{1}{2}\langle \mathbf{H}(\boldsymbol{x})\boldsymbol{h}, \boldsymbol{h}\rangle| \quad \leq \quad \frac{\|\boldsymbol{g}\|_*\|\boldsymbol{h}\|^2}{2\gamma}. \tag{32}$$

*Proof.* Indeed, we have

$$|f(\boldsymbol{x} + \boldsymbol{h}) - f(\boldsymbol{x}) - \langle \nabla f(\boldsymbol{x}), \boldsymbol{h}\rangle - \tfrac{1}{2}\langle \mathbf{H}(\boldsymbol{x})\boldsymbol{h}, \boldsymbol{h}\rangle|$$

$$= \left|\int_0^1 \langle \nabla f(\boldsymbol{x} + \tau\boldsymbol{h}) - \nabla f(\boldsymbol{x}) - \tau\mathbf{H}(\boldsymbol{x})\boldsymbol{h}, \boldsymbol{h}\rangle d\tau\right|$$

$$\leq \int_0^1 \|\nabla f(\boldsymbol{x} + \tau\boldsymbol{h}) - \nabla f(\boldsymbol{x}) - \tau\mathbf{H}(\boldsymbol{x})\boldsymbol{h}\|_* d\tau \cdot \|\boldsymbol{h}\|$$

$$\leq \frac{\|\boldsymbol{g}\|_*\|\boldsymbol{h}\|^2}{2\gamma},$$

where we used the definition of $\gamma(\boldsymbol{x}, \boldsymbol{g})$ in the last inequality. $\qquad\square$

**Lemma 4.** *Let $0 < \gamma \leq \gamma(\boldsymbol{x}, \boldsymbol{g})$. Then, for any $\boldsymbol{s} \in \mathbb{R}^n$ s.t. $\|\boldsymbol{s}\| = 1$ and $\langle \boldsymbol{g}, \boldsymbol{s}\rangle < 0$ we have*

$$\|(\nabla^2 f(\boldsymbol{x}) - \mathbf{H}(\boldsymbol{x}))\boldsymbol{s}\|_* \quad \leq \quad \frac{\|\boldsymbol{g}\|_*}{\gamma}. \tag{33}$$

*Proof.* Let us take $\boldsymbol{h} := \tau\boldsymbol{s}$, where $0 < \tau \leq \gamma \leq \gamma(\boldsymbol{x}, \boldsymbol{g})$. Clearly, $\boldsymbol{h} \in B_\gamma$. Moreover,

$$\|\boldsymbol{h}\|_{\boldsymbol{x}}^2 + \langle \boldsymbol{g}, \boldsymbol{h}\rangle \quad = \quad \tau\Big(\tau\langle \nabla^2 f(\boldsymbol{x})\boldsymbol{s}, \boldsymbol{s}\rangle + \langle \boldsymbol{g}, \boldsymbol{s}\rangle\Big) \quad \leq \quad 0,$$

for sufficiently small $\tau > 0$. Hence, for sufficiently small $\tau$, we have:

$$\left\|\tfrac{1}{\tau}(\nabla f(\boldsymbol{x} + \tau\boldsymbol{s}) - \nabla f(\boldsymbol{x})) - \mathbf{H}(\boldsymbol{x})\boldsymbol{s}\right\|_* \quad \leq \quad \frac{\|\boldsymbol{g}\|_*}{\gamma}.$$

Taking the limit $\tau \to +0$ completes the proof. $\qquad\square$

Note that according to Lemma 2, normalized direction of our method, $s := \frac{x^+ - x}{\|x^+ - x\|}$ satisfies $\langle g, s \rangle < 0$ for $g := F'(x)$ (inequality 29). Therefore, we obtain the following direct result.

**Corollary 4.** *Let $0 < \gamma \le \gamma(x, F'(x))$. Then, one composite step (26) satisfies*

$$\|x^+ - x\|_x^2 \le \langle F'(x), x - x^+ \rangle. \tag{34}$$

*Thus,*

$$x^+ - x \in B_\gamma \cap \mathcal{O}_{x, F'(x)}. \tag{35}$$

Due to inclusion (35), we show the following one step progress for our method. Note that Lemma 1 is a simple direct consequence of this result, using $\psi \equiv 0$.

**Lemma 5.** *Let $0 < \gamma \le \gamma(x, F'(x))$. Then,*

$$F(x) - F(x^+) \ge \frac{\gamma}{8} \frac{\|F'(x^+)\|_*^2}{\|F'(x)\|_*} \tag{36}$$

*Proof.* Substituting the optimality condition (27) into global bound on the function progress (32), we get

$$f(x^+) \le f(x) - \tfrac{1}{2}\langle \mathbf{H}(x)(x^+ - x), x^+ - x \rangle - \frac{\|F'(x)\|_* \|x^+ - x\|^2}{2\gamma} + \langle \psi'(x^+), x - x^+ \rangle$$

$$\le F(x) - \tfrac{1}{2}\langle \mathbf{H}(x)(x^+ - x), x^+ - x \rangle - \frac{\|F'(x)\|_* \|x^+ - x\|^2}{2\gamma} - \psi(x^+),$$

which gives

$$F(x) - F(x^+) \ge \frac{\|F'(x)\|_* \|x^+ - x\|^2}{2\gamma}. \tag{37}$$

At the same time,

$$\left\|F'(x^+) + \frac{\|F'(x)\|_*}{\gamma}\mathbf{B}(x^+ - x)\right\|_* = \|\nabla f(x^+) - \nabla f(x) - \mathbf{H}(x^+ - x)\|_*$$

$$\le \frac{\|F'(x)\|_* \|x^+ - x\|}{\gamma},$$

where we used the definition of $\gamma(x, F'(x)) \ge \gamma$ in the last inequality. Hence, applying triangle inequality, we obtain:

$$\|F'(x^+)\|_* \le \frac{2\|F'(x)\|_* \|x^+ - x\|}{\gamma}.$$

Combining this inequality with (37) gives the required bound. □

### C.2 THE ALGORITHM FOR COMPOSITE OPTIMIZATION

We are ready to formalize our method for the general composite case, as follows.

---

**Algorithm 2** Composite Gradient-Regularized Newton with Approximate Hessians

**Initialization:** $x_0 \in Q$ and $\psi'(x_0) \in \partial\psi(x_0)$. Set $F'(x_0) \leftarrow \nabla f(x_0) + \psi'(x_0)$.
1: **for** $k \ge 0$ **do**
2:  Choose $\mathbf{H}(x_k) \succeq 0$ and $\gamma_k > 0$.
3:  Compute

$$x_{k+1} \leftarrow \arg\min_{y \in Q} \left\{ \langle \nabla f(x_k), y - x_k \rangle + \tfrac{1}{2}\langle \mathbf{H}(x_k)(y - x_k), y - x_k \rangle \right.$$

$$\left. + \frac{\|F'(x_k)\|_*}{2\gamma_k}\|y - x_k\|^2 + \psi(y) \right\}.$$

4:  Set $\psi'(x_{k+1}) \leftarrow -\nabla f(x_k) - \mathbf{H}(x_k)(x_{k+1} - x_k) - \frac{\|F'(x_k)\|_*}{\gamma_k}\mathbf{B}(x_{k+1} - x_k)$.
5:  Set $F'(x_{k+1}) \leftarrow \nabla f(x_{k+1}) + \psi'(x_{k+1})$.
6: **end for**

---

In the case $\psi \equiv 0$, this method is the same as Algorithm 1.

# D    THE CHOICE OF THE REGULARIZATION PARAMETER

Note that the only parameter of Algorithm 2 is a (second-order) step-size $\gamma_k > 0$ that describes the radius of the ball where the iteration belongs to: $\|\boldsymbol{x}_{k+1} - \boldsymbol{x}_k\| \leq \gamma_k$, similarly to trust-region approach (Conn et al., 2000).

Our theory suggests that the right choice of this parameter is provided by the Gradient-Normalized Smoothness (Definition 1), that is, in the general composite case:

$$\boxed{\gamma_k \quad := \quad \gamma(\boldsymbol{x}_k, F'(\boldsymbol{x}_k)).}$$

Then, according to Lemma 5, we ensure the following progress of each step:

$$F(\boldsymbol{x}_k) - F(\boldsymbol{x}_{k+1}) \quad \geq \quad \frac{\gamma_k}{8} \frac{\|F'(\boldsymbol{x}_{k+1})\|_*^2}{\|F'(\boldsymbol{x}_k)\|_*}. \tag{38}$$

Therefore, $\gamma_k \approx \gamma(\boldsymbol{x}_k)$ is *the best value* of a step-size that we can use. However, in practice, it can be difficult to compute the exact value of the Gradient-Normalized Smoothness. In this section, we show two different strategies that can work for a practical implementation of our method.

The first choice (Section D.1) is the **constant rule** for selecting $\gamma_k$:

$$\gamma_k \quad \equiv \quad \gamma_\star, \qquad \forall k \geq 0,$$

where $\gamma_\star > 0$ is a certain value, fixed once for *all iterations* of the method. Hence, since this is just one hyperparameter, one can perform a simple grid search for choosing $\gamma_\star$, in a similar spirit to step-size tuning for the stochastic gradient descent (SGD). However, we noticed in our experiments, that the value $\gamma_\star \approx 1$ is always a good guess for second-order methods, which also ensures local quadratic convergence of the method with exact Hessian, as for the classical Newton's method.

The second choice (Section D.2) that we will present is the use of **adaptive search** to estimate $\gamma_k$ at each iteration, which is a standard and cheap procedure, which also equips the method with fast global rates, without the need to know the exact value of $\gamma(\boldsymbol{x})$ or $\gamma_\star$.

In this section, for simplicity we focus on convex optimization problems, while our results can be generalized to other classes of problems. We consider non-convex optimization in Section E and the gradient-dominated objectives in Section F.

## D.1    THE CONSTANT RULE

Let us assume that the desired accuracy $\varepsilon > 0$ is fixed. This assumption is not very restrictive. Additionally, it allows to have a *stopping condition* for the method. Then, we denote the following set of function suboptimality:

$$\mathfrak{F}_\varepsilon \quad := \quad \Big\{ \boldsymbol{x} \in Q \ : \ F(\boldsymbol{x}) - F^\star \geq \varepsilon \Big\},$$

and, for a fixed initialization $\boldsymbol{x}_0 \in \mathfrak{F}_\varepsilon$, we denote the sublevel set by

$$\mathcal{F}_0 \quad := \quad \Big\{ \boldsymbol{x} \in Q \ : \ F(\boldsymbol{x}) \leq F(\boldsymbol{x}_0) \Big\},$$

which we assume to be bounded, i.e., $D := \mathrm{diam}(\mathcal{F}_0) < +\infty$. Note that by convexity, we obtain, for any subgradient $F'(\boldsymbol{x}) \in \partial F(\boldsymbol{x})$, with $\boldsymbol{x} \in \mathfrak{F}_\varepsilon \cap \mathcal{F}_0$:

$$\|F'(\boldsymbol{x})\|_* \quad \geq \quad \frac{F(\boldsymbol{x}) - F^\star}{D} \quad \geq \quad \delta \quad := \quad \frac{\varepsilon}{D}. \tag{39}$$

We are ready to formulate our main result, for the constant selection of $\gamma_k$ in our algorithm.

> **Theorem 3.** *Let $\varepsilon > 0$ be fixed, and assume that there exists $\gamma_\star > 0$ satisfying*
>
> $$\gamma_\star \;\leq\; \inf\Big\{\gamma(\boldsymbol{x}, F'(\boldsymbol{x})) \;:\; \boldsymbol{x} \in \mathfrak{F}_\varepsilon \cap \mathcal{F}_0,\; F'(\boldsymbol{x}) \in \partial F(\boldsymbol{x})\Big\}. \tag{40}$$
>
> *Consider $K \geq 1$ iterations of Algorithm 2 with*
>
> $$\gamma_k \;\equiv\; \gamma_\star, \tag{41}$$
>
> *and assume that $\boldsymbol{x}_k \in \mathfrak{F}_\varepsilon$, for all $0 \leq k \leq K$. Then,*
>
> $$K \;\leq\; \tfrac{8D}{\gamma_\star} \ln \tfrac{F(\boldsymbol{x}_0) - F^\star}{\varepsilon} + 2\ln \tfrac{\|F'(\boldsymbol{x}_0)\|_\ast D}{\varepsilon} \tag{42}$$

*Proof.* Indeed, by Lemma 5, our constant choice of $\gamma_k$ ensure the following progress of every iteration, denoting $F_k := F(\boldsymbol{x}_k) - F^\star$ and $g_k := \|F'(\boldsymbol{x}_k)\|_\ast$:

$$F_k - F_{k+1} \;\geq\; \tfrac{\gamma_\star}{8} \tfrac{g_{k+1}^2}{g_k} \;=\; \tfrac{\gamma_\star}{8}\Big(\tfrac{g_{k+1}}{g_k}\Big)^2 g_k \;\overset{(39)}{\geq}\; \tfrac{\gamma_\star}{8D}\Big(\tfrac{g_{k+1}}{g_k}\Big)^2 F_k. \tag{43}$$

Then, using concavity of the logarithm, we have

$$\ln \tfrac{F_k}{F_{k+1}} \;\geq\; \tfrac{F_k - F_{k+1}}{F_k} \;\overset{(43)}{\geq}\; \tfrac{\gamma_\star}{8D}\Big(\tfrac{g_{k+1}}{g_k}\Big)^2. \tag{44}$$

Telescoping this bound for the first $K$ iterations of the method, and using the inequality between arithmetic and geometric means, we obtain

$$\ln \tfrac{F_0}{\varepsilon} \;\geq\; \ln \tfrac{F_0}{F_K} \;\overset{(43)}{\geq}\; \tfrac{\gamma_\star K}{8D} \cdot \tfrac{1}{K}\sum_{k=0}^{K-1}\Big(\tfrac{g_{i+1}}{g_i}\Big)^2 \;\geq\; \tfrac{\gamma_\star K}{8D}\cdot\Big(\tfrac{g_K}{g_0}\Big)^{\tfrac{2}{K}}$$

$$\overset{(39)}{\geq}\; \tfrac{\gamma_\star K}{8D}\cdot\Big(\tfrac{\varepsilon}{g_0 D}\Big)^{\tfrac{2}{K}} \;=\; \tfrac{\gamma_\star K}{8D}\cdot\exp\Big[\tfrac{2}{K}\ln\tfrac{\varepsilon}{g_0 D}\Big] \;\geq\; \tfrac{\gamma_\star K}{8D}\cdot\Big[1 + \tfrac{2}{K}\ln\tfrac{\varepsilon}{g_0 D}\Big].$$

Rearranging the terms proves the required bound. $\qquad\square$

Note that the result of Theorem 3 is very general, as it does not assume any particular smoothness conditions, except separation from zero of the Gradient-Normalized Smoothness $\gamma(\cdot)$ on the bounded set $\mathfrak{F}_\varepsilon \cap \mathcal{F}_0$: $\gamma_\star > 0$. Under this condition, we show that our method with a constant rule $\gamma_k \equiv \gamma_\star$ needs

$$K \;=\; \tilde{\mathcal{O}}\Big(\gamma_\star^{-1} D\Big) \tag{45}$$

iterations to solve the problem, up to logarithmic terms.

Despite the constant rule (41) seems too conservative, it appears that it recovers the correct rates in all particular cases. For example, for the functions with $L_2$-Lipschitz Hessian, we have

$$\gamma(\boldsymbol{x}, F'(\boldsymbol{x})) \;\geq\; \sqrt{\tfrac{2}{L_2}\|F'(\boldsymbol{x})\|_\ast} \;\overset{(39)}{\geq}\; \sqrt{\tfrac{2\varepsilon}{L_2 D}} \;\equiv\; \gamma_\star.$$

Substituting this value of $\gamma_\star$ into (45) gives the complexity of $\tilde{O}\Big(\sqrt{\tfrac{L_2 D^3}{\varepsilon}}\Big)$, which matches the complexity of the Cubic Newton Nesterov & Polyak (2006) on convex functions, up to a logarithmic factor.

In Section F, we develop a more refined analysis that covers convex functions as a particular case, and allows us to avoid a logarithmic factor in some particular cases.

## D.2 THE METHOD WITH ADAPTIVE SEARCH

In this section, we provide another practical choice for $\gamma_k$, which is to adaptively ensure inequality (38). We present this strategy in the following algorithmic form. This method needs a parameter $\delta > 0$, which is a desired accuracy in terms of the gradient norm. It is used for the stopping condition.

This is the same method as Algorithm 2, but with a specific adaptive procedure to choose parameter $\gamma_k > 0$. It is clear that the method is well defined, as for a sufficiently large $t_k \geq 0$ we can ensure

---

**Algorithm 3** Adaptive Method with Approximate Hessians

---

**Initialization:** $\boldsymbol{x}_0 \in Q$, $\psi'(\boldsymbol{x}_0) \in \partial\psi(\boldsymbol{x}_0)$, $\gamma_0 > 0$, and $\delta > 0$. Set $F'(\boldsymbol{x}_0) \leftarrow \nabla f(\boldsymbol{x}_0) + \psi'(\boldsymbol{x}_0)$.

1: **for** $k \geq 0$ **do**
2:     If $\|F'(\boldsymbol{x}_k)\|_* \leq \delta$ then **stop** and **return** $\boldsymbol{x}_k$.
3:     Choose $\mathbf{H}(\boldsymbol{x}_k) \succeq \mathbf{0}$.
4:     Find the smallest integer $t_k \geq 0$ such that for $\gamma := 2^{-t_k} \cdot \gamma_k$ and $\mathbf{T}(\gamma), \boldsymbol{g}(\gamma)$ computed as

$$\mathbf{T}(\gamma) \quad \leftarrow \quad \arg\min_{\boldsymbol{y} \in Q}\Big\{\langle\nabla f(\boldsymbol{x}_k), \boldsymbol{y} - \boldsymbol{x}_k\rangle + \tfrac{1}{2}\langle\mathbf{H}(\boldsymbol{x}_k)(\boldsymbol{y} - \boldsymbol{x}_k), \boldsymbol{y} - \boldsymbol{x}_k\rangle$$
$$+ \tfrac{\|F'(\boldsymbol{x}_k)\|_*}{2\gamma}\|\boldsymbol{y} - \boldsymbol{x}_k\|^2 + \psi(\boldsymbol{y})\Big\},$$

and

$$\boldsymbol{g}(\gamma) \quad \leftarrow \quad \nabla f(\mathbf{T}(\gamma)) - \nabla f(\boldsymbol{x}_k) - \mathbf{H}(\boldsymbol{x}_k)(\mathbf{T}(\gamma) - \boldsymbol{x}_k) - \tfrac{\|F'(\boldsymbol{x}_k)\|_*}{\gamma}\mathbf{B}(\mathbf{T}(\gamma) - \boldsymbol{x}_k)$$

it holds
$$F(\boldsymbol{x}_k) - F(\mathbf{T}(\gamma)) \quad \geq \quad \tfrac{\gamma}{8}\tfrac{\|\boldsymbol{g}(\gamma)\|_*^2}{\|F'(\boldsymbol{x}_k)\|_*} \quad \text{or} \quad \|\boldsymbol{g}(\gamma)\|_* \leq \delta.$$

5:     Set $\boldsymbol{x}_{k+1} \leftarrow \mathbf{T}(2^{-t_k} \cdot \gamma_k)$ and $F'(\boldsymbol{x}_{k+1}) \leftarrow \boldsymbol{g}(2^{-t_k} \cdot \gamma_k)$.
6:     Set $\gamma_{k+1} \leftarrow 2^{-t_k+1} \cdot \gamma_k$.
7: **end for**

---

that $2^{-t_k} \cdot \gamma_k \leq \gamma(\boldsymbol{x}_k, F'(\boldsymbol{x}_k))$ and therefore the condition of the adaptive search will be satisfied. At the same time, the total number $N_K$ of oracle calls during $K \geq 0$ iterations is bounded as

$$N_K \quad := \quad \sum_{k=0}^{K-1}(1 + t_k) = 2K + \sum_{k=0}^{K-1}\log_2\tfrac{\gamma_k}{\gamma_{k+1}} = 2K + \log_2\tfrac{\gamma_0}{\gamma_{K-1}} \quad \leq \quad 2K + \log_2\tfrac{\gamma_0}{\bar{\gamma}_K}, \quad (46)$$

where $\bar{\gamma}_K := \min_{0 \leq k \leq K-1}\gamma_k$.

Note also that Algorithm 3 with $\mathbf{H}(\boldsymbol{x}_k) \equiv \nabla^2 f(\boldsymbol{x}_k)$ (exact Hessian) is equivalent to the Super-Universal Newton Method from (Doikov et al., 2024a), using a different stopping condition in the adaptive search. Even in the exact case, out theory enhances the complexity results from (Doikov et al., 2024a) to the broader classes of generalized Self-Concordant functions (Sun & Tran-Dinh, 2018) and beyond, including problems with $(L_0, L_1)$-functions (Zhang et al., 2019).

Moreover, our results allow us to use an arbitrary positive semidefinite approximation $\mathbf{H}(\boldsymbol{x}_k) \approx \nabla^2 f(\boldsymbol{x}_k)$ of the Hessian in our methods, and all our algorithms are applicable to possibly non-convex problems as well, while the method in (Doikov et al., 2024a) works primarily for convex optimization, using the exact Hessian.

We establish the following result about this algorithm.

---

**Theorem 4.** *Let $\varepsilon > 0$ be fixed, and assume that there exists $\gamma_\star > 0$ satisfying*

$$\gamma_\star \quad \leq \quad \inf\Big\{\gamma(\boldsymbol{x}, F'(\boldsymbol{x})) \; : \; \boldsymbol{x} \in \mathfrak{F}_\varepsilon \cap \mathcal{F}_0, \; F'(\boldsymbol{x}) \in \partial F(\boldsymbol{x})\Big\}.$$

*Let $\delta := \tfrac{\varepsilon}{D}$. Assume that Algorithm 3 does not stop for the first $K \geq 1$ iterations, and that $\boldsymbol{x}_k \in \mathfrak{F}_\varepsilon$ for all $0 \leq k \leq K$. Then,*

$$K \quad \leq \quad \tfrac{16D}{\min\{\gamma_0, \gamma_\star\}}\ln\tfrac{F(\boldsymbol{x}_0) - F^\star}{\varepsilon} + 2\ln\tfrac{\|F'(\boldsymbol{x}_0)\|_* D}{\varepsilon} \quad (47)$$

*and the total number of oracle calls during these iterations is bounded as*

$$N_K \quad \leq \quad 2K + \log_2\tfrac{\gamma_0}{\gamma_\star}.$$

---

*Proof.* First, we note that the method is well-defined. Indeed, by our assumption, there exists a global value of $\gamma_\star$ such that the first stopping condition of the adaptive search will be satisfied at least for $t_k \geq 0$ such that $2^{-t_k} \cdot \gamma_k \leq \gamma_\star$, unless $\|F'(\boldsymbol{x}_k)\|_* \leq \delta$. The last inequality implies that we solved the problem with the desired accuracy and we stop the algorithm.

Therefore, by induction we have the following lower bound on values of our step-sizes:

$$\gamma_k \;\geq\; \min\{\gamma_0, \gamma_\star\}, \qquad 0 \leq k \leq K. \tag{48}$$

Hence, for every iteration $k \geq 0$ of the method, we ensure

$$F(\boldsymbol{x}_k) - F(\boldsymbol{x}_{k+1}) \;\geq\; \frac{\gamma_{k+1}}{16} \frac{\|F'(\boldsymbol{x}_{k+1})\|_*^2}{\|F'(\boldsymbol{x}_k)\|_*} \overset{(48)}{\geq} \frac{\min\{\gamma_\star, \gamma_0\}}{16} \frac{\|F'(\boldsymbol{x}_{k+1})\|_*^2}{\|F'(\boldsymbol{x}_k)\|_*}.$$

Now, repeating the reasoning from the proof of Theorem 3 we establish (47), and using (46) we immediately obtain the bound on the total number of oracle calls. $\qquad\square$

# E  CONVERGENCE FOR NON-CONVEX FUNCTIONS

First, let us formulate Theorem 1 for more general composite optimization problems (25). Then, Theorem 1 is a direct consequence of this result for $\psi \equiv 0$.

> **Theorem 5.** *Let $K \geq 1$ be a fixed number of iterations of Algorithm 2 and let (5) hold for every step. Assume that $\min_{1 \leq i \leq K} \|F'(\boldsymbol{x}_i)\|_* \geq \varepsilon$ and let $\gamma_\star = \min_{1 \leq i \leq K} \gamma_i > 0$. Then,*
>
> $$K \;\leq\; \frac{8F_0}{\gamma_\star \varepsilon} + \log \frac{\|F'(\boldsymbol{x}_0)\|_*}{\varepsilon}, \qquad where \quad F_0 := F(\boldsymbol{x}_0) - F^\star. \tag{49}$$

*Proof.* According to (5), we have for every iteration of the method,

$$F(\boldsymbol{x}_k) - F(\boldsymbol{x}_{k+1}) \;\geq\; \frac{\gamma_k}{8} \frac{\|F'(\boldsymbol{x}_{k+1})\|_*^2}{\|F'(\boldsymbol{x}_k)\|_*} \;\geq\; \frac{\gamma_\star \varepsilon}{8} \frac{\|F'(\boldsymbol{x}_{k+1})\|_*}{\|F'(\boldsymbol{x}_k)\|_*}.$$

Telescoping this bound and using the inequality between arithmetic and geometric means, we get

$$F(\boldsymbol{x}_0) - F^\star \;\geq\; F(\boldsymbol{x}_0) - F(\boldsymbol{x}_k) \;\geq\; \frac{k\gamma_\star \varepsilon}{8} \frac{1}{k} \sum_{i=1}^{k} \frac{\|F'(\boldsymbol{x}_i)\|_*}{\|F'(\boldsymbol{x}_{i-1})\|_*} \;\geq\; \frac{k\gamma_\star \varepsilon}{8} \left[ \frac{\|F'(\boldsymbol{x}_k)\|_*}{\|F'(\boldsymbol{x}_0)\|_*} \right]^{1/k}$$

$$\geq\; \frac{k\gamma_\star \varepsilon}{8} \left[ \frac{\varepsilon}{\|F'(\boldsymbol{x}_0)\|_*} \right]^{1/k} \;=\; \frac{k\gamma_\star \varepsilon}{8} \exp\left[ \frac{1}{k} \log \frac{\varepsilon}{\|F'(\boldsymbol{x}_0)\|_*} \right]$$

$$\geq\; \frac{k\gamma_\star \varepsilon}{8} \left[ 1 + \frac{1}{k} \log \frac{\varepsilon}{\|F'(\boldsymbol{x}_0)\|_*} \right].$$

Rearranging the terms proves the required complexity bound. $\qquad\square$

We see that this result is very general: we did not specify anything about the problem class our function belongs to. Theorem 5 shows that for general composite objectives, with possibly non-convex smooth part, our method will have a global convergence to a stationary point. To quantify the convergence rate further, we need to impose some structural assumption on the Gradient-Normalized Smoothness $\gamma(\cdot)$ of the function. Following our assumption 10 from the main part, let us assume that $\gamma(\cdot)$ is lower bounded by the harmonic mean of monomials of (sub)gradient norms:

> $$\gamma(\boldsymbol{x}, F'(\boldsymbol{x})) \;\geq\; \pi(\|F'(\boldsymbol{x})\|_*) \;:=\; \left( \sum_{i=1}^{d} \frac{M_{1-\alpha_i}}{\|F'(\boldsymbol{x})\|_*^{\alpha_i}} \right)^{-1} \;\geq\; \frac{1}{d} \min_{1 \leq i \leq d} \frac{\|F'(\boldsymbol{x})\|_*^{\alpha_i}}{M_{1-\alpha_i}}, \tag{50}$$

where for all $i$, $0 \leq \alpha_i \leq 1$ are fixed degrees and $\{M_{1-\alpha}\}_{0 \leq \alpha \leq 1}$ are non-negative coefficients describing the complexity of the problem. Note that this assumption holds for all particular examples of problem classes that we consider (see Section 2). Substituting this bound into Theorem 5, we immediately obtain the following corollary.

> **Corollary 5.** *Let us choose $\gamma_k = \gamma(\boldsymbol{x}_k)$ in Algorithm 2, or by performing an adaptive search. Under assumptions of Theorem 5, we can bound $\gamma_\star \geq \pi(\varepsilon)$. Therefore, to ensure $\min_{1 \leq i \leq K} \|F'(\boldsymbol{x}_i)\|_* \leq \varepsilon$ it is enough to perform a number of iterations of*
>
> $$K \;=\; \left\lceil 8dF_0 \cdot \max_{1 \leq i \leq d} \frac{M_{1-\alpha_i}}{\varepsilon^{1+\alpha_i}} + \log \frac{\|F'(\boldsymbol{x}_0)\|_*}{\varepsilon} \right\rceil.$$

### E.1 Convergence for Inexact Hölder Hessian

Let us consider a particular important consequence of our result. For simplicity, we set $\psi \equiv 0$ (unconstrained minimization). We assume that the Hessian of $f$ is Hölder continuous of a certain degree $0 \leq \nu \leq 1$ (Example 1). Then, according to 7, the Gradient-Normalized Smoothness $\gamma_{\text{NEWTON}}(\cdot)$ using the *exact Hessian*, is bounded by

$$\gamma_{\text{NEWTON}}(\boldsymbol{x}) \geq \left( \tfrac{1+\nu}{L_{2,\nu}} \|\nabla f(\boldsymbol{x})\|_* \right)^{\frac{1}{1+\nu}}. \tag{51}$$

At the same time, in out method we use *inexact Hessian matrix*, with the following general guarantee (see Section 5):

$$\|\nabla^2 f(\boldsymbol{x}) - \mathbf{H}(\boldsymbol{x})\|_* \leq \mathbf{C_1} + \mathbf{C_2} \|\nabla f(\boldsymbol{x})\|_*^{1-\beta}, \tag{52}$$

for a certain $0 \leq \beta \leq 1$. Then, according to the basic properties, we can lower bound the Gradient-Normalized Smoothness $\gamma(\cdot)$ for our problem and with inexact Hessian, as follows:

$$\gamma(\boldsymbol{x}) \geq \left[ \gamma_{\text{NEWTON}}(\boldsymbol{x})^{-1} + \frac{\mathbf{C_1}}{\|\nabla f(\boldsymbol{x})\|_*} + \frac{\mathbf{C_2}}{\|\nabla f(\boldsymbol{x})\|_*^{\beta}} \right]^{-1}. \tag{53}$$

Therefore, substituting our condition 51, we obtain the following lower bound for the Gradient-Normalized Smoothness, that matches the structure of (50):

$$\gamma(\boldsymbol{x}) \geq \left[ \left( \frac{L_{2,\nu}}{(1+\nu)\|\nabla f(\boldsymbol{x})\|_*} \right)^{\frac{1}{1+\nu}} + \frac{\mathbf{C_1}}{\|\nabla f(\boldsymbol{x})\|_*} + \frac{\mathbf{C_2}}{\|\nabla f(\boldsymbol{x})\|_*^{\beta}} \right]^{-1}.$$

Therefore, out theory immediately provides us with the following complexity result.

> **Corollary 6.** *Let us choose $\gamma_k = \gamma(\boldsymbol{x}_k)$ in Algorithm 2, or by performing an adaptive search, using an inexact Hessian that satisfies (52). Then, to ensure $\min_{1 \leq i \leq K} \|\nabla f(\boldsymbol{x}_i)\|_* \leq \varepsilon$ it is enough to perform a number of iterations of*
>
> $$K = O\left( F_0 \cdot \left[ \left( \frac{L_{2,\nu}}{\varepsilon^{2+\nu}} \right)^{\frac{1}{1+\nu}} + \frac{\mathbf{C_1}}{\varepsilon^2} + \frac{\mathbf{C_2}}{\varepsilon^{1+\beta}} \right] + \log \frac{\|\nabla f(\boldsymbol{x}_0)\|_*}{\varepsilon} \right).$$

For example, for $\nu = 1$ (Lipschitz continuous Hessian), we obtain the complexity of

$$F_0 \cdot O\left( \frac{\sqrt{L_{2,1}}}{\varepsilon^{3/2}} + \frac{\mathbf{C_1}}{\varepsilon^2} + \frac{\mathbf{C_2}}{\varepsilon^{1+\beta}} \right),$$

up to an additive logarithmic term. Note that the first term corresponds to the state-of-the-art rate of the Cubically regularized Newton method (Nesterov & Polyak, 2006; Cartis et al., 2011a). We see that when $\mathbf{C_1} \approx 0$ and $\beta \leq 1/2$, which corresponds to our examples, Algorithm 1 with inexact Hessian has the same complexity as the exact Newton method. In the following section, we show how to improve these complexity bounds further, under additional assumptions on our objective, such as convexity.

## F    Improved Rates for Gradient-Dominated Functions

In this section, let us assume additionally that the objective function $F$ satisfies the following inequality, for a certain $0 \leq c \leq 1$ and constant $D_c > 0$ (see (Nesterov & Polyak, 2006; Fatkhullin et al., 2022; Doikov et al., 2024a)):

$$\|F'(\boldsymbol{x}_k)\|_*^{1+c} D_c \geq F_k := F(\boldsymbol{x}_k) - F^\star, \qquad k \geq 0. \tag{54}$$

Let us denote by $\mathcal{F}_0$ the initial level set of the function

$$\mathcal{F}_0 := \left\{ \boldsymbol{x} \in Q : F(\boldsymbol{x}) \leq F(\boldsymbol{x}_0) \right\}.$$

Note that due to Lemma 5, all iterations of our method belong to this set: $\{\boldsymbol{x}_k\}_{k \geq 0} \subset \mathcal{F}_0$. Then, we denote by $D_0$ the diameter of this set, which we assume to be bounded:

$$D_0 := \sup_{\boldsymbol{x}, \boldsymbol{y} \in \mathcal{F}_0} \|\boldsymbol{x} - \boldsymbol{y}\| < +\infty.$$

- **Convex Functions.** Assume that $F$ is convex. Then,

$$F(\boldsymbol{x}_k) - F^\star \;\leq\; \langle F'(\boldsymbol{x}_k), \boldsymbol{x}_k - \boldsymbol{x}^\star \rangle \;\leq\; \|F'(\boldsymbol{x}_k)\|_* \|\boldsymbol{x}_k - \boldsymbol{x}^\star\| \;\leq\; \|F'(\boldsymbol{x}_k)\|_* D_0.$$

  Therefore, inequality (54) is satisfied with $c := 0$.

- **Uniformly Convex Functions.** Assume that $F$ satisfies the following inequality, for a certain $p \geq 2$ and $\sigma_p > 0$ (see (Nesterov, 2018)):

$$F(\boldsymbol{y}) \;\geq\; F(\boldsymbol{x}) + \langle F'(\boldsymbol{x}), \boldsymbol{y} - \boldsymbol{x} \rangle + \tfrac{\sigma_p}{p} \|\boldsymbol{y} - \boldsymbol{x}\|^p. \tag{55}$$

  Then (54) is satisfied with

$$c \;:=\; \tfrac{1}{p-1} \qquad \text{and} \qquad D_c \;:=\; \tfrac{p-1}{p}\left(\tfrac{1}{\sigma_p}\right)^{\frac{1}{p-1}}.$$

- **Strongly Convex Functions** correspond to the previous case, when $p = 2$.

We are ready to establish improved global convergence rates for our method under condition (54) of gradient-dominance. Then, Theorem 2 from the main part is a direct consequence of this result for $\psi \equiv 0$ and $c = 0$ (Convex Unconstrained Minimization).

---

**Theorem 6.** *Let us choose $\gamma_k = \gamma(\boldsymbol{x}_k)$ in Algorithm 2 or by performing an adaptive search. Let the Gradient-Normalized Smoothness $\gamma(\cdot)$ satisfies our structural assumption (50), and denote $\alpha := \min_{1 \leq i \leq d} \alpha_i$. Assume that $F$ is gradient-dominated (54) of degree*

$$c \;\leq\; \alpha. \tag{56}$$

*Then, for $\varepsilon > 0$, to ensure $F(\boldsymbol{x}_K) - F^\star \leq \varepsilon$, it is enough to perform a number of iterations of*

$$K \;=\; \left\lceil \mathcal{C}(\varepsilon) + 2\log \frac{\|F'(\boldsymbol{x}_0)\|_* D_0}{\varepsilon} \right\rceil,$$

*where*

$$\mathcal{C}(\varepsilon) \;:=\; \tfrac{8d}{\eta} \max_{1 \leq i \leq d} \left( M_{1-\alpha_i} \left[ \tfrac{D_c^{1+\alpha_i}}{\varepsilon^{\alpha_i - \alpha}} \right]^{\frac{1}{1+c}} \right) \left( \tfrac{1}{\varepsilon^\eta} - \tfrac{1}{F_0^\eta} \right), \quad \text{for} \quad \eta := \tfrac{\alpha - c}{1+c} > 0,$$

*and, for $\eta = 0$, we have the limit:*

$$\mathcal{C}(\varepsilon) \;:=\; 8d \max_{1 \leq i \leq d} \left( M_{1-\alpha_i} \left[ \tfrac{D_c^{1+\alpha_i}}{\varepsilon^{\alpha_i - \alpha}} \right]^{\frac{1}{1+c}} \right) \log \tfrac{F_0}{\varepsilon}. \tag{57}$$

---

*Proof.* Let us fix some $k \geq 0$ and assume that $F_k := F(\boldsymbol{x}_k) - F^\star \geq \varepsilon$. Let $g_k := \|F'(\boldsymbol{x}_k)\|_*$. By assumption (50), we have

$$g_k \;\geq\; \left( \tfrac{F_k}{D_c} \right)^{\frac{1}{1+c}}. \tag{58}$$

Then, from Lemma 5, we obtain

$$F_k - F_{k+1} \;\geq\; \tfrac{\gamma_k}{8}\left(\tfrac{g_{k+1}}{g_k}\right)^2 \cdot g_k \;\overset{(50)}{\geq}\; \tfrac{1}{8d}\left(\tfrac{g_{k+1}}{g_k}\right)^2 \cdot \min_{1 \leq i \leq d}\left(\tfrac{g_k^{1+\alpha_i}}{M_{1-\alpha_i}}\right)$$

$$\overset{(54)}{\geq}\; \tfrac{1}{8d}\left(\tfrac{g_{k+1}}{g_k}\right)^2 \cdot \min_{1 \leq i \leq d}\left(\tfrac{1}{M_{1-\alpha_i}}\left[\tfrac{F_k}{D_c}\right]^{\frac{1+\alpha_i}{1+c}}\right). \tag{59}$$

Recall that $\alpha := \min_{1 \leq i \leq d} \alpha_i$. Denote $\eta := \tfrac{\alpha - c}{1+c} \overset{(56)}{\geq} 0$. Applying the Mean Value Theorem for $y(x) = x^\eta$ we get

$$b^\eta - a^\eta \;\geq\; \tfrac{\eta}{b^{1-\eta}}(b-a), \qquad b \geq a \geq 0. \tag{60}$$

Thus, we have, assuming that $\eta > 0$:

$$\frac{1}{\eta}\left(\frac{1}{F_{k+1}^{\eta}} - \frac{1}{F_k^{\eta}}\right) = \frac{F_k^{\eta}-F_{k+1}^{\eta}}{\eta \cdot F_k^{\eta}F_{k+1}^{\eta}} \overset{(60)}{\geq} \frac{F_k-F_{k+1}}{F_kF_{k+1}^{\eta}}$$

$$\overset{(59)}{\geq} \frac{1}{8d}\left(\frac{g_{k+1}}{g_k}\right)^2\left(\frac{F_k}{F_{k+1}}\right)^{\eta}\min_{1\leq i\leq d}\left(\frac{1}{M_{1-\alpha_i}}\left[\frac{F_k^{\alpha_i-\alpha}}{D_c^{1+\alpha_i}}\right]^{\frac{1}{1+c}}\right)$$

$$\geq A(\varepsilon)\cdot\left(\frac{g_{k+1}}{g_k}\right)^2\left(\frac{F_k}{F_{k+1}}\right)^{\eta},$$

where

$$A(\varepsilon) := \frac{1}{8d}\min_{1\leq i\leq d}\left(\frac{1}{M_{1-\alpha_i}}\left[\frac{\varepsilon^{\alpha_i-\alpha}}{D_c^{1+\alpha_i}}\right]^{\frac{1}{1+c}}\right).$$

Telescoping this bound and using the inequality between arithmetic and geometric means, we obtain

$$\frac{1}{\eta}\left(\frac{1}{F_k^{\eta}} - \frac{1}{F_0^{\eta}}\right) \geq A(\varepsilon)\cdot\sum_{i=1}^{k-1}\left(\frac{g_{i+1}}{g_i}\right)^2\left(\frac{F_i}{F_{i+1}}\right)^{\eta} \geq kA(\varepsilon)\cdot\left(\frac{g_k^2F_0^{\eta}}{g_0^2F_k^{\eta}}\right)^{\frac{1}{k}}$$

$$\geq kA(\varepsilon)\cdot\left(\frac{F_0^{\eta}\varepsilon^{2-\eta}}{g_0^2D_0^2}\right)^{\frac{1}{k}} \geq kA(\varepsilon)\cdot\left(\frac{\varepsilon}{g_0D_0}\right)^{\frac{2}{k}}$$

$$= kA(\varepsilon)\cdot\exp\left(-\frac{2}{k}\log\frac{g_0D_0}{\varepsilon}\right) \geq kA(\varepsilon)\cdot\left(1-\frac{2}{k}\log\frac{g_0D_0}{\varepsilon}\right).$$

Rearranging the terms, we get

$$k \leq \frac{1}{\eta A(\varepsilon)}\left(\frac{1}{\varepsilon^{\eta}} - \frac{1}{F_0^{\eta}}\right) + 2\log\frac{g_0D_0}{\varepsilon}.$$

Note that for $\eta = 0$, we can use the following limit

$$\lim_{\eta\to 0}\frac{1}{\eta}\left(\frac{1}{a^{\eta}} - \frac{1}{b^{\eta}}\right) = \log\frac{a}{b}, \qquad a,b > 0.$$

Therefore, in this case, we obtain

$$k \leq \frac{1}{A(\varepsilon)}\log\frac{F_0}{\varepsilon} + 2\log\frac{g_0D_0}{\varepsilon},$$

which completes the proof. $\qquad\square$

Let us consider an important particular case of convex functions ($c = 0$), and for specific assumptions on smoothness. In these cases and for the exact Hessian, we have that $\gamma(\boldsymbol{x}_k, F'(\boldsymbol{x}_k)) \geq \pi(\|F'(\boldsymbol{x}_k)\|_*) = \|F'(\boldsymbol{x}_k)\|_*^{\alpha}M_{1-\alpha}^{-1}$, thus $\pi(\cdot)$ is a simple monomial of degree $0 \leq \alpha \leq 1$.

---

**Corollary 7** (Convex Function). *Consider exact Newton method:* $\mathbf{H}(\boldsymbol{x}_k) := \nabla^2 f(\boldsymbol{x}_k)$.
• *Let the Hessian have bounded variation (Ex. 1, $\nu = 0$), then $\alpha = 1$, $M_0 = L_{2,0}$ and we get:*

$$K = O\left(\frac{M_0D_0^2}{\varepsilon}\right) = O\left(\frac{L_{2,0}D_0^2}{\varepsilon}\right).$$

• *Let the Hessian be Lipschitz continuous (Ex. 1, $\nu = 1$), then $\alpha = 1/2$, $M_{1/2} = \sqrt{L_{2,1}}$, and our method has the same rate as the Cubic Newton (Nesterov & Polyak, 2006):*

$$K = O\left(\frac{M_{1/2}D_0^{3/2}}{\varepsilon^{1/2}}\right) = O\left(\sqrt{\frac{L_{2,1}D_0^3}{\varepsilon}}\right).$$

• *Let the third derivative be Lipschitz continuous (Ex. 2, $\nu = 1$), then $\alpha = 1/3$, $M_{2/3} = L_{3,1}^{1/3}$, and we obtain the rate as that of the third-order tensor method (Nesterov, 2021a):*

$$K = O\left(\frac{M_{2/3}D_0^{4/3}}{\varepsilon^{1/3}}\right) = O\left(\left[\frac{L_{3,1}D_0^4}{\varepsilon}\right]^{1/3}\right).$$

• *Let $f$ be Quasi-Self-Concordant (Ex. 3), then $\alpha = 0$, and we obtain the global liner rate (Doikov, 2023):*

$$K = O\left(M_1D_0\log\frac{F_0}{\varepsilon}\right).$$

---

We see that our theory covers all the known state-of-the-art global convergence rates of the Newton method in a unified manner.

Now, assume that we use an inexact Hessian, $\mathbf{H}(\boldsymbol{x}_k) \approx \nabla^2 f(\boldsymbol{x}_k)$, that satisfies condition (52). Then, the corresponding Gradient-Normalized Smoothness $\gamma(\cdot)$ will be changed accordingly (53) and Theorem 6 leads us to the following result. We assume $\psi \equiv 0$ (unconstrained minimization).

> **Corollary 8** (Inexact Hessian). *Let us choose $\gamma_k = \gamma(\boldsymbol{x}_k)$ in Algorithm 1, or by performing an adaptive search, using an inexact Hessian that satisfies (52). Assume that $c \leq \beta$. Then, to ensure $f(\boldsymbol{x}_K) - f^\star \leq \varepsilon$ it is enough to perform a number of iterations of*
>
> $$K = \widetilde{O}\Big(C_{\text{NEWTON}}(\varepsilon) + \mathbf{C_1}\Big[\tfrac{D_c^2}{\varepsilon^{1-c}}\Big]^{\frac{1}{1+c}} + \mathbf{C_2}\Big[\tfrac{D_c^{1+\beta}}{\varepsilon^{\beta-c}}\Big]^{\frac{1}{1+c}}\Big), \tag{61}$$
>
> *where $C_{\text{NEWTON}}(\varepsilon)$ is the complexity of the method with exact Hessian.*

According to (61), we see that a large degree $c \geq 0$ of gradient dominance helps to accelerate the method. Thus, for $c := 0$ (convex functions), we obtain

$$K \quad = \quad \widetilde{O}\Big(C_{\text{NEWTON}}(\varepsilon) + \mathbf{C_1}\tfrac{D_0^2}{\varepsilon} + \mathbf{C_2}\tfrac{D_0^{1+\beta}}{\varepsilon^\beta}\Big).$$

At the same time, for $c := 1/2$ (e.g., uniformly convex functions of degree 3), we already obtain a complexity of

$$K \quad = \quad \widetilde{O}\Big(C_{\text{NEWTON}}(\varepsilon) + \mathbf{C_1}\tfrac{D_{1/2}^{4/3}}{\varepsilon^{1/3}} + \mathbf{C_2}\Big[\tfrac{D_{1/2}^{1+\beta}}{\varepsilon^{\beta-1/2}}\Big]^{\frac{2}{3}}\Big),$$

which is much better in terms of dependence on $\varepsilon$, etc. It is important that all these rates correspond to the same algorithm, with a universal step-size selection. Therefore, the method is able to *automatically* adapt to the best degree of smoothness and gradient dominance.

Combining Corollary 8 with Corollary 7, we obtain the following classification of complexities, for Convex Unconstrained Minimization ($c = 0$, $\psi \equiv 0$), with inexact Hessians.

> **Corollary 9** (Inexact Hessian: Convex Functions). *Consider inexact Hessians (52).*
> • *Let the Hessian have bounded variation, and $\alpha = \beta = 1$, Then,*
>
> $$K \quad = \quad O\Big(\tfrac{(M_0 + \mathbf{C_2})D_0}{\varepsilon} + \tfrac{\mathbf{C_1}D_0^2}{\varepsilon}\Big).$$
>
> • *Let the Hessian be Lipschitz continuous, and $\alpha = \beta = 1/2$. Then,*
>
> $$K \quad = \quad O\Big(\tfrac{(M_{1/2} + \mathbf{C_2})D_0^{3/2}}{\varepsilon^{1/2}} + \tfrac{\mathbf{C_1}D_0^2}{\varepsilon}\Big).$$
>
> • *Let the third derivative be Lipschitz continuous, and $\alpha = \beta = 1/3$. Then*
>
> $$K \quad = \quad O\Big(\tfrac{(M_{2/3} + \mathbf{C_2})D_0^{4/3}}{\varepsilon^{1/3}} + \tfrac{\mathbf{C_1}D_0^2}{\varepsilon}\Big).$$
>
> • *Let $f$ be Quasi-Self-Concordant, and $\alpha = \beta = 0$. Then*
>
> $$K \quad = \quad O\Big(\Big[(M_1 + \mathbf{C_2})D_0 + \tfrac{\mathbf{C_1}D_0^2}{\varepsilon}\Big]\log \tfrac{F_0}{\varepsilon}\Big).$$

## G  APPLICATIONS

In this section, we provide concrete examples of problems that satisfy our assumptions of smoothness and Hessian approximation, and that offer direct, practical applications of our theory.

Let us study the case of the exact Hessian: $\mathbf{H}(\boldsymbol{x}) \equiv \nabla^2 f(\boldsymbol{x})$, and consider some standard assumptions on the smoothness of our objective. We demonstrate that any such global assumption can be effectively translated into appropriate bounds on our Gradient-Normalized Smoothness $\gamma(\cdot)$. As a consequence, by Theorems 5 and 6, we immediately obtain global convergence guarantees for our algorithms.

For simplicity of our presentation, we always assume that $K \geq 2 \log \frac{\|\nabla f(\boldsymbol{x}_0)\|_*}{\varepsilon}$ in our complexity bounds, to omit an additive logarithmic term.

### G.1 FUNCTIONS WITH HÖLDER HESSIAN

Let us assume that the Hessian of $f$ is Hölder continuous of degree $\nu \in [0, 1]$, with some constant $L_{2,\nu} > 0$:

$$\|\nabla^2 f(\boldsymbol{x}) - \nabla^2 f(\boldsymbol{y})\| \leq L_{2,\nu} \|\boldsymbol{x} - \boldsymbol{y}\|^\nu, \qquad \boldsymbol{x}, \boldsymbol{y} \in \mathbb{R}^n. \tag{62}$$

Therefore, by direct integration, we obtain the following bound, for any $\boldsymbol{h} \in \mathbb{R}^n$:

$$\|\nabla f(\boldsymbol{x} + \boldsymbol{h}) - \nabla f(\boldsymbol{x}) - \nabla^2 f(\boldsymbol{x})\boldsymbol{h}\| \leq \frac{L_{2,\nu}}{1+\nu} \|\boldsymbol{h}\|^{1+\nu}. \tag{63}$$

Now, let us choose $\gamma := \left( \frac{1+\nu}{L_{2,\nu}} \|\boldsymbol{g}\|_* \right)^{\frac{1}{1+\nu}}$, for an arbitrary fixed $\boldsymbol{g} \in \mathbb{R}^n$ and consider $\boldsymbol{h} \in B_\gamma := \{\boldsymbol{h} \in \mathbb{R}^n : \|\boldsymbol{h}\| \leq \gamma\}$. We have

$$\|\nabla f(\boldsymbol{x} + \boldsymbol{h}) - \nabla f(\boldsymbol{x}) - \nabla^2 f(\boldsymbol{x})\boldsymbol{h}\| \overset{(63)}{\leq} \frac{L_{2,\nu} \gamma^\nu}{1+\nu} \|\boldsymbol{h}\| = \frac{\|\boldsymbol{g}\|_* \|\boldsymbol{h}\|}{\gamma}, \tag{64}$$

where the last equation holds due to our choice of $\gamma$. By our definition $\gamma(\boldsymbol{x}, \boldsymbol{g})$ is the *maximal* such value that (64) holds. Hence, we obtain the following bound.

> **Proposition 1.** *Let $f$ satisfy (62) for some $\nu \in [0, 1]$ and $L_{2,\nu} > 0$. Then,*
>
> $$\gamma_f(\boldsymbol{x}, \boldsymbol{g}) \geq \left( \frac{1+\nu}{L_{2,\nu}} \|\boldsymbol{g}\|_* \right)^{\frac{1}{1+\nu}}.$$

Plugging this estimate into Theorem 5 we obtain the complexity to find $\|F'(\boldsymbol{x}_k)\|_* \leq \varepsilon$ of order

$$K = O\left( \frac{F_0 L_{2,\nu}^{1/(1+\nu)}}{\varepsilon^{(2+\nu)/(1+\nu)}} \right)$$

for our algorithms, up to an additive logarithmic terms. For $\nu = 1$, this corresponds to the complexity of the Cubic Newton method (Nesterov & Polyak, 2006), and for $\nu = 0$, this is the same rate as for the Gradient Descent on general non-convex problems (Nesterov, 2018). At the same time, for convex problems (Theorem 6, $c = 0$), we get the complexity to find global solution in terms of the functional residual, $F(\boldsymbol{x}_K) - F^\star \leq \varepsilon$ of order

$$K = O\left( \left[ \frac{L_{2,\nu} D_0^{2+\nu}}{\varepsilon} \right]^{\frac{1}{1+\nu}} \right).$$

### G.2 CONVEX FUNCTIONS WITH HÖLDER THIRD DERIVATIVE

Now, we assume that function $f$ is convex and its third derivative is Hölder continuous of degree $\nu \in [0, 1]$, with constant $L_{3,\nu} > 0$:

$$\|\nabla^3 f(\boldsymbol{y}) - \nabla^3 f(\boldsymbol{x})\| \leq L_{3,\nu} \|\boldsymbol{x} - \boldsymbol{y}\|^\nu, \qquad \boldsymbol{x}, \boldsymbol{y} \in \mathbb{R}^n. \tag{65}$$

Following (Nesterov, 2021a; Doikov et al., 2024a), we can integrate this inequality and, using convexity, obtain, for an arbitrary directions $\boldsymbol{v} \in \mathbb{R}^n$ and $\boldsymbol{u} \in \mathbb{R}^n$:

$$0 \leq \langle \nabla^2 f(\boldsymbol{x} + \boldsymbol{v})\boldsymbol{h}, \boldsymbol{h} \rangle \leq \langle \nabla^2 f(\boldsymbol{x})\boldsymbol{h}, \boldsymbol{h} \rangle + \nabla^3 f(\boldsymbol{x})[\boldsymbol{h}, \boldsymbol{h}, \boldsymbol{v}] + \frac{L_{3,\nu}}{1+\nu} \|\boldsymbol{h}\|^2 \cdot \|\boldsymbol{v}\|^{1+\nu}.$$

Now, substituting $\boldsymbol{v} = \pm\tau\boldsymbol{u}$, for some $\tau > 0$ and $\boldsymbol{u} \in \mathbb{R}^n$, we get

$$|\nabla^3 f(\boldsymbol{x})[\boldsymbol{h}, \boldsymbol{h}, \boldsymbol{u}]| \leq \frac{1}{\tau} \langle \nabla^2 f(\boldsymbol{x})\boldsymbol{h}, \boldsymbol{h} \rangle + \tau^\nu \cdot \frac{L_{3,\nu}}{1+\nu} \|\boldsymbol{h}\|^2 \cdot \|\boldsymbol{u}\|^{1+\nu}.$$

Balancing the right hand side, we can choose $\tau := \left( \frac{(1+\nu)\langle \nabla^2 f(\boldsymbol{x})\boldsymbol{h}, \boldsymbol{h} \rangle}{L_{3,\nu} \|\boldsymbol{h}\|^2 \|\boldsymbol{u}\|^{1+\nu}} \right)^{\frac{1}{1+\nu}}$, which gives:

$$
\begin{aligned}
|\nabla^3 f(\boldsymbol{x})[\boldsymbol{h}, \boldsymbol{h}, \boldsymbol{u}]| &\leq 2 \cdot \left( \frac{L_{3,\nu}}{1+\nu} \right)^{\frac{1}{1+\nu}} \langle \nabla^2 f(\boldsymbol{x})\boldsymbol{h}, \boldsymbol{h} \rangle^{\frac{\nu}{1+\nu}} \cdot \|\boldsymbol{h}\|^{\frac{2}{1+\nu}} \cdot \|\boldsymbol{u}\| \\
&= 2 \cdot \left( \frac{L_{3,\nu}}{1+\nu} \right)^{\frac{1}{1+\nu}} \|\boldsymbol{h}\|_{\boldsymbol{x}}^{\frac{2\nu}{1+\nu}} \cdot \|\boldsymbol{h}\|^{\frac{2}{1+\nu}} \cdot \|\boldsymbol{u}\|, \qquad \boldsymbol{h}, \boldsymbol{u} \in \mathbb{R}^n.
\end{aligned}
\tag{66}
$$

Then, using Taylor's formula for the gradient approximation, we obtain

$$\|\nabla f(\boldsymbol{x} + \boldsymbol{h}) - \nabla f(\boldsymbol{x}) - \nabla^2 f(\boldsymbol{x})\boldsymbol{h} - \tfrac{1}{2}\nabla^3 f(\boldsymbol{x})[\boldsymbol{h}, \boldsymbol{h}]\|_* \overset{(65)}{\leq} \frac{L_{3,\nu}}{(1+\nu)(2+\nu)}\|\boldsymbol{h}\|^{2+\nu}.$$

Hence, applying the triangle inequality and our bound (66), we conclude that

$$\|\nabla f(\boldsymbol{x} + \boldsymbol{h}) - \nabla f(\boldsymbol{x}) - \nabla^2 f(\boldsymbol{x})\boldsymbol{h}\|_*$$
$$\leq \frac{L_{3,\nu}}{(1+\nu)(2+\nu)}\|\boldsymbol{h}\|^{2+\nu} + \left(\frac{L_{3,\nu}}{1+\nu}\right)^{\frac{1}{1+\nu}}\|\boldsymbol{h}\|_{\boldsymbol{x}}^{\frac{2\nu}{1+\nu}} \cdot \|\boldsymbol{h}\|^{\frac{2}{1+\nu}}. \tag{67}$$

Now, for an arbitrary $\boldsymbol{g} \in \mathbb{R}^n$ and a fixed $\gamma > 0$, we consider only the directions $\boldsymbol{h} \in B_\gamma \cap \mathcal{O}_{\boldsymbol{x},\boldsymbol{g}}$, i.e. it holds:

$$\|\boldsymbol{h}\| \leq \gamma \quad \text{and} \quad \|\boldsymbol{h}\|_{\boldsymbol{x}}^2 \leq -\langle \boldsymbol{g}, \boldsymbol{h}\rangle \leq \|\boldsymbol{g}\|_*\|\boldsymbol{h}\|.$$

For such directions, we can continue our bound, as follows:

$$\|\nabla f(\boldsymbol{x} + \boldsymbol{h}) - \nabla f(\boldsymbol{x}) - \nabla^2 f(\boldsymbol{x})\boldsymbol{h}\|_* \leq \frac{L_{3,\nu}\gamma^{1+\nu}}{(1+\nu)(2+\nu)}\|\boldsymbol{h}\| + \left(\frac{L_{3,\nu}}{1+\nu}\right)^{\frac{1}{1+\nu}}\gamma^{\frac{1}{1+\nu}}\|\boldsymbol{g}\|_*^{\frac{\nu}{1+\nu}}\|\boldsymbol{h}\|.$$

Now, we notice that to ensure

$$\frac{L_{3,\nu}\gamma^{1+\nu}}{(1+\nu)(2+\nu)} + \left(\frac{L_{3,\nu}}{1+\nu}\right)^{\frac{1}{1+\nu}}\gamma^{\frac{1}{1+\nu}}\|\boldsymbol{g}\|_*^{\frac{\nu}{1+\nu}} \leq \frac{\|\boldsymbol{g}\|_*}{\gamma},$$

it is sufficient to choose

$$\gamma := \left(\frac{1+\nu}{2^{1+\nu}L_{3,\nu}}\|\boldsymbol{g}\|_*\right)^{\frac{1}{2+\nu}}.$$

Therefore, we finally conclude the following bound.

> **Proposition 2.** *Let $f$ be convex and satisfy (65) for some $\nu \in [0,1]$ and $L_{3,\nu} > 0$. Then,*
> $$\gamma_f(\boldsymbol{x}, \boldsymbol{g}) \geq \left(\frac{1+\nu}{2^{1+\nu}L_{3,\nu}}\|\boldsymbol{g}\|_*\right)^{\frac{1}{2+\nu}}.$$

Using this estimate with Theorem 6, for convex problems ($c = 0$), we get the complexity to find global solution in terms of the functional residual, $F(\boldsymbol{x}_K) - F^\star \leq \varepsilon$ of order

$$K = O\left(\left[\frac{L_{3,\nu}D_0^{3+\nu}}{\varepsilon}\right]^{\frac{1}{2+\nu}}\right) \tag{68}$$

for our algorithm. For $\nu = 1$, this gives $O\left([1/\varepsilon]^{\frac{1}{3}}\right)$. This result recovers fast rates for the Super-Universal Newton method from (Doikov et al., 2024a). Note that our algorithms and theory generalize these rate to the case of inexact Hessian (see Corollaries 8 and 9).

### G.3 Quasi-Self-Concordant Functions

Important in applications with softmax problems, logistic and exponential regressions, matrix balancing and matrix scaling problems, are convex objectives that satisfy the following condition (Bach, 2010; Sun & Tran-Dinh, 2018; Karimireddy et al., 2018; Carmon et al., 2020; Doikov, 2023), with some parameter $M_1 \geq 0$:

$$\langle \nabla^3 f(\boldsymbol{x})\boldsymbol{h}, \boldsymbol{h}, \boldsymbol{u}\rangle \leq M_1\|\boldsymbol{h}\|_{\boldsymbol{x}}^2\|\boldsymbol{u}\|, \qquad \boldsymbol{x}, \boldsymbol{h}, \boldsymbol{u} \in \mathbb{R}^n. \tag{69}$$

By integrating this inequality, we obtain (see, e.g., Lemma 2.7 in (Doikov, 2023)), for any $\boldsymbol{x}, \boldsymbol{h} \in \mathbb{R}^n$:

$$\|\nabla f(\boldsymbol{x} + \boldsymbol{h}) - \nabla f(\boldsymbol{x}) - \nabla^2 f(\boldsymbol{x})\boldsymbol{h}\|_* \leq M_1\|\boldsymbol{h}\|_{\boldsymbol{x}}^2\varphi(M_1\|\boldsymbol{h}\|), \tag{70}$$

where $\varphi(t) := \frac{e^t - t - 1}{t^2} \geq 0$ is a convex and monotone function. Now, let us assume that $\boldsymbol{h} \in B_\gamma \cap \mathcal{O}_{\boldsymbol{x},\boldsymbol{g}}$, for an arbitrary $\gamma > 0$ and $\boldsymbol{g} \in \mathbb{R}^n$:

$$\|\boldsymbol{h}\| \leq \gamma \quad \text{and} \quad \|\boldsymbol{h}\|_{\boldsymbol{x}}^2 \leq -\langle \boldsymbol{g}, \boldsymbol{h}\rangle \leq \|\boldsymbol{g}\|_*\|\boldsymbol{h}\|. \tag{71}$$

Substituting these bounds into (70), we get

$$\|\nabla f(\boldsymbol{x} + \boldsymbol{h}) - \nabla f(\boldsymbol{x}) - \nabla^2 f(\boldsymbol{x})\boldsymbol{h}\|_* \leq M_1\|\boldsymbol{g}\|_*\|\boldsymbol{h}\| \cdot \varphi(\gamma M_1),$$

and for $\gamma := \frac{1}{M_1}$ we ensure $M_1 \varphi(\gamma M_1) \leq \frac{1}{\gamma}$. Hence, we have established the following result.

---

**Proposition 3.** *Let $f$ be Quasi-Self-Concordant (69) for some $M_1 > 0$. Then,*
$$\gamma_f(\boldsymbol{x}, \boldsymbol{g}) \quad \geq \quad \tfrac{1}{M_1}.$$

---

Using this bound on Gradient-Normalized Smoothness in our Theorem 6 for convex functions ($c = 0$) immediately gives the global linear rate of convergence for our method, and to achieve $F(\boldsymbol{x}_K) - F^\star \leq \varepsilon$, it is enough to perform

$$K \quad = \quad O\Big( M_1 D_0 \cdot \log \tfrac{F_0}{\varepsilon} \Big)$$

iterations of the algorithm.

Note that this is the same rate established in (Doikov, 2023) for the Newton method with gradient regularization. This result shows that the Newton method can achieve a global linear rate of convergence without any additional assumptions of uniform or strong convexity on the objective. In contrast, first-order methods can attain only sublinear rates on these problems unless additional regularization is applied.

In this work, we generalize this result to methods with an *inexact Hessian*. It appears that, as soon as $\mathbf{C_1} \approx 0$ and $\beta = 0$ in condition (12), the method with an inexact Hessian has *the same fast global rate* as the full Newton method. Remarkably, as we show, this holds—for example—for the logistic regression problem (Example 6), where the Fisher approximation of the Hessian yields $\mathbf{C_1} = f^\star$ (the global optimum), which can be close to zero in well-separable case.

### G.4 GENERALIZED SELF-CONCORDANT FUNCTIONS

Combining the previous two examples, let us consider the following class of convex *Generalized Self-Concordant* functions (Sun & Tran-Dinh, 2018), for some degree $0 \leq q < 2$ and $G_q \geq 0$:

$$\nabla^3 f(\boldsymbol{x})[\boldsymbol{h}, \boldsymbol{h}, \boldsymbol{u}] \quad \leq \quad G_q \|\boldsymbol{h}\|_{\boldsymbol{x}}^q \|\boldsymbol{h}\|^{2-q} \|\boldsymbol{u}\|, \qquad \boldsymbol{x}, \boldsymbol{h}, \boldsymbol{u} \in \mathbb{R}^n. \tag{72}$$

Note that $q = 2$ corresponds to Quasi-Self-Concordant functions (69), and for the convex functions with Hölder continuous third derivative (66) of degree $\nu \in [0, 1]$, we have $q = \frac{2\nu}{1+\nu}$. Let us present the following example that provides us with all intermediate powers $0 \leq q < 2$.

---

**Example 9.** *For $p \geq 2$, consider*
$$f(\boldsymbol{x}) \quad = \quad \tfrac{1}{p} \|\boldsymbol{x}\|^p.$$

*Then, (72) is satisfied with $q := \frac{2(p-3)}{p-2}$ and $G_q := (p-1)(p-2)$.*

---

*Proof.* Indeed, for arbitrary $\boldsymbol{h}, \boldsymbol{u} \in \mathbb{R}^n$, we have:

$$\langle \nabla f(\boldsymbol{x}), \boldsymbol{h} \rangle \quad = \quad \|\boldsymbol{x}\|^{p-2} \langle \mathbf{B}\boldsymbol{x}, \boldsymbol{h} \rangle$$

$$\langle \nabla^2 f(\boldsymbol{x})\boldsymbol{h}, \boldsymbol{h} \rangle \quad = \quad (p-2)\|\boldsymbol{x}\|^{p-4} \langle \mathbf{B}\boldsymbol{x}, \boldsymbol{h} \rangle^2 + \|\boldsymbol{x}\|^{p-2}\|\boldsymbol{h}\|^2 \quad \geq \quad \|\boldsymbol{x}\|^{p-2}\|\boldsymbol{h}\|^2,$$

$$\nabla^3 f(\boldsymbol{x})[\boldsymbol{h}, \boldsymbol{h}, \boldsymbol{u}] \quad = \quad 2(p-2)\|\boldsymbol{x}\|^{p-4}\langle \mathbf{B}\boldsymbol{x}, \boldsymbol{h} \rangle \langle \mathbf{B}\boldsymbol{u}, \boldsymbol{h} \rangle$$

$$+ (p-2)(p-4)\|\boldsymbol{x}\|^{p-6}\langle \mathbf{B}\boldsymbol{x}, \boldsymbol{u} \rangle \langle \mathbf{B}\boldsymbol{x}, \boldsymbol{h} \rangle^2$$

$$+ (p-2)\|\boldsymbol{x}\|^{p-4}\|\boldsymbol{h}\|^2 \langle \mathbf{B}\boldsymbol{x}\boldsymbol{u} \rangle$$

$$\leq \quad (p-1)(p-2)\|\boldsymbol{x}\|^{p-3}\|\boldsymbol{h}\|^2\|\boldsymbol{u}\|$$

$$\leq \quad (p-1)(p-2)\|\boldsymbol{h}\|_{\boldsymbol{x}}^{\frac{2(p-3)}{p-2}}\|\boldsymbol{h}\|^{\frac{2}{p-2}}\|\boldsymbol{u}\|,$$

which is the required bound. $\qquad \square$

Using direct computation, we also immediately obtain the following simple proposition.

> **Proposition 4** (Affine Substitution)**.** *Let $f$ satisfy (72) for some $0 \le q < 2$ and $G_q > 0$. Then, $g(\boldsymbol{x}) := f(\mathbf{A}\boldsymbol{x} - \boldsymbol{b})$ satisfy (72) with the same degree $q$ and constant $G_q$, by correcting the global norm accordingly:* $\mathbf{B}' := \mathbf{A}^\top \mathbf{B} \mathbf{A}$.

Now, let us fix a point $\boldsymbol{x} \in \mathbb{R}^n$. For arbitrary given directions $\boldsymbol{u}, \boldsymbol{h} \in \mathbb{R}^n$, we denote the following univariate function

$$\varphi(\tau) := \tfrac{2}{2-q} \langle \nabla^2 f(\boldsymbol{x} + \tau \boldsymbol{u}) \boldsymbol{h}, \boldsymbol{h} \rangle^{\frac{2-q}{2}}.$$

Then,

$$|\varphi'(\tau)| = \left| \frac{\nabla^3 f(\boldsymbol{x}+\tau\boldsymbol{u})[\boldsymbol{h},\boldsymbol{h},\boldsymbol{u}]}{\nabla^2 f(\boldsymbol{x}+\tau\boldsymbol{u})\boldsymbol{h},\boldsymbol{h}\rangle^{\frac{q}{2}}} \right| \overset{(72)}{\le} G_q \|\boldsymbol{h}\|^{2-q} \|\boldsymbol{u}\|. \tag{73}$$

Hence, for arbitrary $\boldsymbol{x}, \boldsymbol{y}, \boldsymbol{h} \in \mathbb{R}^n$, setting $\boldsymbol{u} := \boldsymbol{y} - \boldsymbol{x}$, we have

$$\|\boldsymbol{h}\|_{\boldsymbol{y}}^{2-q} - \|\boldsymbol{h}\|_{\boldsymbol{x}}^{2-q} = \tfrac{2-q}{2}\big(\varphi(1) - \varphi(0)\big) \overset{(73)}{\le} \tfrac{2-q}{2} G_q \|\boldsymbol{h}\|^{2-q} \|\boldsymbol{y} - \boldsymbol{x}\|. \tag{74}$$

Therefore, for an arbitrary $\boldsymbol{h} \in \mathbb{R}^n$ and $\boldsymbol{u} \in \mathbb{R}^n$ such that $\|\boldsymbol{u}\| = 1$ we have

$$\langle \nabla f(\boldsymbol{x} + \boldsymbol{h}) - \nabla f(\boldsymbol{x}) - \nabla^2 f(\boldsymbol{x})\boldsymbol{h}, \boldsymbol{u} \rangle = \int_0^1 (1 - \tau)\nabla^3 f(\boldsymbol{x} + \tau\boldsymbol{h})[\boldsymbol{h}, \boldsymbol{h}, \boldsymbol{u}]d\tau$$

$$\overset{(72)}{\le} G_q \|\boldsymbol{h}\|^{2-q} \int_0^1 (1 - \tau)\|\boldsymbol{h}\|_{\boldsymbol{x}+\tau\boldsymbol{h}}^q d\tau$$

$$\overset{(74)}{\le} G_q \|\boldsymbol{h}\|^{2-q} \int_0^1 (1 - \tau)\Big(\|\boldsymbol{h}\|_{\boldsymbol{x}}^{2-q} + \tfrac{2-q}{2}G_q\|\boldsymbol{h}\|^{3-q}\tau\Big)^{\frac{q}{2-q}} d\tau$$

$$\le 2^{\frac{q}{2-q}} G_q \|\boldsymbol{h}\|^{2-q} \cdot \left[ \|\boldsymbol{h}\|_{\boldsymbol{x}}^q \int_0^1 (1 - \tau)d\tau + \Big(\tfrac{2-q}{2}G_q\Big)^{\frac{q}{2-q}} \|\boldsymbol{h}\|^{\frac{q(3-q)}{2-q}} \int_0^1 (1 - \tau)\tau^{\frac{q}{2-q}} d\tau \right]$$

$$= 2^{\frac{2(q+1)}{2-q}} G_q \|\boldsymbol{h}\|^{2-q} \|\boldsymbol{h}\|_{\boldsymbol{x}}^q + \frac{(2-q)^{\frac{4-q}{2-q}}}{2 \cdot (4-q)} G_q^{\frac{2}{2-q}} \|\boldsymbol{h}\|^{\frac{4-q}{2-q}}.$$

Therefore, we have proved the following bound.

> **Proposition 5.** *Let $f$ satisfy (72) for some $0 \le q < 2$ and $G_q > 0$. Then,*
>
> $$\|\nabla f(\boldsymbol{x} + \boldsymbol{h}) - \nabla f(\boldsymbol{x}) - \nabla^2 f(\boldsymbol{x})\boldsymbol{h}\|_* \le c_1 \cdot G_q \|\boldsymbol{h}\|^{2-q} \|\boldsymbol{h}\|_{\boldsymbol{x}}^q + c_2 \cdot G_q^{\frac{2}{2-q}} \|\boldsymbol{h}\|^{\frac{4-q}{2-q}}, \tag{75}$$
>
> *with $c_1 := 2^{\frac{2(q+1)}{2-q}}$ and $c_2 := \frac{(2-q)^{\frac{4-q}{2-q}}}{2 \cdot (4-q)}$.*

Note that in view of (66), this inequality recovers up to numerical constants the bound (67) for the convex functions with Hölder third derivative, which correspond to $q := \frac{2\nu}{1+\nu}$, $G_q := L_{3,\nu}^{1/(1+\nu)}$ and $0 \le \nu \le 1$ covers the interval $0 \le q \le 1$.

It remains to establish the bound for the Gradient Normalized Smoothness $\gamma(\cdot)$. We fix an arbitrary $\boldsymbol{g} \in \mathbb{R}^n$ and $\gamma > 0$, and consider the directions $\boldsymbol{h} \in B_\gamma \cap \mathcal{O}_{\boldsymbol{x},\boldsymbol{g}}$, i.e.

$$\|\boldsymbol{h}\| \le \gamma \quad \text{and} \quad \|\boldsymbol{h}\|_{\boldsymbol{x}}^2 \le -\langle \boldsymbol{g}, \boldsymbol{h} \rangle \le \|\boldsymbol{g}\|_* \|\boldsymbol{h}\|.$$

For such $\boldsymbol{h}$, our bound (75) leads to

$$\|\nabla f(\boldsymbol{x} + \boldsymbol{h}) - \nabla f(\boldsymbol{x}) - \nabla^2 f(\boldsymbol{x})\boldsymbol{h}\|_* \le c_1 G_q \cdot \gamma^{\frac{2-q}{2}} \cdot \|\boldsymbol{g}\|_*^{\frac{q}{2}} \cdot \|\boldsymbol{h}\| + c_2 G_q^{\frac{2}{2-q}} \cdot \gamma^{\frac{2}{2-q}} \cdot \|\boldsymbol{h}\|.$$

We notice that to ensure

$$c_1 G_q \cdot \gamma^{\frac{2-q}{2}} \cdot \|\boldsymbol{g}\|_*^{\frac{q}{2}} + c_2 G_q^{\frac{2}{2-q}} \cdot \gamma^{\frac{2}{2-q}} \le \frac{\|\boldsymbol{g}\|_*}{\gamma},$$

it is sufficient to choose

$$\gamma := \left[\tfrac{1}{2}\right]^{\frac{8+2q}{(2-q)(4-q)}} \cdot \left[\frac{\|\boldsymbol{g}\|_*^{2-q}}{G_q^2}\right]^{\frac{1}{4-q}},$$

and thus we establish the following result.

> **Proposition 6.** *Let $f$ satisfy* (72) *for some $0 \leq q < 2$ and $G_q > 0$. Then,*
> $$\gamma_f(\boldsymbol{x}, \boldsymbol{g}) \;\; \geq \;\; \left[\tfrac{1}{2}\right]^{\frac{8+2q}{(2-q)(4-q)}} \cdot \left[\frac{\|\boldsymbol{g}\|_*^{2-q}}{G_q^2}\right]^{\frac{1}{4-q}}.$$

This bound generalizes that one from Proposition 2 as a particular case $0 \leq q \leq 1$. We also see that, ignoring the numerical constant and substituting formally $q := 2$ provides us with the right power that corresponds to the Quasi-Self-Concordant functions from Proposition 3.

Using this bound in our Theorem 6 with $\alpha := \frac{2-q}{4-q}$ for convex functions $(c = 0)$ immediately gives us the following complexity to find $F(\boldsymbol{x}_K) - F^* \leq \varepsilon$ of order

$$K \;\; = \;\; O\left(\left[\frac{G_q^{\frac{2}{2-q}} D_0^{\frac{6-2q}{2-q}}}{\varepsilon}\right]^{\frac{2-q}{4-q}}\right) \tag{76}$$

for our algorithm, minimizing Generalized Quasi-Self-Concordant functions (72) of degree $0 \leq q < 2$. To the best of our knowledge, this global complexity is completely new and has not been covered in the prior literature. This result recovers complexity (68) as a particular case and naturally interpolates the complexities for convex functions with Hölder continuous third derivative and Quasi-Self-Concordant functions (see also Table 1).

To illustrate the power of our results, we return to Example (9) to examine the direct consequences of our theory.

> **Example 10.** *Let $f(\boldsymbol{x}) = \frac{1}{p}\|\mathbf{A}\boldsymbol{x} - \boldsymbol{b}\|_2^p$, for some $p \geq 2$, and $\|\cdot\|_2$ is the standard Euclidean norm. Let us choose the norm in our space with $\mathbf{B} := \mathbf{A}^\top \mathbf{A}$, assuming $\mathbf{B} \succ 0$. Then, according to our previous observations, function $f$ belongs to class* (72) *with*
> $$q \;\; := \;\; \tfrac{2(p-3)}{p-2}, \qquad and \qquad G_q \;\; := \;\; (p-1)(p-2).$$
> *According to Proposition 6, the Gradient-Normalized Smoothness for this function is bounded as*
> $$\gamma_f(\boldsymbol{x}) \;\; \geq \;\; \tfrac{\|\nabla f(\boldsymbol{x})\|_*^\alpha}{M_{1-\alpha}},$$
> *with $\alpha := \frac{2-q}{4-q} = \frac{1}{p-1}$ and $M_{1-\alpha} := \left[(p-1)(p-2)2^{3p-7}\right]^{\frac{p-2}{p-1}}$. At the same time, this objective is uniformly convex* (55) *of degree $p$ with constant $\sigma_p = 2^{2-p}$ (Doikov & Nesterov, 2021). Hence, the gradient-dominance condition* (54) *is satisfied with*
> $$c \;\; := \;\; \tfrac{1}{p-1} \qquad and \qquad D_c \;\; := \;\; \tfrac{p-1}{p} \cdot 2^{\frac{p-2}{p-1}}.$$
> *Therefore, since $\alpha \equiv c$, by Theorem 6, our algorithm has the global linear rate, and the number of iterations to achieve $f(\boldsymbol{x}_K) - f^* \leq \varepsilon$ is bounded as*
> $$K \;\; = \;\; 8M_{1-\alpha}D_c \log \tfrac{f(\boldsymbol{x}_0) - f^*}{\varepsilon} \;\; = \;\; O\left(\log \tfrac{f(\boldsymbol{x}_0) - f^*}{\varepsilon}\right),$$
> *where $O(\cdot)$ hides a numerical constant that depends only on $p$.*

### G.5 $(L_0, L_1)$-SMOOTH FUNCTIONS

Let us assume that $f$ satisfies the following inequality (Zhang et al., 2019):

$$\|\nabla^2 f(\boldsymbol{x})\| \;\; \leq \;\; L_0 + L_1 \|\nabla f(\boldsymbol{x})\|_*, \qquad \boldsymbol{x} \in \mathbb{R}^n. \tag{77}$$

Then, for such functions, we have the following bound (see Lemma 2.5 in (Vankov et al., 2024)), for any $\boldsymbol{x}, \boldsymbol{h} \in \mathbb{R}^n$:

$$
\begin{aligned}
\|\nabla f(\boldsymbol{x}+\boldsymbol{h}) - \nabla f(\boldsymbol{x}) - \nabla^2 f(\boldsymbol{x})\boldsymbol{h}\|_* \quad &\leq \quad \|\nabla f(\boldsymbol{x}+\boldsymbol{h}) - \nabla f(\boldsymbol{x})\|_* + \|\nabla^2 f(\boldsymbol{x})\| \cdot \|\boldsymbol{h}\| \\
&\leq \quad \Big(L_0 + L_1\|\nabla f(\boldsymbol{x})\|_*\Big) \cdot \Big[\|\boldsymbol{h}\| + \tfrac{e^{L_1\|\boldsymbol{h}\|}-1}{L_1}\Big] \\
&\leq \quad \Big(L_0 + L_1\|\nabla f(\boldsymbol{x})\|_*\Big) \cdot \|\boldsymbol{h}\| \cdot \big(1 + e^{L_1\|\boldsymbol{h}\|}\big).
\end{aligned}
$$

We fix $\gamma > 0$, $\boldsymbol{g} \in \mathbb{R}^n$, and consider $\boldsymbol{h} \in B_\gamma$. Then, we have an upper bound

$$
\|\nabla f(\boldsymbol{x}+\boldsymbol{h}) - \nabla f(\boldsymbol{x}) - \nabla^2 f(\boldsymbol{x})\boldsymbol{h}\|_* \quad \leq \quad \frac{\|\boldsymbol{g}\|_*\|\boldsymbol{h}\|}{\gamma},
$$

as soon as

$$
\Big(L_0 + L_1\|\nabla f(\boldsymbol{x})\|_*\Big) \cdot \big(1 + e^{L_1\gamma}\big) \quad \leq \quad \frac{\|\boldsymbol{g}\|_*}{\gamma}.
$$

It it easy to check that it is satisfied for $\gamma := \frac{1}{1+\exp(\|\boldsymbol{g}\|_*/\|\nabla f(\boldsymbol{x})\|_*)} \cdot \frac{\|\boldsymbol{g}\|_*}{L_0+L_1\|\nabla f(\boldsymbol{x})\|_*}$. Therefore, we have established the following lower bound.

> **Proposition 7.** *Let $f$ satisfy (77) for some $L_0, L_1 > 0$. Then,*
>
> $$
> \gamma_f(\boldsymbol{x}, \boldsymbol{g}) \quad \geq \quad \frac{\|\boldsymbol{g}\|_*}{L_0+L_1\|\nabla f(\boldsymbol{x})\|_*} \cdot \Big(1 + \exp\big(\tfrac{\|\boldsymbol{g}\|_*}{\|\nabla f(\boldsymbol{x})\|_*}\big)\Big)^{-1},
> $$
>
> *and for $\boldsymbol{g} := \nabla f(\boldsymbol{x})$, we obtain*
>
> $$
> \gamma_f(\boldsymbol{x}) \quad \geq \quad \frac{\|\nabla f(\boldsymbol{x})\|_*}{\rho(L_0+L_1\|\nabla f(\boldsymbol{x})\|_*)},
> $$
>
> *where $\rho := 1 + e \approx 3.718$.*

Using these bounds directly in our Theorems 1 and 2, we obtain the following complexity results:

- For unconstrained minimization of a non-convex function $f(\cdot)$, to achieve $\|\nabla f(\boldsymbol{x}_K)\|_* \leq \varepsilon$ it is enough to perform

$$
K \quad = \quad F_0 \cdot O\Big(\tfrac{L_0}{\varepsilon^2} + \tfrac{L_1}{\varepsilon}\Big)
$$

  iterations of our algorithm.

- For unconstrained minimization of a convex function $f(\cdot)$, to achieve $f(\boldsymbol{x}_K) - f^\star \leq \varepsilon$, it is enough to perform

$$
K \quad = \quad O\Big(\Big[\tfrac{L_0 D_0^2}{\varepsilon} + L_1 D_0\Big] \log \tfrac{F_0}{\varepsilon}\Big)
$$

  iterations of our algorithm.

Therefore, we see that our method has a global convergence guarantee, at least as strong as that of first-order methods on $(L_0, L_1)$-smooth functions. Moreover, these convergence rates are achieved automatically, and the actual speed of the method will be the best within these problem classes.

### G.6 SECOND-ORDER $(M_0, M_1)$-SMOOTH FUNCTIONS

Following (Xie et al., 2024; Gratton et al., 2025), let us assume that $f$ satisfies the following inequality:

$$
\|\nabla^2 f(\boldsymbol{x}) - \nabla^2 f(\boldsymbol{y})\| \quad \leq \quad (M_0 + M_1\|\nabla f(\boldsymbol{x})\|_*)\|\boldsymbol{x} - \boldsymbol{y}\|, \qquad \boldsymbol{x}, \boldsymbol{y} \in \mathbb{R}^n, \qquad (78)
$$

for some constants $M_0, M_1 \geq 0$. Then, we have the bound, for all $\boldsymbol{h} \in \mathbb{R}^n$

$$
\|\nabla f(\boldsymbol{x}+\boldsymbol{h}) - \nabla f(\boldsymbol{x}) - \nabla^2 f(\boldsymbol{x})\boldsymbol{h}\|_* \quad \leq \quad \tfrac{M_0+M_1\|\nabla f(\boldsymbol{x})\|_*}{2}\|\boldsymbol{h}\|^2.
$$

Restricting our direction onto a ball, $\boldsymbol{h} \in B_\gamma$, we have that

$$
\|\nabla f(\boldsymbol{x}+\boldsymbol{h}) - \nabla f(\boldsymbol{x}) - \nabla^2 f(\boldsymbol{x})\boldsymbol{h}\|_* \quad \leq \quad \gamma \cdot \tfrac{M_0+M_1\|\nabla f(\boldsymbol{x})\|_*}{2}\|\boldsymbol{h}\| \quad = \quad \frac{\|\boldsymbol{g}\|_*\|\boldsymbol{h}\|}{\gamma},
$$

where the last equation holds for the particular choice $\gamma := \sqrt{\frac{2\|\boldsymbol{g}\|_*}{M_0+M_1\|\nabla f(\boldsymbol{x})\|_*}}$. Therefore, we obtain the following statement.

> **Proposition 8.** *Let $f$ satisfy (78) for some $M_0, M_1 > 0$. Then,*
> $$\gamma_f(\boldsymbol{x}, \boldsymbol{g}) \ \geq \ \left( \frac{2\|\boldsymbol{g}\|_*}{M_0 + M_1 \|\nabla f(\boldsymbol{x})\|_*} \right)^{1/2}.$$

Using this bound in our Theorems 1 and 2, we obtain:

- For unconstrained minimization, in the non-convex case, to achieve $\|\nabla f(\boldsymbol{x}_K)\|_* \leq \varepsilon$ it is enough to perform
$$K \ = \ F_0 \cdot O\left( \frac{M_0^{1/2}}{\varepsilon^{3/2}} + \frac{M_1^{1/2}}{\varepsilon} \right)$$
  iterations of our algorithm.

- For unconstrained minimization, in the convex case, to achieve $f(\boldsymbol{x}_K) - f^\star \leq \varepsilon$, it is enough to perform
$$K \ = \ O\left( \left[ \left( \frac{M_0 D_0^3}{\varepsilon} \right)^{1/2} + M_1^{1/2} D_0 \right] \log \frac{F_0}{\varepsilon} \right)$$
  iterations of our algorithm.

We see that a stronger second-order $(M_0, M_1)$-smoothness condition allows for improved complexity results compared to first-order $(L_0, L_1)$-smooth functions. It is important that all these problem classes are covered by our framework, which also allows for inexact Hessians.

## H   BOUNDS ON EFFECTIVE HESSIAN APPROXIMATIONS

### H.1   SOFT MAXIMUM

> **Example 11** (Soft Maximum: Extended). *In applications with multiclass classification, graph problems, and matrix games, we have*
> $$f(\boldsymbol{x}) \ := \ s(\boldsymbol{u}(\boldsymbol{x})),$$
> *where $\boldsymbol{u} : \mathbb{R}^n \to \mathbb{R}^d$ is an operator (e.g. a linear or nonlinear model), and $s(\boldsymbol{y}) := \log \sum_{i=1}^d e^{y_i}$ is the LogSumExp loss. Note that $s(\cdot)$ is Quasi-Self-Concordant (Section G.3), and its gradient is the softmax: $[\nabla s(\boldsymbol{y})]_i = e^{y_i} \cdot \left( \sum_{j=1}^d e^{y_j} \right)^{-1}$. Assume that*
> $$\|\nabla \boldsymbol{u}(\boldsymbol{x})\| \ \leq \ \xi_0, \qquad \|\nabla^2 \boldsymbol{u}(\boldsymbol{x})\| \ \leq \ \xi_1, \qquad \boldsymbol{x} \in \mathbb{R}^n,$$
> *for some $\xi_0, \xi_1 \geq 0$, and that operator $\boldsymbol{u}(\cdot)$ is non-degenerate [4], for some $\mu > 0$:*
> $$\nabla \boldsymbol{u}(\boldsymbol{x}) \mathbf{B}^{-1} \nabla \boldsymbol{u}(\boldsymbol{x})^\top \ \succeq \ \mu \mathbf{I}_d, \qquad \boldsymbol{x} \in \mathbb{R}^n. \tag{79}$$
> *We introduce the following approximations and derive corresponding bounds:*
>
> • *If $\mathbf{H}(\boldsymbol{x}) := \nabla \boldsymbol{u}(\boldsymbol{x})^\top \nabla^2 s(\boldsymbol{u}(\boldsymbol{x})) \nabla \boldsymbol{u}(\boldsymbol{x}) \succeq \mathbf{0}$, we have*
> $$\|\nabla^2 f(\boldsymbol{x}) - \mathbf{H}(\boldsymbol{x})\| \ \leq \ \frac{\xi_1}{\sqrt{\mu}} \|\nabla f(\boldsymbol{x})\|_*.$$
>
> • *If $\mathbf{H}(\boldsymbol{x}) := \nabla \boldsymbol{u}(\boldsymbol{x})^\top \nabla \boldsymbol{u}(\boldsymbol{x}) \succeq \mathbf{0}$  (Gauss-Newton), we have*
> $$\|\nabla^2 f(\boldsymbol{x}) - \mathbf{H}(\boldsymbol{x})\| \ \leq \ \xi_0^2 + \left( \xi_0 + \frac{\xi_1}{\sqrt{\mu}} \right) \|\nabla f(\boldsymbol{x})\|_*.$$
>
> • *If $\mathbf{H}(\boldsymbol{x}) := \nabla \boldsymbol{u}(\boldsymbol{x})^\top \mathrm{Diag}(\nabla s(\boldsymbol{u}(\boldsymbol{x}))) \nabla \boldsymbol{u}(\boldsymbol{x}) \succeq \mathbf{0}$  (Weighted Gauss-Newton), we have*
> $$\|\nabla^2 f(\boldsymbol{x}) - \mathbf{H}(\boldsymbol{x})\| \ \leq \ \left( \xi_0 + \frac{\xi_1}{\sqrt{\mu}} \right) \|\nabla f(\boldsymbol{x})\|_*.$$

---

[4]This assumption can be relaxed. It holds, for example, when the model is overparametrized (i.e. $n \gg d$).

*Proof.* Note that
$$\nabla f(\boldsymbol{x}) \;=\; \nabla \boldsymbol{u}(\boldsymbol{x})^{\top} \nabla s\left(\boldsymbol{u}(\boldsymbol{x})\right),$$
where $\nabla \boldsymbol{u}(\boldsymbol{x}) \in \mathbb{R}^{d \times n}$ denotes the Jacobian of mapping $\boldsymbol{u}$, and

$$
\begin{aligned}
\nabla^2 f(\boldsymbol{x}) \;&=\; \nabla \boldsymbol{u}(\boldsymbol{x})^{\top} \nabla^2 s\left(\boldsymbol{u}(\boldsymbol{x})\right) \nabla \boldsymbol{u}(\boldsymbol{x}) \;+\; \sum_{i=1}^{d} \left[\nabla s\left(\boldsymbol{u}(\boldsymbol{x})\right)\right]_i \nabla^2 u_i(\boldsymbol{x}) \\
&=\; \nabla \boldsymbol{u}(\boldsymbol{x})^{\top} \left( \operatorname{Diag}\left(\nabla s\left(\boldsymbol{u}(\boldsymbol{x})\right)\right) - \nabla s\left(\boldsymbol{u}(\boldsymbol{x})\right) \nabla s\left(\boldsymbol{u}(\boldsymbol{x})\right)^{\top} \right) \nabla \boldsymbol{u}(\boldsymbol{x}) \\
&\quad + \sum_{i=1}^{d} \left[\nabla s\left(\boldsymbol{u}(\boldsymbol{x})\right)\right]_i \nabla^2 u_i(\boldsymbol{x}),
\end{aligned}
$$

where $\nabla^2 \boldsymbol{u}(\boldsymbol{x})$ is the tensor of second derivatives or $\boldsymbol{u}$, which we assume to be bounded.

1. Consider the approximation
$$\mathbf{H}(\boldsymbol{x}) \;:=\; \nabla \boldsymbol{u}(\boldsymbol{x})^{\top} \nabla^2 (s\left(\boldsymbol{u}(\boldsymbol{x})\right) \nabla \boldsymbol{u}(\boldsymbol{x}) \;\succeq\; \mathbf{0}.$$
Note that when operator $\boldsymbol{u}(\cdot)$ is linear, $\mathbf{H}(\boldsymbol{x})$ is the exact Hessian. In general non-linear case, we can bound

$$
\begin{aligned}
\|\nabla^2 f(\boldsymbol{x}) - \mathbf{H}(\boldsymbol{x})\| \;&=\; \left\| \sum_{i=1}^{d} \left[\nabla s\left(\boldsymbol{u}(\boldsymbol{x})\right)\right]_i \nabla^2 u_i(\boldsymbol{x}) \right\| \\
&\leq\; \xi_1 \|\nabla s(\boldsymbol{u}(\boldsymbol{x}))\|.
\end{aligned}
$$

On the other hand,

$$
\begin{aligned}
\|\nabla f(\boldsymbol{x})\|_*^2 \;&:=\; \langle \nabla f(\boldsymbol{x}), \mathbf{B}^{-1} \nabla f(\boldsymbol{x}) \rangle \;=\; \langle \nabla \boldsymbol{u}(\boldsymbol{x}) \mathbf{B}^{-1} \nabla \boldsymbol{u}(\boldsymbol{x})^{\top} \nabla s\left(\boldsymbol{u}(\boldsymbol{x})\right), \nabla s\left(\boldsymbol{u}(\boldsymbol{x})\right) \rangle \\
&\geq\; \mu \|\nabla s(\boldsymbol{u}(\boldsymbol{x}))\|^2,
\end{aligned}
$$

where in the last inequality we used the non-degeneracy condition and the standard Euclidean norm in $\mathbb{R}^d$. Thus, we have the following bound $\|\nabla s(\boldsymbol{u}(\boldsymbol{x}))\| \leq \frac{1}{\sqrt{\mu}} \|\nabla f(\boldsymbol{x})\|_*$, that yields

$$\|\nabla^2 f(\boldsymbol{x}) - \mathbf{H}(\boldsymbol{x})\| \;\leq\; \frac{\xi_1}{\sqrt{\mu}} \|\nabla f(\boldsymbol{x})\|_*.$$

2. Consider the approximation
$$\mathbf{H}(\boldsymbol{x}) \;:=\; \nabla \boldsymbol{u}(\boldsymbol{x})^{\top} \nabla \boldsymbol{u}(\boldsymbol{x}) \;\succeq\; \mathbf{0}.$$
Then, as in the previous case, we have:

$$
\begin{aligned}
\|\nabla^2 f(\boldsymbol{x}) - \mathbf{H}(\boldsymbol{x})\| \;&=\; \left\| \nabla \boldsymbol{u}(\boldsymbol{x})^{\top} \left( \nabla^2 s\left(\boldsymbol{u}(\boldsymbol{x})\right) - \mathbf{I}_d \right) \nabla \boldsymbol{u}(\boldsymbol{x}) + \sum_{i=1}^{d} \left[\nabla s\left(\boldsymbol{u}(\boldsymbol{x})\right)\right]_i \nabla^2 u_i(\boldsymbol{x}) \right\| \\
&\leq\; \left\| \nabla \boldsymbol{u}(\boldsymbol{x})^{\top} \left( \nabla^2 s(\boldsymbol{u}(\boldsymbol{x})) - \mathbf{I}_d \right) \nabla \boldsymbol{u}(\boldsymbol{x}) \right\| + \frac{\xi_1}{\sqrt{\mu}} \|\nabla f(\boldsymbol{x})\|_*,
\end{aligned}
$$

and it remains to bound the following term:

$$
\begin{aligned}
&\left\| \nabla \boldsymbol{u}(\boldsymbol{x})^{\top} \left( \nabla^2 s(\boldsymbol{u}(\boldsymbol{x})) - \mathbf{I}_d \right) \nabla \boldsymbol{u}(\boldsymbol{x}) \right\| \\
&= \left\| \nabla \boldsymbol{u}(\boldsymbol{x})^{\top} \left[ \operatorname{Diag}\left(\nabla s(\boldsymbol{u}(\boldsymbol{x}))\right) - \mathbf{I}_d \right] \nabla \boldsymbol{u}(\boldsymbol{x}) - \nabla \boldsymbol{u}(\boldsymbol{x})^{\top} \nabla s\left(\boldsymbol{u}(\boldsymbol{x})\right) \nabla s\left(\boldsymbol{u}(\boldsymbol{x})\right)^{\top} \nabla \boldsymbol{u}(\boldsymbol{x}) \right\| \\
&\leq \left\| \nabla \boldsymbol{u}(\boldsymbol{x})^{\top} \left[ \operatorname{Diag}\left(\nabla s(\boldsymbol{u}(\boldsymbol{x}))\right) - \mathbf{I}_d \right] \nabla \boldsymbol{u}(\boldsymbol{x}) \right\| + \left\| \nabla f(\boldsymbol{x})^{\top} \nabla f(\boldsymbol{x}) \right\|.
\end{aligned}
$$

The first term can be bounded as follows:

$$\left\| \nabla \boldsymbol{u}(\boldsymbol{x})^{\top} \left[ \operatorname{Diag}\left(\nabla s(\boldsymbol{u}(\boldsymbol{x}))\right) - \mathbf{I}_d \right] \nabla \boldsymbol{u}(\boldsymbol{x}) \right\| \;\leq\; \|\nabla \boldsymbol{u}(\boldsymbol{x})\|^2 \left\| \operatorname{Diag}\left(\nabla s\left(\boldsymbol{u}(\boldsymbol{x})\right)\right) - \mathbf{I}_d \right\| \;\leq\; \xi_0^2,$$

where we used the fact that $\max_{1 \leq i \leq d} \left| \left[\nabla s\left(\boldsymbol{u}(\boldsymbol{x})\right)\right]_i - 1 \right| \leq 1$ and our assumption regarding the boundedness of $\|\nabla \boldsymbol{u}(\boldsymbol{x})\|$. For the second term, we notice that

$$\|\nabla f(\boldsymbol{x})\|_* \;\leq\; \|\nabla \boldsymbol{u}(\boldsymbol{x})\| \cdot \|\nabla s(\boldsymbol{u}(\boldsymbol{x}))\| \;\leq\; \xi_0,$$

since $\nabla s(\boldsymbol{u}(\boldsymbol{x}))$ is from the simplex. Hence,

$$\|\nabla f(\boldsymbol{x})^\top \nabla f(\boldsymbol{x})\| \;=\; \|\nabla f(\boldsymbol{x})\|_*^2 \;\leq\; \xi_0 \|\nabla f(\boldsymbol{x})\|_*,$$

and we finally obtain the following bound:

$$\|\nabla^2 f(\boldsymbol{x}) - \mathbf{H}(\boldsymbol{x})\| \;\leq\; \xi_0^2 + \left(\xi_0 + \tfrac{\xi_1}{\sqrt{\mu}}\right) \|\nabla f(\boldsymbol{x})\|_*.$$

3. Consider the approximation

$$\mathbf{H}(\boldsymbol{x}) \;\coloneqq\; \nabla \boldsymbol{u}(\boldsymbol{x})^\top \operatorname{Diag}\left(\nabla s\left(\boldsymbol{u}(\boldsymbol{x})\right)\right) \nabla \boldsymbol{u}(\boldsymbol{x}) \;\succeq\; \mathbf{0}.$$

Repeating the reasoning from the previous case, it follows immediately that

$$\|\nabla^2 f(\boldsymbol{x}) - \mathbf{H}(\boldsymbol{x})\| \;\leq\; \left(\xi_0 + \tfrac{\xi_1}{\sqrt{\mu}}\right) \|\nabla f(\boldsymbol{x})\|_*,$$

which is the required bound. $\qquad\square$

## H.2 NONLINEAR EQUATIONS

**Example 12** (Nonlinear Equations: Extended). *Let $\boldsymbol{u} : \mathbb{R}^n \to \mathbb{R}^d$ be a nonlinear operator, and set*

$$f(\boldsymbol{x}) \;\coloneqq\; \tfrac{1}{p}\|\boldsymbol{u}(\boldsymbol{x})\|^p = \tfrac{1}{p}\langle \mathbf{G}\boldsymbol{u}(\boldsymbol{x}), \boldsymbol{u}(\boldsymbol{x})\rangle^{\frac{p}{2}},$$

*for some $\mathbf{G} = \mathbf{G}^\top \succ \mathbf{0}$, and $p \geq 2$. Note that $\tfrac{1}{p}\|\cdot\|^p$ is a generalized self-concordant loss function (Section G.4). As in the previous example, we assume that*

$$\|\nabla \boldsymbol{u}(\boldsymbol{x})\| \;\leq\; \xi_0, \qquad \|\nabla^2 \boldsymbol{u}(\boldsymbol{x})\| \;\leq\; \xi_1, \qquad \boldsymbol{x} \in \mathbb{R}^n,$$

*for some $\xi_0, \xi_1 \geq 0$, and that the operator is non-degenerate, for some $\mu > 0$:*

$$\nabla \boldsymbol{u}(\boldsymbol{x})\mathbf{B}^{-1}\nabla \boldsymbol{u}(\boldsymbol{x})^\top \;\succeq\; \mu \mathbf{G}^{-1}, \qquad \boldsymbol{x} \in \mathbb{R}^n. \tag{80}$$

*We introduce the following approximations and derive corresponding bounds:*

- *If $\mathbf{H}(\boldsymbol{x}) \coloneqq \|\boldsymbol{u}(\boldsymbol{x})\|^{p-2}\nabla \boldsymbol{u}(\boldsymbol{x})^\top \mathbf{G}\nabla \boldsymbol{u}(\boldsymbol{x}) + \tfrac{p-2}{\|\boldsymbol{u}(\boldsymbol{x})\|^p}\nabla f(\boldsymbol{x})\nabla f(\boldsymbol{x})^\top \succeq \mathbf{0}$, we have*

$$\|\nabla^2 f(\boldsymbol{x}) - \mathbf{H}(\boldsymbol{x})\| \;\leq\; \tfrac{\xi_1}{\sqrt{\mu}}\|\nabla f(\boldsymbol{x})\|_*.$$

- *If $\mathbf{H}(\boldsymbol{x}) \coloneqq \|\boldsymbol{u}(\boldsymbol{x})\|^{p-2}\nabla \boldsymbol{u}(\boldsymbol{x})^\top \mathbf{G}\nabla \boldsymbol{u}(\boldsymbol{x}) \succeq \mathbf{0}$, we have*

$$\|\nabla^2 f(\boldsymbol{x}) - \mathbf{H}(\boldsymbol{x})\| \;\leq\; (p-2)\xi_0^{\frac{p}{p-1}}\|\nabla f(\boldsymbol{x})\|_*^{\frac{p-2}{p-1}} + \tfrac{\xi_1}{\sqrt{\mu}}\|\nabla f(\boldsymbol{x})\|_*.$$

- *If $\mathbf{H}(\boldsymbol{x}) \coloneqq \tfrac{p-2}{\|\boldsymbol{u}(\boldsymbol{x})\|^p}\nabla f(\boldsymbol{x})\nabla f(\boldsymbol{x})^\top \succeq \mathbf{0}$  (Fisher-type), we have*

$$\|\nabla^2 f(\boldsymbol{x}) - \mathbf{H}(\boldsymbol{x})\| \;\leq\; \xi_0^2 \mu^{\frac{2-p}{2(p-1)}}\|\nabla f(\boldsymbol{x})\|_*^{\frac{p-2}{p-1}} + \tfrac{\xi_1}{\sqrt{\mu}}\|\nabla f(\boldsymbol{x})\|_*.$$

*Proof.* Note that

$$\nabla f(\boldsymbol{x}) \;=\; \|\boldsymbol{u}(\boldsymbol{x})\|^{p-2}\nabla \boldsymbol{u}(\boldsymbol{x})^\top \mathbf{G}\boldsymbol{u}(\boldsymbol{x}),$$

where $\nabla \boldsymbol{u}(\boldsymbol{x}) \in \mathbb{R}^{d\times n}$ denotes the Jacobian matrix of mapping $\boldsymbol{u}$, and, for any direction $\boldsymbol{h} \in \mathbb{R}^n$, we have

$$\langle \nabla^2 f(\boldsymbol{x})\boldsymbol{h}, \boldsymbol{h}\rangle \;=\; \|\boldsymbol{u}(\boldsymbol{x})\|^{p-2}\langle \mathbf{G}\nabla \boldsymbol{u}(\boldsymbol{x})\boldsymbol{h}, \nabla \boldsymbol{u}(\boldsymbol{x})\boldsymbol{h}\rangle + \|\boldsymbol{u}(\boldsymbol{x})\|^{p-2}\langle \mathbf{G}\boldsymbol{u}(\boldsymbol{x}), \nabla^2 \boldsymbol{u}(\boldsymbol{x})[\boldsymbol{h}, \boldsymbol{h}]\rangle$$
$$+ (p-2)\|\boldsymbol{u}(\boldsymbol{x})\|^{p-4}\langle \mathbf{G}\boldsymbol{u}(\boldsymbol{x}), \nabla \boldsymbol{u}(\boldsymbol{x})\boldsymbol{h}\rangle^2,$$

where $\nabla^2 \boldsymbol{u}(\boldsymbol{x})$ is the tensor of second derivatives of $\boldsymbol{u}$, which we assume to be bounded.

1. Consider the approximation

$$\mathbf{H}(\boldsymbol{x}) \;\coloneqq\; \|\boldsymbol{u}(\boldsymbol{x})\|^{p-2}\nabla \boldsymbol{u}(\boldsymbol{x})^\top \mathbf{G}\nabla \boldsymbol{u}(\boldsymbol{x}) + \tfrac{p-2}{\|\boldsymbol{u}(\boldsymbol{x})\|^p}\nabla f(\boldsymbol{x})\nabla f(\boldsymbol{x})^\top \;\succeq\; \mathbf{0}. \tag{81}$$

Note that it resembles a combination of the Gauss-Newton and Fisher approximation matrices, and for $p = 2$ it gives the classic Gauss-Newton approximation. Moreover, when the operator $\boldsymbol{u}$ is linear, the problem is convex, and $\xi_1 = 0$. Thus (81) gives us exact Hessian in this case. Let us consider

$$\left|\langle \nabla^2 f(\boldsymbol{x})\boldsymbol{h}, \boldsymbol{h}\rangle - \langle \mathbf{H}(\boldsymbol{x})\boldsymbol{h}, \boldsymbol{h}\rangle\right| = \|\boldsymbol{u}(\boldsymbol{x})\|^{p-2}\left|\langle \mathbf{G}\boldsymbol{u}(\boldsymbol{x}), \nabla^2\boldsymbol{u}(\boldsymbol{x})[\boldsymbol{h}, \boldsymbol{h}]\rangle\right|$$

$$\leq \|\boldsymbol{u}(\boldsymbol{x})\|^{p-2}\|\boldsymbol{u}(\boldsymbol{x})\|\|\nabla^2\boldsymbol{u}(\boldsymbol{x})\|\|\boldsymbol{h}\|^2,$$

therefore

$$\|\nabla^2 f(\boldsymbol{x}) - \mathbf{H}(\boldsymbol{x})\| := \max_{\boldsymbol{h}:\|\boldsymbol{h}\|=1}\left|\langle\left(\nabla^2 f(\boldsymbol{x}) - \mathbf{H}(\boldsymbol{x})\right)\boldsymbol{h}, \boldsymbol{h}\rangle\right|$$

$$\leq \xi_1\|\boldsymbol{u}(\boldsymbol{x})\|^{p-1} = \xi_1\left(pf(\boldsymbol{x})\right)^{\frac{p-1}{p}}.$$

Using our non-degeneracy condition, we can further bound

$$\|\nabla f(\boldsymbol{x})\|_*^2 = \|\boldsymbol{u}(\boldsymbol{x})\|^{2(p-2)}\|\nabla\boldsymbol{u}(\boldsymbol{x})^\top\mathbf{G}\boldsymbol{u}(\boldsymbol{x})\|_*^2$$

$$= \|\boldsymbol{u}(\boldsymbol{x})\|^{2(p-2)}\left\langle\mathbf{G}\boldsymbol{u}(\boldsymbol{x}), \nabla\boldsymbol{u}(\boldsymbol{x})\mathbf{B}^{-1}\nabla\boldsymbol{u}(\boldsymbol{x})^\top\mathbf{G}\boldsymbol{u}(\boldsymbol{x})\right\rangle$$

$$\overset{(80)}{\geq} \mu\|\boldsymbol{u}(\boldsymbol{x})\|^{2(p-1)}.$$

Thus, we have $\|\boldsymbol{u}(\boldsymbol{x})\|^{p-1} \leq \frac{1}{\sqrt{\mu}}\|\nabla f(\boldsymbol{x})\|_*$, which gives us the following bound on the approximation error:

$$\|\nabla^2 f(\boldsymbol{x}) - \mathbf{H}(\boldsymbol{x})\| \leq \frac{\xi_1}{\sqrt{\mu}}\|\nabla f(\boldsymbol{x})\|_*.$$

2. Consider the approximation

$$\mathbf{H}(\boldsymbol{x}) := \|\boldsymbol{u}(\boldsymbol{x})\|^{p-2}\nabla\boldsymbol{u}(\boldsymbol{x})^\top\mathbf{G}\nabla\boldsymbol{u}(\boldsymbol{x}) \succeq 0.$$

Using observations from the previous step,

$$\|\nabla^2 f(\boldsymbol{x}) - \mathbf{H}(\boldsymbol{x})\| := \max_{\boldsymbol{h}:\|\boldsymbol{h}\|=1}\left|\langle\left(\nabla^2 f(\boldsymbol{x}) - \mathbf{H}(\boldsymbol{x})\right)\boldsymbol{h}, \boldsymbol{h}\rangle\right|$$

$$\leq (p-2)\|\boldsymbol{u}(\boldsymbol{x})\|^{p-4}\max_{\boldsymbol{h}:\|\boldsymbol{h}\|=1}\langle\mathbf{G}\boldsymbol{u}(\boldsymbol{x}), \nabla\boldsymbol{u}(\boldsymbol{x})\boldsymbol{h}\rangle^2$$

$$+ \|\boldsymbol{u}(\boldsymbol{x})\|^{p-2}\max_{\boldsymbol{h}:\|\boldsymbol{h}\|=1}\left|\langle\mathbf{G}\boldsymbol{u}(\boldsymbol{x}), \nabla^2\boldsymbol{u}(\boldsymbol{x})[\boldsymbol{h}, \boldsymbol{h}]\rangle\right|$$

$$\leq (p-2)\|\boldsymbol{u}(\boldsymbol{x})\|^{p-4}\max_{\boldsymbol{h}:\|\boldsymbol{h}\|=1}\langle\mathbf{G}\boldsymbol{u}(\boldsymbol{x}), \nabla\boldsymbol{u}(\boldsymbol{x})\boldsymbol{h}\rangle^2 + \frac{\xi_1}{\sqrt{\mu}}\|\nabla f(\boldsymbol{x})\|_*.$$

It remains to notice that, for $\|\boldsymbol{h}\| = 1$, we have:

$$\|\boldsymbol{u}(\boldsymbol{x})\|^{p-4}\langle\mathbf{G}\boldsymbol{u}(\boldsymbol{x}), \nabla\boldsymbol{u}(\boldsymbol{x})\boldsymbol{h}\rangle^2 = \|\boldsymbol{u}(\boldsymbol{x})\|^{p-4}|\langle\mathbf{G}\boldsymbol{u}(\boldsymbol{x}), \nabla\boldsymbol{u}(\boldsymbol{x})\boldsymbol{h}\rangle|^{\frac{p}{p-1}}|\langle\mathbf{G}\boldsymbol{u}(\boldsymbol{x}), \nabla\boldsymbol{u}(\boldsymbol{x})\boldsymbol{h}\rangle|^{\frac{p-2}{p-1}}$$

$$= \frac{1}{\|\boldsymbol{u}(\boldsymbol{x})\|^{\frac{p}{p-1}}}|\langle\mathbf{G}\boldsymbol{u}(\boldsymbol{x}), \nabla\boldsymbol{u}(\boldsymbol{x})\boldsymbol{h}\rangle|^{\frac{p}{p-1}}|\langle\nabla f(\boldsymbol{x}), \boldsymbol{h}\rangle|^{\frac{p-2}{p-1}}$$

$$\leq \xi_0^{\frac{p}{p-1}}\|\nabla f(\boldsymbol{x})\|_*^{\frac{p-2}{p-1}},$$

which gives the desired bound.

3. Consider the approximation

$$\mathbf{H}(\boldsymbol{x}) := \frac{p-2}{\|\boldsymbol{u}(\boldsymbol{x})\|^p}\nabla f(\boldsymbol{x})\nabla f(\boldsymbol{x})^\top \succeq 0.$$

Note this matrix can be equivalently represented as

$$\mathbf{H}(\boldsymbol{x}) = (p-2)\|\boldsymbol{u}(\boldsymbol{x})\|^{p-4}\nabla\boldsymbol{u}(\boldsymbol{x})^\top\mathbf{G}\boldsymbol{u}(\boldsymbol{x})\boldsymbol{u}(\boldsymbol{x})^\top\mathbf{G}\nabla\boldsymbol{u}(\boldsymbol{x}).$$

Therefore, we have

$$\|\nabla^2 f(\boldsymbol{x}) - \mathbf{H}(\boldsymbol{x})\| \leq \max_{\boldsymbol{h}:\|\boldsymbol{h}\|=1} \|\boldsymbol{u}(\boldsymbol{x})\|^{p-2} \langle \mathbf{G}\nabla \boldsymbol{u}(\boldsymbol{x})\boldsymbol{h}, \nabla \boldsymbol{u}(\boldsymbol{x})\boldsymbol{h}\rangle + \frac{\xi_1}{\sqrt{\mu}}\|\nabla f(\boldsymbol{x})\|_*$$

$$\leq \xi_0^2 \|\boldsymbol{u}(\boldsymbol{x})\|^{p-2} + \frac{\xi_1}{\sqrt{\mu}}\|\nabla f(\boldsymbol{x})\|_*$$

$$\leq \xi_0^2 \mu^{\frac{2-p}{2(p-1)}}\|\nabla f(\boldsymbol{x})\|_*^{\frac{p-2}{p-1}} + \frac{\xi_1}{\sqrt{\mu}}\|\nabla f(\boldsymbol{x})\|_*.$$

$\square$

## H.3 SEPARABLE OPTIMIZATION

**Example 13** (Separable Optimization: Extended). *Consider the following structure of the objective,*

$$f(\boldsymbol{x}) := \sum_{i=1}^{d} f_i(\boldsymbol{x}),$$

*where $f_i(\boldsymbol{x}) := \ell(u_i(\boldsymbol{x}))$, for a convex nonnegative loss function $\ell$ and mappings $u_i : \mathbb{R}^n \to \mathbb{R}$. Consider logistic regression, $\ell(t) := \log(1 + \exp(t))$, and the following Fisher-type Hessian approximation:*

$$\mathbf{H}(\boldsymbol{x}) := \sum_{i=1}^{d} \nabla f_i(\boldsymbol{x})\nabla f_i(\boldsymbol{x})^\top \succeq \mathbf{0},$$

*• Let each $u_i$ be a nonlinear mapping, and $f$ be a gradient-dominated (54) function. Assume that, for some $\xi_0, \xi_1 \geq 0$: $\|\nabla u_i(\boldsymbol{x})\| \leq \xi_0$, $\|\nabla^2 u_i(\boldsymbol{x})\| \leq \xi_1$, $\forall 1 \leq i \leq d$. Then, we have*

$$\|\nabla^2 f(\boldsymbol{x}) - \mathbf{H}(\boldsymbol{x})\| \leq (\xi_0^2 + \xi_1)(f^\star + D_c\|\nabla f(\boldsymbol{x})\|_*^{1+c}).$$

*• If the mappings $u_i(\boldsymbol{x}) := \langle \boldsymbol{a}_i, \boldsymbol{x}\rangle - b_i$ are linear models, then, by setting $\mathbf{B} := \sum_{i=1}^{n} \boldsymbol{a}_i \boldsymbol{a}_i^\top$, we have*

$$\|\nabla^2 f(\boldsymbol{x}) - \mathbf{H}(\boldsymbol{x})\| \leq f(\boldsymbol{x}) \leq f^\star + D\|\nabla f(\boldsymbol{x})\|,$$

*for $\boldsymbol{x} \in \mathcal{F}_0$.*

*Proof.* Note that

$$\nabla f(\boldsymbol{x}) = \sum_{i=1}^{d} \nabla f_i(\boldsymbol{x}) = \sum_{i=1}^{d} \ell'(u_i(\boldsymbol{x}))\nabla u_i(\boldsymbol{x}),$$

and

$$\nabla^2 f(\boldsymbol{x}) = \sum_{i=1}^{d} \left[\ell''(u_i(\boldsymbol{x}))\nabla u_i(\boldsymbol{x})\nabla u_i(\boldsymbol{x})^\top + \ell'(u_i(\boldsymbol{x}))\nabla^2 u_i(\boldsymbol{x})\right].$$

Consider approximation

$$\mathbf{H}(\boldsymbol{x}) := \sum_{i=1}^{d} \nabla f_i(\boldsymbol{x})\nabla f_i(\boldsymbol{x})^\top = \sum_{i=1}^{d} \ell'(u_i(\boldsymbol{x}))^2 \nabla u_i(\boldsymbol{x})\nabla u_i(\boldsymbol{x})^\top \succeq \mathbf{0}.$$

Then, we have

$$\|\nabla^2 f(\boldsymbol{x}) - \mathbf{H}(\boldsymbol{x})\|$$

$$= \left\|\sum_{i=1}^{d} \left[\left(\ell''(u_i(\boldsymbol{x})) - \ell'(u_i(\boldsymbol{x}))^2\right)\nabla u_i(\boldsymbol{x})\nabla u_i(\boldsymbol{x})^\top + \ell'(u_i(\boldsymbol{x}))\nabla^2 u_i(\boldsymbol{x})\right]\right\|$$

$$= \left\|\sum_{i=1}^{d} \left[\ell'(u_i(\boldsymbol{x}))(1 - 2\ell'(u_i(\boldsymbol{x})))\nabla u_i(\boldsymbol{x})\nabla u_i(\boldsymbol{x})^\top + \ell'(u_i(\boldsymbol{x}))\nabla^2 u_i(\boldsymbol{x})\right]\right\|$$

$$\leq \sum_{i=1}^{d} \left[\ell'(u_i(\boldsymbol{x}))|1 - 2\ell'(u_i(\boldsymbol{x}))|\|\nabla u_i(\boldsymbol{x})\|^2\right] + \sum_{i=1}^{d} \ell'(u_i(\boldsymbol{x}))\|\nabla^2 u_i(\boldsymbol{x})\|,$$

where we used that $\ell''(t) = \ell'(t) \cdot (1 - \ell'(t))$ and $\|\nabla u_i(\boldsymbol{x})\nabla u_i(\boldsymbol{x})^\top\| = \|\nabla u_i(\boldsymbol{x})\|^2$.

Applying our bounds on $\|\nabla u_i(\boldsymbol{x})\|$ and $\|\nabla^2 u_i(\boldsymbol{x})\|$ for any $i$, and using the fact that $|1 - 2\ell'(t)| < 1$ for any $t$, we have

$$\|\nabla^2 f(\boldsymbol{x}) - \mathbf{H}(\boldsymbol{x})\| \leq (\xi_0^2 + \xi_1) \sum_{i=1}^{d} \ell'(u_i(\boldsymbol{x})) \leq (\xi_0^2 + \xi_1) f(\boldsymbol{x}),$$

where in the last inequality we used that $\ell'(t) < \ell(t)$ for all $t$. Now, consider two important cases.

1. Let $u_i(\boldsymbol{x})$ be non-linear mappings and let $f(\boldsymbol{x})$ be gradient-dominated, i.e., condition (54) holds. Then, we have the bound:

$$\|\nabla^2 f(\boldsymbol{x}) - \mathbf{H}(\boldsymbol{x})\| \leq (\xi_0^2 + \xi_1) f(\boldsymbol{x}) \leq (\xi_0^2 + \xi_1) \left[f^\star + D_c \|\nabla f(\boldsymbol{x})\|_*^{1+c}\right], \quad 0 \leq c \leq 1.$$

2. Another important case is when $u_i := \langle \boldsymbol{a}_i, \boldsymbol{x} \rangle - b_i$ are linear models, where $\{\boldsymbol{a}_i, b_i\}_{i=1}^{d}$ are given data. Note that in this case, the *Gauss-Newton matrix* is constant, and we can set

$$\mathbf{B} := \sum_{i=1}^{d} \nabla u_i(\boldsymbol{x})\nabla u_i(\boldsymbol{x})^\top = \sum_{i=1}^{d} \boldsymbol{a}_i \boldsymbol{a}_i^\top \succeq \mathbf{0},$$

and it is natural to use it as our choice of the Euclidean norm. At the same time, our *Fisher approximation* becomes

$$\mathbf{H}(\boldsymbol{x}) := \sum_{i=1}^{d} \nabla f_i(\boldsymbol{x})\nabla f_i(\boldsymbol{x})^\top = \sum_{i=1}^{d} \ell'(u_i(\boldsymbol{x}))^2 \boldsymbol{a}_i \boldsymbol{a}_i^\top \succeq \mathbf{0}.$$

Therefore, we result in bound

$$\|\nabla^2 f(\boldsymbol{x}) - \mathbf{H}(\boldsymbol{x})\| = \|\sum_{i=1}^{d} (\ell''(u_i(\boldsymbol{x})) - \ell'(u_i(\boldsymbol{x}))) \boldsymbol{a}_i \boldsymbol{a}_i^\top\|$$

$$\leq \sum_{i=1}^{d} \ell'(u_i(\boldsymbol{x})) |1 - 2\ell'(u_i(\boldsymbol{x}))| \leq \sum_{i=1}^{d} \ell'(u_i(\boldsymbol{x})) \leq f(\boldsymbol{x}),$$

which corresponds to the previous case with $\xi_0 = 1$ and $\xi_1 = 0$. Due to convexity of $f$, we have

$$\|\nabla^2 f(\boldsymbol{x}) - \mathbf{H}(\boldsymbol{x})\| \leq f(x) \leq f^\star + D_0\|\nabla f(\boldsymbol{x})\|_*,$$

for all points from the initial sublevel set: $\boldsymbol{x} \in \mathcal{F}_0$, where all iterates of our algorithm belong to. $\qquad \square$

## H.4 RECOVERING COMPLEXITIES FOR PRACTICAL APPROXIMATIONS

**Contribution of the Degrees of $\pi$.** As we saw, the general form of our lower bound is given by structural assumption (10), for all problem cases, it appears to be the harmonic mean of simple monomials: $\pi(t)^{-1} = \sum_{i=1}^{d} M_{1-\alpha_i} t^{-\alpha_i}$, where $\alpha_i \in [0, 1]$ are some degrees that depend on the problem class and on the level of Hessian approximation $\beta$. For example, let us assume that $\pi(t)$ is the harmonic mean of two monomials (as, e.g. for $(L_0, L_1)$-functions (77)): $\pi(t) = (M_1 t^{-\alpha_1} + M_2 t^{-\alpha_2})^{-1}$, for some $M_1, M_2 > 0$ and $0 \leq \alpha_1, \alpha_2 \leq 1$. Then, for the non-convex case, the global complexity of the method is (Corollary 1): $K = O\left(F_0 \cdot \left[\frac{M_1}{\varepsilon^{1+\alpha_1}} + \frac{M_2}{\varepsilon^{1+\alpha_2}}\right]\right)$ iterations to solve the problem, where $\varepsilon > 0$ is the target accuracy for the gradient norm. We show that the fastest possible rate corresponds to the smallest degree, $\alpha := \min_{1 \leq i \leq d} \alpha_i$, while the other exponents correspond to additional slow terms. Notably, our proof is based on first selecting the smallest $\alpha$, to establish the progress, which highlights its importance.

**The definition of $\pi(\cdot)$.** This paragraph is an extended version of a short note in Section 5. The notion of $\pi$ (10) is a structural assumption on the global behavior of the Gradient-Normalized Smoothness $\gamma(\cdot)$. It is needed to translate our knowledge of a problem class to the complexity bounds in their standard form. Formally there could be many choices for $\pi$, while $\gamma(\cdot)$ is defined in a unique way. However, it is important that our method does not need to know the particular problem class or the particular $\pi$, and by implementing a simple adaptive search (Algorithm 3) the method becomes

parameter-free. Let us consider several important examples of known structures of the lower bound $\pi$.

- Function with $L$-Lipschitz Hessian (Example 1 with $\nu = 1$). Then, $\gamma(x) \geq (\frac{2}{L}\|\nabla f(x)\|)^{1/2}$. Hence, $\pi(t) = (\frac{2}{L}t)^{1/2}$, and the corresponding global complexity in the convex case (Theorem 2) is

$$
O\left(\sqrt{\frac{LD^3}{\varepsilon}}\right),
$$

where $\varepsilon > 0$ is the target accuracy for the functional residual, and $D$ is the diameter of the initial sublevel set.

- Convex functions with L-Lipschitz Third Derivative (Example 2 with $\nu = 1$). Then, $\gamma(x) \geq (\frac{1}{2L}\|\nabla f(x)\|)^{1/3}$. Hence, $\pi(t) = (\frac{1}{2L}t)^{1/3}$. The corresponding complexity of the method (Theorem 2) is

$$
O\left(\left[\frac{LD^4}{\varepsilon}\right]^{1/3}\right).
$$

- Quasi-Self-Concordant functions (Example 3). Then, $\gamma(x) \geq \frac{1}{M}$. Hence, $\pi(t) \equiv \frac{1}{M}$, and the corresponding complexity (Theorem 2) is

$$
O\left(MD\log\frac{1}{\varepsilon}\right) \qquad \textbf{(global linear rate of convergence)}.
$$

- $(L_0, L_1)$-smooth functions (Example 4). Then, $\gamma(x) \geq \frac{1}{1+e}\frac{\|\nabla f(x)\|}{L_0+L_1\|nabla f(x)\|}$. Hence, $\pi(t) = \frac{1}{1+e}\left[\frac{L_0}{t} + L_1\right]^{-1}$. Note that this expression also matches the structural assumption on $\pi$ in (10). And the corresponding complexity of the method becomes (Theorem 2):

$$
O\left((L_1D + \frac{L_0D^2}{\varepsilon})\log\frac{1}{\varepsilon}\right).
$$

We see that we recover the right complexities in all known special cases. Every particular problem class leads to the specific structure of the lower bound $\pi(\cdot)$. However, the power of our result is that we do not need to know and fix the problem class in the method, adapting to the best possible bound. One interesting example follows from the basic properties of $\gamma$ under simple operations (Section 2). Let us assume that our objective is represented as a finite sum of functions: $f(x) = \frac{1}{n}\sum_{i=1}^{n} f_i(x)$. Every function in the sum might belong to a different problem class, and therefore, every function $f_i$ might have a different lower bound $\pi_i(t)$ (e.g. as in the above examples). In this case, the whole objective does not belong to any 'standard' problem class, and therefore, simple assumptions such as Lipschitzness of the Hessian are not applicable in this case. However, the lower bound $\pi(\cdot)$ for the whole objective can be computed as the Harmonic mean of the lower bounds for the components:

$$
\pi(t) \geq \left(\sum_{i=1}^{n}\pi_i(t)\right)^{-1}
$$

**The effect of inexact Hessian.** This paragraph is also an extension of Section 5, designed to show how we derive complexities for a method with inexact Hessian using condition (12). If we assume (12), then the Gradient-Normalized Smoothness, when using the Hessian approximation, is bounded according to our rules, as:

$$
\gamma(\boldsymbol{x}) \geq (\gamma_1(\boldsymbol{x})^{-1} + \frac{\mathbf{C_1}}{\|\nabla f(\boldsymbol{x})\|} + \frac{\mathbf{C_2}}{\|\nabla f(\boldsymbol{x})\|^\beta})^{-1},
$$

where $\gamma_1(\boldsymbol{x})$ is the Gradient-Normalized Smoothness for the exact Hessian. In other words, if we know the lower bound $\pi(\cdot)$ for the exact Hessian (e.g. any of the problem classes above), then $\pi(\cdot)$ for the method with inexact Hessian can be computed in a form that satisfies the structural assumption 10:

$$
\pi(t) = \left(\frac{1}{\pi_1(t)} + \frac{\mathbf{C_1}}{t} + \frac{\mathbf{C_2}}{t^\beta}\right)^{-1}.
$$

And we immediately obtain the complexity result for the method with inexact Hessian (Corollaries 2 and 3). We see that the total complexity of the method becomes the sum of the complexity for the exact case plus two additional terms that depend on $\mathbf{C_1}$, $\mathbf{C_2}$, and the degree of approximation $\beta$.

One important consequence of our theory is that when $\mathbf{C_1} \approx 0$ is very small or zero, and $\beta < \alpha$, where $\alpha$ is the minimal degree of the monomial in the expression for $\pi$, then the rate of convergence is not affected by the Hessian inexactness (see also Figure 1). We see that these conditions hold, e.g., for the Fisher and Gauss-Newton approximations in several applications (Examples 6, 7, 8), where we have $\beta = 0$. Therefore, in these applications, the use of the inexact Hessian will give us *the same global rate* as the exact Newton method, while computation of every step is much more efficient.

Let us consider the concrete example with the logistic regression problem and the Fisher approximation matrix (3). The logistic regression is Quasi-Self-Concordant, and hence $\pi_1(t) \equiv \frac{1}{M}$ (Example 3), where $\pi_1(t)$ is the bound for the Gradient-Normalized Smoothness with exact Hessian. When using the Fisher approximation, we have (12) satisfied with $\mathbf{C_1} = f^\star$, $\mathbf{C_2} = D$, and $\beta = 0$ (Example 6). Therefore, the Gradient-Normalized Smoothness for the Fisher approximation is bounded as:

$$\gamma(\boldsymbol{x}) \geq \pi\left(\|\nabla f(\boldsymbol{x})\|\right), \quad \text{with} \quad \pi(t) = \left(M + D + \frac{f^\star}{t}\right)^{-1},$$

and the corresponding complexity becomes:

$$O\left(\left[MD + D^2 + \frac{f^\star D^2}{\varepsilon}\right] \cdot \log \frac{1}{\varepsilon}\right).$$

If $f^\star \approx 0$ (well separated data), this gives a very fast global linear rate. To the best of our knowledge, we are the first to establish such a rate for the inexact Newton method with the Fisher approximation matrix. Similar reasoning also work for applications with Nonlinear Equations (Example 7) and Soft Maximum (LogSumExp) (Example 8) with Gauss-Newton approximations.

Below, we present a formal statement, serving as a good example of the practical applicability of our notion. In Proposition 9, we show that our method (1) achieves a global linear rate of convergence on the logistic regression problem.

> **Proposition 9** (A global linear rate of convergence for the inexact Hessian.)**.** *Consider the logistic regression objective $f(\boldsymbol{x}) = \sum_{i=1}^{n} f_i(\boldsymbol{x})$, where $f_i(\boldsymbol{x}) := \log\left(1 + \exp\left(\langle \boldsymbol{a}_i, \boldsymbol{x}\rangle - b_i\right)\right)$. Then, for Algorithm 1 with*
>
> $$\mathbf{H}(\boldsymbol{x}) \quad := \quad \sum_{i=1}^{n} \nabla f_i(\boldsymbol{x}) \nabla f_i(\boldsymbol{x})^\top \quad \succeq \quad \mathbf{0}, \quad \textit{(Fisher approximation matrix)}$$
>
> *the corresponding complexity is*
>
> $$O\left(\left[MD + D^2 + \frac{f^\star D^2}{\varepsilon}\right] \cdot \log \frac{1}{\varepsilon}\right).$$
>
> *If the data is well separated ($f^\star \approx 0$), this gives a global linear rate.*

*Proof.* Assuming (12), the Gradient-Normalized Smoothness when using the inexact Hessian is bounded as

$$\gamma(\boldsymbol{x}) \quad \geq \quad \left(\gamma_1(\boldsymbol{x})^{-1} + \frac{\mathbf{C_1}}{\|\nabla f(\boldsymbol{x})\|} + \frac{\mathbf{C_2}}{\|\nabla f(\boldsymbol{x})\|^\beta}\right)^{-1}, \quad \text{(The ``\textit{Hessian inexactness}'' property)}$$

where $\gamma_1(\boldsymbol{x})$ is the Gradient-Normalized Smoothness for the exact Hessian. Since $f(\boldsymbol{x})$ is Quasi-Self-Concordant, $\gamma_1(\boldsymbol{x}) \geq \frac{1}{M}$. According to Example 6, $\mathbf{H}(\boldsymbol{x})$ satisfies condition (12) with $\mathbf{C_1} = f^\star$, $\mathbf{C_2} = D$, and $\beta = 0$. Then, we result in the following bound:

$$\gamma(\boldsymbol{x}) \quad \geq \quad \left(M + D + \frac{f^\star}{\|\nabla f(\boldsymbol{x})\|}\right)^{-1}.$$

According to Corollary 2, complexity for the method with inexact Hessian becomes:

$$K \quad = \quad O\left(\left[MD + D^2 + \frac{f^\star D^2}{\varepsilon}\right] \cdot \log \frac{1}{\varepsilon}\right).$$

Here, the term $MD \cdot \log \frac{1}{\varepsilon}$ corresponds to the previously established complexity for the method with exact Hessian (see Table 1). When the optimal value $f^\star \approx 0$, we result in the global linear rate of converge. $\square$

