# OpenReview forum: "Gradient-Normalized Smoothness for Optimization with Approximate Hessians"
_ICLR.cc/2026/Conference — ICLR 2026 Poster_

### Official Review · Reviewer_scSv · 2025-10-24

**Soundness:** 3
**Presentation:** 3
**Contribution:** 3
**Rating:** 4
**Confidence:** 3

**Summary:**

The paper present a very impressive general theory for gradient regularized Newton methods. The key observation is characterizing the amount of regularization $\gamma(x)$ through an implicit definition. What is also noteworthy is the many instantiations including Holder smooth Hessians, quasi-Self-Concordant,  $(L_0,L_1)$--smoothness, all of which provide a lower bound on this implicitly defined $\gamma(x),$ which is needed to execute the algorithm. The main issues I see are regarding computing $\gamma(x)$ and generally the fit within ICLR.

**Strengths:**

This is elegant work in deterministic optimization for Newton type methods. Both unifying previous theories, extending to in-exact Hessians, and developing many examples.

**Weaknesses:**

This seems to be mostly a follow up work of Doikov2023 and Doikov 2024a, which allows for inexact Hessians, and unifies some of these previous results. It's a dense literature, and difficult from reading this paper to know exactly what is new and what is not. Here I think the authors should write a more extensive background/related work section in the appendix, which makes it clear which parts are taken from which papers, and which parts are entirely new. The main paper itself does not make this very clear, and I had to open some of the prior work to get a sense of what had been done.

Another weakness is generally the fit with ICLR, its reviewing process, and to a lesser degree the audience.  Your paper and theory look very solid. But I regret to say, having it reviewed by ICLR is not a good fit. First the topic, which is full batch optimization. Even convex learning problems, such as GLMs, are only difficult to solve when there is so much data we need stochastic gradients. Your theory depends heavily on being in the deterministic setting. Furthermore, your result rely on the proofs, and your paper is (including proofs+ appendix), 48 pages long.  The turn around time of ICLR does not permit reviewers to go over your proofs in detail. I for one have only sampled your proofs. You would be much better served by submitting this substantial piece of work to an optimization journal (SIOPT, Math programming ...etc) where you would get a proper review.


Not really clear/transparent on limitations of Newton's method in machine learning. You introduction frames it as if Newton methods  are extensively used in ML. But the issues around dealing with stochastic estimates and non-convexity, makes most Newton type approach inapplicable.

**Questions:**

*Questions*

1. Implicitly defined $\gamma(x)$ and lower bounds.
In general, $\gamma(x,g)$ is implicitly defined, and needs to be computed or bounded to be used. The paper is clear about this, and gives many useful lower bounds for certain problem classes. But still seem some awkward issues around this. For instance, you give a type of adaptive line search procedure in Algorithm 3, and bound the number of oracle calls to $\mathcal{O}(K)$ where $K$ is the number of iterations of Algorithm K. But here you rely on the $\gamma_k$ computed by Algorithm 3 will satisfy the lowers bounds you give for $\gamma(x_k, F'(x_k)).$ Here I think you need a formal statement about the complexity of this adaptive line search, you can use it to guarantee that Algorithm 3 will have a meaningful complexity. Another clear issue is with your main result for convex functions in Theorem 2, which relies on the exact computation $\gamma_k = \gamma(x_k).$ Here I assume it is hard to characterize the complexity of solving this equation, and of course, exact solutions are probably inaccessible.




2. Lines 069-065: You claim that you will present a linear convergence result for when using inexact Hessians, and no strong convexity type assumptions. But where exactly is this result? The only linear convergence I see in the main paper is the instantiation of Theorem 2 for the class of Quasi-Self-Concordant functions, which relies on using the exact Hessian. Could you clarify this statement and/or correct it?


*Minor*

1. A small mistake you repeat is to write "...method equation 1" where all prepositions have been dropped. It should be "... the method in equation 1".

2. At a high level, it seems your gradient normalized smoothness does to Newton, what Directional smoothness, has done for gradient descent methods. That is, you identity an implicitly defined stepwise, that always guarantees descent, and generalizing other notations of smoothness including $(L\_0, L\_1)$-smooothness. This is exactly what the claims of the paper [] are, but for gradient based methods. So this in no way diminishes your contributions, it just seems it could be an interesting connection.

3. The assumption in eq (10) should be formally stated as such. It's a bit confusing how it is now stated. It would be much clearer if you have

Assumption 1.  There exists $0 \leq \alpha\_i \leq 1$ and $M_{1-\alpha\_i} \geq 0$ for $i=1, ..., d$ such that Eq (10) holds.

The same things goes for its counterpart eq (40) in the appendix.

4. Corollary 1. You "Under assumptions of Theorem 1 we can bound ..." But don't you mean, if equation (8) holds and Assumption 1 (above) hold? This is exactly one of the reasons I'm a bit confused about (10) being an assumption or not.

5. There are a few type-Os, quite a few missing prepositions, and similar minor mistakes throughout. You should take a pass with LLM or grammar check to fix theses.

---

> ### Author Response · Authors · 2025-11-21
>
> We are grateful for your expertise, time taken to review the paper, and for the feedback that helped to strengthen our manuscript.
> We are happy to address the mentioned issues, which we have also implemented with the careful revision of our paper.
>
> > the authors should write a more extensive background/related work section in the appendix
>
> We added the corresponding section to the appendix. Mainly, it highlights the differences between our findings and the results from Doikov 2023 and Doikov 2024a. We also discuss differences between our notion of Gradient-Normalized Smoothness and other types of recently studied first-order methods anisotropic and relative smoothness.
>
> > Your theory depends heavily on being in the deterministic setting
>
> Yes, our theory covers deterministic settings, and it is an open question how our concepts translate to the stochastic case, which we slightly cover in the discussions of the submitted manuscript. However, optimization papers with a deterministic analysis, especially on high-order methods, are also welcome at top-tier venues such as ICLR. In our opinion, this work is motivated by direct practical applications, and our theory provides a strict guarantee of the method on problems such as softmax, logistic regression, that are widely adapted in the field. Furthermore, we demonstrate the efficiency of our methods through numerical experiments on the mentioned problems and beyond them.
>
> > your paper is (including proofs+ appendix), 48 pages long
>
> We acknowledge that Reviewers are not obliged to revisit massive texts and proofs in the appendix. And we do not count on Reviewers to do so.
> We understand that the size of 48 pages for the paper, once the Reviewer sees it in his batch, does not look appealing. However, we would like to highlight that our long appendix is there to give a sufficient amount of detail regarding proofs and experiments to readers involved in the same research. We believe that a clear appendix with no hidden tricks would be helpful for the community of researchers working on similar topics. Additionally, most of the appendix is just an extended discussion of what we have covered in the main part, with the main proofs being quite compact. The main factors affecting the size of our appendix are:
> (1) 15 pages of our appendix are dedicated to experiments and experimental setup with a detailed description;
> (2) extensions of our theory from the main part to the composite optimization;
> (3) description of the algorithm with adaptive search;
> (5) different practical approximations that go beyond examples from the main text, and the enhanced bounds for them;
> (6) examples of the bounds on Gradient-Normalized Smoothness derived for different problem classes — to give a better understanding of how to work with our notion.
> Despite factors (1)-(6), our main proofs — Theorem 3 and Theorem 4 — are roughly 2 pages long. And main corollaries that recover rates are just “plug-in statements“ once the corresponding theorem has been proven.
>
> > introduction frames it as if Newton methods are extensively used in ML
>
> We acknowledge that Newton-type methods, especially with exact Hessian, cannot be applied to large-scale model training due to the unaffordable memory and computational costs.
> As a theoretical work, we believe we do not need to focus on the fact that second-order methods are not applied for large neural network training.
> Our work is just a step forward towards approximate second-order methods, which are practical.
>
> **Q2.**
>
> Sure. We added a section named “Recovering Complexities for Practical Approximations” to the Appendix, where we describe in more details how to obtain this result. Feel free to check.
> In a few words, Log Reg (as well as many other examples, e.g., Exp. Reg, Soft Maximum, Matrix Scaling, etc.) satisfy quasi Self-Concordance (QSC).
> Thus, our general complexity result for QSC functions can be applied. For the inexact Hessian, when using our condition (12), it appears that the error coming from the approximation can be bounded with a gradient norm if $f^\star$ is supposedly zero. This is a very powerful bound that results in the global linear rate. See a formal statement on Page 50. We also formulated a Proposition 9 regarding this case, see Page 55.
>
> **Minor questions.**
>
> Thank you for pointing us to those issues.
>
> * For comments (1,5) we did a few cleaning paths throughout the text and hope that we corrected all of them.
>
> * For comment (3), we are very thankful to the Reviewer for their willingness to improve our work, and we have changed the formulation of the Eq. 10 in this regard.
>
> * You are write about comment (4); when we wrote “under assumptions of Theorem 1”, we mean that the objective is non-convex, but the actual bound on $\gamma_*$ comes from our structural assumption in Eq. 10, we have corrected this issue in the text.
> Thank you for the connections with the directional smoothness notion. While the link

---

> > ### Author Response · Authors · 2025-11-24
> >
> > **Q1**
> >
> > Thank you for the thoughtful comments on our algorithm with adaptive search and Theorem 2.
> > In the newest revision of our manuscript, we tried to be even more clear about the practical applicability of our algorithms and that adaptive search is used in practice. We also formulate rigorous statements regarding the convergence of our method with adaptive search — see Theorem 4 (Section D.2).
> >
> > Before, in our paper, we studied the theoretical step-size rule (Def. 1) and the adaptive search that ensures the inequality from Lemma 1 that is described in Algorithm 3 (Section D.2).
> > In addition to the two presented choices for the second-order step-size $\gamma_k$, we also present a version of our method with the constant step-size $\gamma_k = \gamma^\star$ for a certain $\gamma^\star > 0$ (see Section D.1 and Theorem 3). Our value of $\gamma^\star$ is defined for a wide range of classes. However, despite seeming too conservative, this rule recovers all state-of-the-art rates for the Newton method with gradient regularization,
> > including those from Doikov 2024a and Doikov 2023. To the best of our knowledge, this constant choice is new.
> >
> > We believe we have clarified all concerns of the Reviewer regarding the method with adaptive search. We kindly invite the Reviewer to look at the newly added Theorems 3 and 4.

---

> ### Author Response · Authors · 2025-11-21
>
> * Thank you for the connections with the directional smoothness notion. While the link in your review is missing, we believe you have mentioned this paper “Glocal Smoothness: Line Search can really help!”. We do cite it now, along with the directional smoothness work in the revisited manuscript.
>
> All revisited issues in the text are colored blue.
>
> We believe we addressed Reviewer's concerns and made the paper more refined. We stay at their disposal to answer any further questions.

---

### Official Review · Reviewer_C1PU · 2025-10-30

**Soundness:** 4
**Presentation:** 4
**Contribution:** 4
**Rating:** 10
**Confidence:** 3

**Summary:**

The paper introduces a new assumption on Hessian inexactness and gradient-normalized smoothness. Under the gradient-normalized smoothness assumption, they provide the best-known rates of the Gradient-Regularized Newton method for (a) bounded Hessian, (b) Lipschitz Hessian, (c) third-order Lipschitz smoothness, (d) quasi self-concordant functions, and new rates for generalized self-concordant functions. Additionally, they introduce Gradient-Regularized Newton with Approximate Hessians and prove new convergence rates.

**Strengths:**

The paper is very well-written, accurate, and clear. The results are correct, new, and interesting for the community. It is an outstanding paper. I list several strengths of the paper:

1. The paper proposes new concepts of smoothness that generalize and easily connect many well-known and modern classes of functions. For this class of gradient-normalized smooth functions, they provide examples of functions satisfying it. For classical function classes, they provide a lower bound on the smoothness constant $\gamma(x)$.

2. The method with proposed step-sizes is universal and adapts to the problem class, achieving the best-known rates with second-order information. Moreover, the same method works with only first-order oracles by using inexact Hessians, and the paper provides new rates in this setting.

3. The experiments show the effectiveness of the proposed step-sizes.

**Weaknesses:**

I couldn’t find any major weakness in the paper. Minor:
1. While the numerical experiments show the effectiveness of the Gradient-Regularized Newton method with the proposed stepsizes, it would be beneficial to compare it with other adaptive or universal methods, both with exact and inexact Hessians.

**Questions:**

No questions.

---

> ### Author Response · Authors · 2025-11-21
>
> We thank the Reviewer for their time invested in our manuscript and for the positive feedback. We also appreciate the Reviewer’s willingness to improve our paper.
>
> To fulfil your issue, we have incorporated additional experiments in the Appendix of our work — all changes are coloured blue.
> We kindly ask the Reviewer to refer to the newly added section: “Comparison with Adaptive and Universal Methods”.
> There, we present comparisons of our method with adaptive search with exact and inexact Hessians and other adaptive / universal methods. Namely, we study two variations of the Damped Newton method with adaptive step-size rules — AICN [1] and GRLS [2] — to the best of our knowledge. Those methods achieve state-of-the-art rates among all algorithms based on the Damped Newton method . And we also run Cubic Newton and fast gradient descent. All methods are studied on three problems: LogSumExp, Chebyshev polynomials, and Nonlinear Equations. Overall, we see that all second-order methods perform consistently good, and our practical approximations within our adaptive search procedure for Algorithm 1 allow us to closely match the performance of second-order methods while making the algorithm formally a first-order — this is a new result that not only has been proven theoretically, but also verified on different problems in our numerical experiments. Feel free to check the new plots.
>
> We believe we have addressed the Reviewer’s concerns and enhanced the paper with additional experiments. We remain at your disposal to answer any further questions.
>
> [1] “A Damped Newton Method Achieves Global $O(k^{-2})$  and Local Quadratic Convergence Rate”.
>
> [2] “Newton Method Revisited: Global Convergence Rates up to $O(k^{-3})$ for Stepsize Schedules and Linesearch Procedures”.

---

### Official Review · Reviewer_tjcC · 2025-11-01

**Soundness:** 3
**Presentation:** 2
**Contribution:** 2
**Rating:** 6
**Confidence:** 3

**Summary:**

The paper introduces Gradient Normalized Smoothness as a unifying device to analyze approximate Newton methods. The framework couples objective smoothness with Hessian approximation quality and shows when an approximate Hessian can match exact Newton rates. The analysis recovers state of the art rates across several function classes and provides applications to logistic regression with Fisher and to softmax problems. The experiments are small but consistent with the theory.

**Strengths:**

1. The paper introduces a powerful new theoretical framework for developing optimization algorithms that leverage approximate second-order information while guaranteeing fast global convergence for both convex and non-convex objectives. The central innovation is the Gradient-Normalized Smoothness (GNS), a novel, universal notion that locally characterizes the maximum radius of a ball around the current point where the gradient field is well-approximated. This concept provides a unified mechanism to connect errors stemming from Hessian inexactness and Taylor's approximation, effectively translating local properties into the method's global performance without relying on predefined problem class specific parameters.

2. Algorithm 1 uses a simple gradient regularization structure where the step-size ($\gamma_k$) is chosen based on the GNS quantity, allowing it to adapt automatically to the objective's characteristics and the degree of the Hessian approximation. The theory successfully handles inexact Hessians $H_k$ provided they satisfy a condition related to the gradient norm (Equation 2). When the Hessian is exact, the framework recovers state-of-the-art global convergence rates for various function classes, including those with Hölder-continuous derivatives and quasi-Self-Concordant functions. Notably, the paper demonstrates that if the objective's smoothness degree ($\alpha$) dominates the approximation degree ($\beta$), the method with inexact Hessians achieves the same global rate as the full Newton method.

3. The practical effectiveness is validated using specific approximate Hessians popular in machine learning, such as Fisher and Gauss-Newton matrices, which satisfy the derived bounds and yield new global convergence results. Furthermore, experimental results suggest that Algorithm 1 using an inexact Hessian approximation (when aligned with the theory) is more numerically stable than the Exact Newton method, which can fail to converge on non-convex objectives due to ill-conditioning issues.

**Weaknesses:**

1. The notion of Gradient-Normalized Smoothness is defined mathematically but not well-motivated in terms of optimization geometry or curvature behavior. The crucial element appears to be the specific gradient-based normalization ((1/γ) ||g||* ||h||). The paper should elaborate on why this specific normalization is the key that unlocks the unified analysis. It is also unclear how this notion of smoothness compares to established notions like relative or anisotropic smoothness.

3. The theoretical framework fundamentally relies on the existence of γ(x_k), which is defined via a maximization condition but cannot be computed in practice. While the paper acknowledges this limitation and proposes an adaptive search procedure (Algorithm 3 in Appendix D) for implementation, the authors should be more upfront in the main text about this fact.

4. The structural assumption (equation 10) is key to connecting the local definition of GND to the global rates, but its introduction in section 4 feels abrupt and unexpected. It would help if it was alluded to and motivated in earlier sections.

**Questions:**

1. Could you clarify what makes the gradient-based normalization introduced in Definition 1(i.e., the term (1/γ) ||g||* ||h||) conceptually novel or more powerful than the traditional smoothness bounds used in analyses of cubically regularized Newton or trust-region methods? In particular, what analytical challenges does this specific normalization address or simplify that previous formulations do not?

2. Could you provide more intuition for why the harmonic mean structure is considered the “right” formulation? Does it emerge naturally from combining different smoothness assumptions, or is it primarily introduced as a flexible, unifying model chosen to encompass all the examples presented in the paper?

---

> ### Author Response · Authors · 2025-11-21
>
> We thank the Reviewer for their expertise, time taken to review, and for the positive feedback. We also think that it is better to incorporate additional comments in the text, thus we present a clarified revision with new text being coloured blue. Below, we answer mentioned questions and weaknesses.
>
> **W1.**
> > why this normalization is used
>
> The notion of Gradient-Normalized smoothness has a clear geometric/trust-region interpretation. $\gamma(x, g)$ is literally “how far you can trust the linear model” in directions that matter (the local region (O_{x,g})). The inequality enforces that within that radius the model error is small compared to the directional change you care about ($|| g ||_*$ $|| h ||$). Our type of normalization makes the same condition meaningful when $H$ is the full Hessian, zero (we get the normalized GD in this case) or some other matrix, possibly a Hessian approximation.
> Additionally, since our condition in Def. 1 leverages the relative approximation error coming from the linearization of the gradient, rather than relying on particular smoothness constants, a single step-size based on our framework yields the correct scaling for many problem classes automatically.
>
> > comparison with relative and anisotropic smoothness
>
> Both relative and anisotropic smoothness notions are designed specifically for first-order methods. Unlike those notions, our Gradient-Normalized Smoothness naturally captures second-order behavior and remains invariant under quadratic perturbations of the function.
> Indeed, consider each example.
>
> **Relative Smoothness.** Let $f$ be a relative smooth with parameter L w.r.t. some $\mu$-strongly convex reference function $h$, i.e., for any $x, y \in \mathbb{R}^d$ $f(y) \leq f(x) + \langle \nabla f(x), y-x\rangle +LD_h(y,x)$, where $D_h(y,x)$ is a Bregman divergence. Let $\psi(x) = f(x) + \frac12 x^\top A x$  for some matrix $A \succeq 0$ --- a quadratic perturbation of $f$.
> Using the relative smoothness of $f$ we can conclude that
> $\psi(y) - \psi(x) - \langle \nabla\psi(x), y-x\rangle \leq LD_h(y,x) + \frac12 (y-x)^\top A(y-x)$. To ensure that $\psi$ is relative smooth with some parameter $L^\prime$, we require the bound $\frac12 (y-x)^\top A(y-x)\leq(L^\prime - L)D_h(y,x)$, using the strong convexity of $h$ and that RHS<=$\frac12 \| A\| \|y-x\|^2$, choosing $L^\prime = L + \frac{\|A\|}{\mu}$ ensures $\psi(y) \leq \psi(x) + \langle \nabla\psi(x), y-x\rangle + LD_h(y,x)$, so $\psi$ is relative smooth to $h$ with parameter $(L + \frac{\|A\|}{\mu})$.
>
> **Anisotropic Smoothness.** We follow the definition from the "AdaGrad under Anisotropic Smoothness" paper. Let $f$ be an anisotropic L-smooth with diagonal matrix $L$ with positive entries, i.e., for any $x, y \in \mathbb{R}^d$  $|| \nabla f(x) - \nabla f(y) ||_{L^{-1}} \leq || x - y ||_L$.
>
> Consider a quadratic perturbation
> $\psi(x) = f(x) + \frac12 x^\top A x$.
> Denote $z=x-y$.
> Then for some diagonal $M$, using that $f$ is anisotropic L-smooth, we have $|| \nabla \psi(x) - \nabla \psi(y)||^2_{M^{-1}} \leq 2|| \nabla f(x) -\nabla f(y) ||^2_{M^{-1}}+2||Az||_{M^{-1}}^2 \leq 2\lambda ||z||^2_L + 2z^\top A^\top M^{-1}Az$, where $\lambda = \max_i \frac{L_i}{M_i}=|| M^{-1/2} L^{1/2}||^2_2$.
>
> Choosing $M=\frac{1}{\alpha}L$ and $M^{-1} = \lambda L^{-1}$, and requiring the RHS to be <= $\lambda z^\top L z$ for all $z$, we obtain that $M=L/  \lambda$ is a valid choice, if $1/\lambda^2 \geq 2(1+||L^{-1/2}AL^{-1/2}||^2_2)$.
> We see that in the one-dimensional case, when Hessian is a scalar $a>0$, the spectral norm $L^{-1/2}AL^{1/2}$ becomes $a/L$, and we have $1/\lambda = 1 + a /L$, therefore $M=L+a$.
>
> Thus, both notions are affected by quadratic perturbations.
>
> **Our notion.** Introduce the same quadratic perturbation to $f$.
> Then denote matrices $H_f$ and $H_\psi$.
> Then, according to Definition 1 from the paper for $f$, it is clear that for any $x\in\mathbb{R}^d$ and direction
>
> $g\in\mathbb{R}^d$
>
> $|| \nabla \psi(x+h) - \nabla \psi(x) - H_{\psi}(x)h||_* \leq \frac{1}{\gamma} ||g|| ||h||$
>
> for any $h \in B_\gamma \cap O_{x,g}$.
>
> Importantly, $O_{x,g}$ restricts the set of directions $h$ we need to consider. Define $O_{x,g}^\psi$ a local region where the norm is induced by the Hessian of $\psi$. With a little abuse of notations, consider this set for function $\psi$, i.e. such $h$ that $||h||^2_\psi + \langle g, h \rangle \leq 0$. Since $|| h||^2_\psi = h^\top (H(x)+A)h\geq h^\top H(x)h = ||h||^2_x$, when $A\succeq0$, we have that the local region for $\psi$ is a subset of the local region for $f$! Additionally, from our monotonicity observations (Line 203), if the inequality holds for all $h$ in some set $S$, then it holds for any $h$ in any subset of S. Therefore, if the Gradient-Normalized Smoothness condition holds for $f$ with parameter $\gamma$ over $O_{x,g}$, then it automatically holds for $\psi$ with the same $\gamma$ over $O_{x,g}^{\psi}$.

---

> ### Author Response · Authors · 2025-11-21
>
> **W2.**
> We tried to highlight this in the main formal statements when mentioning the adaptive search procedure — Corollary 1 and Theorem 1. And in Lines 300-301 below the description of Algorithm 1. We do agree with the Reviewer that it is better to clarify that in practice we propose using our adaptive search procedure to ensure the inequality from Lemma 1 at every iteration of the method. Because we cannot put the Algorithm 3 in the text, we decided to add a comment regarding adaptive search to the line 3 of Algorithm 1.
>
>
> **W3 & Q2.**
> Thanks for pointing us to this. We tried to motivate the Equation 10 (harmonic mean) more in the earlier sections, using the examples of lover bounds of $\gamma(x)$ for different problem classes.
> All changes are coloured blue. To be more clear in the rebuttal, we provide a more detailed explanation here as well.
>
> Equation 10 gives us a structural assumption on the global behaviour of the Gradient-Normalized Smoothness $\gamma(\cdot)$ in a form of the lower bound $\pi$. It is needed to translate our knowledge of a problem class to the complexity bounds in their standard form. Formally, there could be many choices for the lower bound $\pi$, while $\gamma(\cdot)$ is defined in a unique way. But because of the adaptive search our method does not need to know the particular problem class or the particular $\pi$, thus, it becomes parameter-free. Once it has been clarified, let us consider several important examples of known structures of the lower bound $\pi$.
>
> * Function with L-Lipschitz Hessian (Example 1 with $\nu = 1$). Then,
>
> $\gamma(x) \geq ( \frac{2}{L} || \nabla f(x) || )^{1/2}$. Hence, $\pi(t) = ( \frac{2}{L} t )^{1/2}$, and the corresponding global complexity in the convex case (Theorem 2) is $O\Bigl(\left( \frac{LD^3}{\varepsilon}\right)^{1/2} \Bigr)$, where $\varepsilon > 0$ is the target accuracy for the functional residual, and $D$ is the diameter of the initial sublevel set.
>
> * Convex functions with L-Lipschitz Third Derivative (Example 2 with $\nu = 1$). Then,
>
> $\gamma(x) \geq ( \frac{1}{2L} || \nabla f(x) || )^{1/3}$. Hence, $\pi(t) = ( \frac{1}{2L} t )^{1/3}$. The corresponding complexity of the method (Theorem 2) is
> $O\left( \left(\frac{LD^4}{\varepsilon}\right)^{1/3} \right) $.
>
> * Quasi-Self-Concordant functions (Example 3). Then,
>
> $\gamma(x) \geq \frac{1}{M}$. Hence, $\pi(t) \equiv \frac{1}{M}$, and the corresponding complexity (Theorem 2) is $O( MD \log \frac{1}{\varepsilon} )$ (global linear rate of convergence).
>
> * (L_0, L_1)-smooth functions (Example 4). Then,
>
> $\gamma(x) \geq \frac{1}{1 + e} \frac{|| \nabla f(x) ||}{L_0 + L_1 || \nabla f(x) ||}$. Hence, $\pi(t) = \frac{1}{1 + e}\Bigl[ \frac{L_0}{t} + L_1 \Bigr]^{-1}$. Note that this expression also matches the structural assumption on $\pi$ in (10). And the corresponding complexity of the method becomes (Theorem 2):
> $O\left( \left(L_1D + \frac{L_0 D^2}{\varepsilon} \right) \log \frac{1}{\varepsilon} \right)$.
> We see that we recover the right complexities in all known special cases. Every particular problem class leads to the specific structure of the lower bound $\pi(\cdot)$. However, the power of our result is that we do not need to know and fix the problem class in the method, adapting to the best possible bound.
> One interesting example follows from the basic properties of $\gamma$ under simple operations (Page 5). Let us assume that our objective is represented as a finite sum of functions:
> $ f(x) = \frac{1}{n} \sum_{i = 1}^n f_i(x) $. Every function in the sum might belong to a different problem class, and therefore, every function $f_i$ might have a different lower bound $\pi_i(t)$ (e.g. as in the above examples).
> In this case, the whole objective does not belong to any 'standard' problem class, and therefore, simple assumptions such as Lipschitzness of the Hessian are not applicable in this case.
> However, the lower bound $\pi(\cdot)$ for the whole objective can be computed as the Harmonic mean of the lower bounds for the components:
> $\pi(t) \geq  \Bigl( \sum\limits_{i = 1}^n \pi_i(t)  \Bigr)^{-1}$
> Thus, we automatically obtain the complexity of minimizing $f(x)$, while it is difficult or even not possible to specify a simple problem class for $f$. Again, this complexity is achieved automatically by the method with an adaptive search, and we do not need to know the exact form of $\pi(t)$ in practice.

---

> ### Author Response · Authors · 2025-11-21
>
> **To sum up the long answer.**
> Since we want $\gamma_k = \gamma(x_k)$ to have a physical meaning of the radius of the ball around $x_k$ (as in the trust-region methods) — Answer to **W1** —  we rewrite the step of the regularized Newton method $x_{k+1} - x_k = (H + \lambda_k B)^{-1}\nabla f(x_k)$ in another parameterization, making $\lambda_k = || \nabla f(x_k) || / \gamma_k$. Then, due to the *sum of functions* property of Gradient-Normalized Smoothness, it naturally appears that $\gamma(x_k)$ is lower bounded by the harmonic mean.
> Additionally, following the same reasoning as for the sum of functions, our notion clearly describes the case of the inexact Hessian.
>
> We believe we addressed Reviewer’s concerns. We stay at their disposal to answer any further questions.

---

### Official Review · Reviewer_F4Yq · 2025-11-01

**Soundness:** 3
**Presentation:** 3
**Contribution:** 3
**Rating:** 6
**Confidence:** 2

**Summary:**

This paper introduces the concept of Gradient-Normalized Smoothness (GNS) as a new local characterization of the gradient field and Hessian approximation in optimization, particularly for the design and convergence analysis of second-order methods with approximate Hessians. Building on this notion, the authors propose a global convergence framework for both convex and non-convex optimization, unifying the treatment of smoothness and approximate second-order information under a single analytic lens. The work claims to recover state-of-the-art complexities for a wide range of problem classes—including Hölder-smooth, self-concordant, and generalized smoothness settings—and demonstrates the practical advantages and competitive empirical performance of the approach in numerous experiments, particularly with logistic regression, softmax, and non-convex losses.

**Strengths:**

**1. Unified Theoretical Framework**: The introduction and analysis of Gradient-Normalized Smoothness offers a conceptually appealing unification of smoothness and Hessian approximation error, allowing the same framework to recover or extend the best known rates for several classes of optimization problems.

**2. Strong Experimental and Visual Evidence.**: The paper provides thorough experimental validation with clear visualizations, which is easy to follow, and I like the layout of the paper (e.g., highlighting of some important examples and definitions).

**Weaknesses:**

I am not an expert in optimization; therefore, from my perspective, the paper does not exhibit any specific weaknesses. I will finalize my rating after reviewing other reviewers’ comments and the authors’ responses.

**Questions:**

**Q1.** The main challenge in second-order optimization lies in the high cost of computing the Hessian matrix. However, in practice, the Hessian–vector product can be computed efficiently without forming the full matrix by simply applying automatic differentiation twice. Could the authors briefly explain why such techniques are not suitable or sufficient for the tasks considered in this paper?

---

> ### Author Response · Authors · 2025-11-21
>
> We are grateful for the time that the Reviewer invested in our work and for the positive feedback. Below we answer the Reviewer's question.
>
> Indeed, the Hessian–vector product can be computed efficiently by differentiating the gradient a second time using automatic differentiation. Frameworks like PyTorch achieve this through a dynamic computational graph that supports higher-order derivatives, while JAX records a symbolic representation of the function by tracing its operations and then applies automatic differentiation to this traced computational graph.
>
> We confirm that all these techniques are suitable for our algorithm. However, for simplicity in presenting our theoretical results, we assume that the linear system (Line 3 in Algorithm 1) is solved exactly at every iteration. This is a standard assumption in optimization theory. At the same time, it is not too restrictive; similar convergence rates can be proved even when the solution is computed inexactly.
>
> In such cases, one can use any practical method for solving these systems, e.g., the conjugate gradient algorithm, which requires only Hessian-vector products (via two backward passes in PyTorch), and does not require computing the full Hessian. We also note that such solvers typically have fast rates of convergence and need only a few Hessian-vector products at each iteration due to the strong convexity of the objective.
>
> We have additionally highlighted these details in the text for practitioners; previously this was  briefly reflected in Lines 302-308.
>
> We believe we addressed Reviewer's concern. We stay at their disposal to answer any further questions.

---

> > ### Comment · Reviewer_F4Yq · 2025-11-28
> >
> > Thank you for your clarifications. My concerns have been addressed, and I will maintain my already positive rating for this paper.

---

### Meta-Review · Area_Chair_1Pok · 2025-12-16

**Summary:**

The reviewers agree that this is a nice work with a new theoretical framework that captures a vast range of problems, recovering the previous state-of-the-art rates and providing the new ones. I agree that this is solid work that is recommended for acceptance. I suggest that the authors consider all reviewers' comments in preparing the camera-ready paper.

**Reviewer Concerns:**

Almost all concerns are addressed by the reviewers. The only important non-addressed weakness, raised by Reviewer scSv, is the extension of this work to the stochastic case. Nevertheless, given the nature of this work and its new results in high-order optimization, even with this weakness, it is important to the ICLR optimization community.

**Reviewer Scores:**

Most scores are already high and indicate acceptance of this paper.

---

### Decision · Program_Chairs · 2026-01-26

Accept (Poster)